

# Generalised Gibbs ensemble
# for spherically constrained harmonic models

Damien Barbier[1,2], Leticia F. Cugliandolo[2,3], Gustavo S. Lozano[4] and Nicolás Nessi [5]

**1** Ecole Polytechnique Fédérale de Lausanne, Information, Learning and Physics lab and
Statistical Physics of Computation lab, CH-1015 Lausanne, Switzerland
**2** Sorbonne Université, Laboratoire de Physique Théorique et Hautes Energies,
CNRS UMR 7589, 4 Place Jussieu, 75252 Paris Cedex 05, France
**3** Institut Universitaire de France, 1 rue Descartes, 75231 Paris Cedex 05, France
**4** Departamento de Física, Universidad de Buenos Aires, and IFIBA CONICET, Argentina
**5** Instituto de Física de La Plata CONICET, Diagonal 113 y 64 (1900) La Plata, Argentina

## Abstract

We build and analytically calculate the Generalised Gibbs Ensemble partition function of the integrable Soft Neumann Model. This is the model of a classical particle which is constrained to move, on average over the initial conditions, on an $N$ dimensional sphere, and feels the effect of anisotropic harmonic potentials. We derive all relevant averaged static observables in the (thermodynamic) $N \to \infty$ limit. We compare them to their long-term dynamic averages finding excellent agreement in all phases of a non-trivial phase diagram determined by the characteristics of the initial conditions and the amount of energy injected or extracted in an instantaneous quench. We discuss the implications of our results for the proper Neumann model in which the spherical constraint is imposed strictly.



# 1  Introduction

This paper is mainly devoted to the analytic calculation of the partition function of the classical integrable Soft Neumann Model in the Generalized Gibbs Ensemble (GGE). From it we derive all relevant averaged static observables. The motivation for this study stems from the interest in characterising the stationary measure that integrable, though non purely quadratic, macroscopic models may reach after instantaneous quenches. Most of previous similar studies focused on quantum models, and several review articles summarise methods and results [1–9]. Less attention has been paid to classical out of equilibrium macroscopic integrable systems [10–24] and we contribute here to their better understanding.

The concrete problem that we chose to analyse is very rich. On the one hand, it relates to fundamental physics concepts, such as the meaning of ergodicity. On the other hand, the selected model connects to pure mathematics. Indeed, Neumann's system [25] is intimately related to celebrated non-linear wave equations [26, 27], among other problems which we will shortly review below. A further relation is to the disordered systems area since the potential harmonic energy maps to the one of the spherical Sherrington-Kirkpatrick spin-glass [28].

Given the large breath of the present study, we organise this Introduction in five separate parts which develop: the ergodicity statement, our program, the model, our goals and main results and, finally, the layout of the paper.

*Ergodicity*

The thermalisation properties of large dimensional classical Hamiltonian systems have regained interest in recent years. This renewed attraction has been boosted by the aim to reach a better understanding of similar issues in the quantum realm. Of particular interest are macroscopic classically integrable systems [29–31] for which the approach to Gibbs-Boltzmann equilibrium is not ensured and alternative asymptotic measures could be the relevant ones in the stationary state.

Typical observables in macroscopic *isolated* integrable classical models are described by a Generalised Microcanonical Ensemble (GME) in which the value of all independent constants

of motion are fixed [32]. More explicitly, their long-time averages

$$\overline{A} = \lim_{\tau \to \infty} \lim_{t_{\text{st}} \gg t_0} \tau^{-1} \int_{t_{\text{st}}}^{t_{\text{st}}+\tau} dt' A(t'), \tag{1}$$

with $t_{\text{st}}$ the time needed to reach stationarity (which could scale with the system size), and their statistical averages calculated with the flat GME measure,

$$\langle A \rangle_{\text{GME}} = \sum_{\text{conf}} A \rho_{\text{GME}} \qquad \text{with} \qquad \rho_{\text{GME}} = \frac{\prod_{\mu=1}^{N} \delta(I_\mu - I_\mu(0))}{\sum_{\text{conf}} \prod_{\mu=1}^{N} \delta(I_\mu - I_\mu(0))}, \tag{2}$$

where the sums run over all allowed phase space variables and the thermodynamic limit is taken, coincide. Here, $I_\mu$ are the phase space expressions of the constants of motion and $I_\mu(0)$ are the values they take at the initial time $t = 0$. In a classical integrable system there are as many constants of motion as degrees of freedom and, consequently, they constrain the phase space manifold visited by the dynamics in a much more restrictive way that in a standard non-integrable system in which there are only a few conserved quantities, *e.g.* the total energy, the linear and angular momentum, *etc.*

In conventional equilibrium situations, it is usually much more convenient to invoke ensemble equivalence and use a canonical representation to calculate statistical averages of local observables. The natural proposal for the canonical GGE measure is

$$\rho_{\text{GGE}} = Z_{\text{GGE}}^{-1} e^{-\sum_\mu \gamma_\mu I_\mu}. \tag{3}$$

The $\gamma_\mu$ are as many Lagrange multipliers as constants of motion, and they are fixed by requiring that the phase space averages of the $N$ constants of motion, $\langle I_\mu \rangle_{\text{GGE}}$, be equal to their values at the initial conditions, $I_\mu(0)$. However, it is not obvious that the expression (3) can be derived from the GME distribution in (2).

The challenge is, then, to construct the GGE of a classical integrable model of non-trivial kind, that is, one that is not just an ensemble of independent harmonic oscillators. Once this done, if the Newtonian dynamics of the model in question were also solvable, one should put to the test the main GGE claim: that in the stationary limit[1] the long-time average, $\overline{A}$ in Eq. (1), and the phase space average,

$$\langle A \rangle_{\text{GGE}} = Z_{\text{GGE}}^{-1} \sum_{\text{conf}} A e^{-\sum_\mu \gamma_\mu I_\mu} \tag{4}$$

calculated in the $N \to \infty$ limit, coincide (for any non explicitly time dependent, non pathological and in some sense local observable $A$),

$$\overline{A} = \langle A \rangle_{\text{GGE}}. \tag{5}$$

In this paper we calculate exactly the GGE measure of a non-trivial integrable classical model, the Soft Neumann model, and we show the equivalence between time averages and statistical averages, by calculating the former with mixed analytic-numerical methods. In the rest of the introduction we give a more extended background to our study.

*The program*

---

[1]The time $t_{\text{st}}$ is the time-scale needed to reach stationarity and it will typically be much longer than a microscopic time-scale $t_0$.

The out of equilibrium dynamics of systems of interest are usually studied by performing quenches, that is, sudden changes of a control parameter. The dynamics of macroscopic *open* and *non-integrable* systems following such quenches have been studied for more than 50 years. Some of the questions asked in this context are: Does a system reach a stationary state? If it does, which is the stationary measure that describes the time average of typical observables? Are thermodynamic concepts playing a role during the approach to the asymptotic state and/or when the system has reached it? These questions have been addressed with analytic, numeric and experimental means in a host of out of equilibrium open classical situations and an interesting picture of critical relaxation [33,34], phase ordering kinetics [35–39], and glassy dynamics [40–42] among other cases has emerged out of these studies. It is to be noted that in all the circumstances just cited the systems remain out of equilibrium forever if the thermodynamic limit is taken from the outset and times are not conveniently scaled with system size.

Knowing that some classical open macroscopic systems, as the ones mentioned in the previous paragraph, can remain far from equilibrium in their thermodynamic limit, one can expect this to happen, and even more so, to *classical closed integrable macroscopic* systems which cannot act as a thermal bath on themselves [29–31]. One may then wonder whether a GGE description could apply to the long-term evolution of local observables in such classical integrable systems.

We launched a program to study this question in a series of papers recently published [10–12, 43]. We picked a family of analytically solvable, but yet non-trivial (mean-field) models which, when evolved with stochastic dynamics due to their coupling to an environment, present rich relaxation dynamics, and do not reach thermal equilibrium on ample variation of the control parameters [40–42]. Since we expected to find interesting behaviour for Newton dynamics, we adapted the quenches to follow the system's evolution in isolation. More concretely, we took thermalised initial conditions at a chosen temperature, we instantaneously switched off any possible connection to a bath, and we let them evolve under classical mechanics rules. We first focused on a non-integrable case, the so-called $p = 3$ spherical disordered spin model with Newtonian dynamics. We showed that this model can act as a bath on itself and equilibrate for certain values of the parameters, while it keeps its glassy properties for others even when evolved in isolation [43]. Next, we identified a classical integrable model with non-trivial dynamics, the Neumann Model, and we introduced its soft version, the Soft Neumann Model, which is easier to treat analytically. Interestingly, this model is connected to the spherical $p = 2$ or Sherrington-Kirkpatrick model [10–12] and its dynamical version introduced in [44, 45] of the disordered systems literature. Due to its almost harmonic character, this model appears as the simplest non-trivial integrable macroscopic system.

We give below a short introduction to the definition of the Soft Neumann Model and some of its more relevant properties in the context of our study.

*The model*

The Neumann Model (NM) describes a classical point-like massive particle constrained to move on a sphere embedded in an $N$ dimensional space, under the effect of an anisotropic harmonic potential [25]. The spring constants along the principal axes are fixed parameters which characterise the potential energy and the kinetic energy is of the usual kind. The NM is an integrable model for which the explicit form of the $N$ constants of motion in involution $I_\mu$ with $\mu = 1, \ldots, N$ are known as quartic functions of the phase space variables [46, 47]. In the mathematical physics community it has been studied for small number of variables [48–50]. In its soft version, which we introduced in [10–12] and we call the Soft Neumann Model (SNM), the spherical constraint is imposed only on average over the initial conditions. Concretely, the initial conditions are drawn from a probability distribution and they are forced to satisfy

the spherical constraint on average. Moreover, the trajectories are also required to verify this constraint on average over the initial configurations.

Models of free oscillators, constrained by a quadratic function of the phase space variables, are integrable and give rise to celebrated non-linear wave equations such as the Korteweg-deVries, non-linear Schrödinger, Sine-Gordon, and Toda lattice equations [26, 46]. They also provide a way to solve "inverse spectral problems" which consist in finding, from a given discrete spectrum, the potential from which it originates, in cases in which the spectrum is a finite band one. This kind of models appear in other areas of physics as well. As shown in [51], and later developed in the string theory literature, a large class of classical solitonic solutions of the classical type IIB string action in the $AdS_5 \times S_5$ background can be classified in terms of solutions of the Neumann integrable system.

The NM and SNM can also be thought of as the classical mechanics extensions [44, 45] of a statistical physics model with only potential energy which was originally introduced as the "simplest" spin-glass [28] and it was later recognised to be a mean-field model for the easier paramagnetic-ferromagnetic transition. In this interpretation, the spring constants are the eigenvalues of a symmetric interaction matrix which couples the (real) spins in a fully connected way. For independent Gaussian couplings between real spins, this model is called the spherical Sherrington-Kirkpatrick or $p = 2$ spherical. In this case, the eigenvalues are distributed according to the Wigner semi-circle law in the infinite size limit. A quite extended list of papers with descriptions of the conventional equilibrium [28, 52–61] and stochastic relaxation [62–70] of this model can be found in the bibliography. This model also appears as a classical limit of the SYK model [71].

The constrained random harmonic potential is sufficiently complex to allow for a phase transition in the ensemble of equilibrated initial conditions that we use. Those belonging to one or the other phase will subsequently evolve after the quench, and dynamic phase transitions will thus be generated. All in all, the Newton dynamical system has a non-trivial phase diagram partially figured out in [10–12] with a variety of methods.

In [10, 11] we adapted techniques from the mean-field disordered systems literature to derive Schwinger-Dyson equations coupling overall correlation and linear response functions averaged over initial conditions, in the thermodynamic limit. From their analysis we identified three dynamic phases differentiated by whether the trajectories depart macroscopically from the initial positions or not, and the susceptibility to infinitesimal perturbations. However, even in equilibrium, a complete understanding of the macroscopic behaviour of this model needs to monitor the components of the particles position, $s_\mu$ with $\mu = 1, \ldots, N$ and, in the case of Newtonian dynamics, the knowledge of the momentum components $p_\mu$ is also necessary to complete the picture. In the $N \to \infty$ limit, these becomes functions of a continuous variable $\lambda$.

*Goals & main results*

The goals of this paper are twofold. On the one hand, we solve the dynamics in a mode-resolved way which allows us to compute the time-averaged $\overline{\langle s_\mu^2 \rangle}_{i.c.}$ and $\overline{\langle p_\mu^2 \rangle}_{i.c.}$ for finite though rather large $N$. On the other hand, and most importantly, we construct the GGE measure and we calculate exactly in the $N \to \infty$ limit the static mode averages, $\langle s^2(\lambda) \rangle_{GGE}$ and $\langle p^2(\lambda) \rangle_{GGE}$. We then compare these expressions to the dynamic ones. In short, our main results are the following.

1. First of all, with a mixed analytic-numeric treatment we exhibit that in the large size limit, $N \gg 1$, and in the long-time limit, $t \gg t_{st}$, with $t_{st}$ a characteristic time-scale which possibly scales with $N$, the SNM reaches a stationary state. More precisely, we show that in this long time limit the averages $\overline{\langle s_\mu^2 \rangle}_{i.c.}$ and $\overline{\langle p_\mu^2 \rangle}_{i.c.}$ take constant values.

2. We revisit the dynamic phase diagram of the SNM by studying the mode resolved evolution. We deduce that it presents four phases which we call "extended", "coordinate quasi-condensed", "coordinate condensed", and "coordinate and momentum quasi-condensed" depending on how the particle covers the sphere and the scaling of $\overline{\langle s_N^2 \rangle}_{i.c.}$ and $\overline{\langle p_N^2 \rangle}_{i.c.}$ with $N$. These are the projections in the direction with the largest spring constant (the edge eigenvector of the Gaussian interaction matrix in the spin model interpretation).

3. Importantly enough, we verify that the asymptotic states in all four phases of the SNM phase diagram satisfy a generalised ergodic hypothesis with the GGE measure, $\overline{\langle s_\mu^2 \rangle}_{i.c.} = \langle s_\mu^2 \rangle_{\mathrm{GGE}}$ and $\overline{\langle p_\mu^2 \rangle}_{i.c.} = \langle p_\mu^2 \rangle_{\mathrm{GGE}}$ where the dynamic averages are computed numerically over sufficiently long time windows.

4. We identify two kinds of initial conditions, both consistent with the spherical constraint, and we call them *symmetric* and *symmetry broken*. The fluctuations of the spherical primary and secondary constraints behave differently for these two groups. In one phase of the SNM phase diagram we evidence condensation phenomena [72–74] for symmetric initial conditions and macroscopic fluctuations which make the connection with the NM invalid.

Our results demonstrate that a meaningful GGE measure can be constructed for a non-trivial classical integrable model. One of the reasons why this model is interesting is that it is interacting, in the sense that it is not simply mappable on an ensemble of independent harmonic oscillators, and its constants of motion are non-trivial quartic functions of the space phase variables. Moreover, in its original formulation it involves long-range interactions. For these two reasons it was not obvious *a priori* that a canonical measure could be applicable. Recall that in long-range interacting systems the notion of a subsystem is not straightforward and the additivity of the conserved quantities is not justified either [75, 76].

*Layout*

The paper is structured as follows. In Sec. 1 we present the three models we use and the relations between them; namely, the Neumann Model (NM), the Soft Neumann Model (SNM) and the spherical Sherrington-Kirkpatrick model also called $p = 2$ spherical in the disordered systems literature. The next Sec. 3 explains the initial conditions that we choose and quench protocol that we implement. In Sec. 4 is the core of our paper: we introduce here the GGE measure and the harmonic *Ansatz* which allows us to evaluate it in the large $N$ limit. Next, in Sec. 5 we derive exact expression for the relevant averaged observables and Lagrange multipliers. The comparison between the dynamic and GGE observables is presented in Sec. 6. Section 7 is devoted to the analysis of the fluctuations in the GGE and dynamic formalisms, especially in cases in which there is condensation. Finally, in Sec. 8 we present our conclusions and we discuss related studies which appeared recently in the literature. The paper is complemented by several Appendices in which we provide technical details.

## 2 The model

In this Section we introduce the model and we relate it to two well-known problems in the integrability and disordered systems literature: the Neumann and the spherical Sherrington-Kirkpatrick (or so-called $p = 2$) models, respectively.

We are concerned with the motion of a particle constrained to stay, on average over the initial conditions, on the $N-1$ dimensional sphere. Calling $\vec{s}$ and $\vec{p}$ its position and momentum,

and $s_\mu$ and $p_\mu$ their projections on orthogonal directions $\vec{v}_\mu$, with $\mu = 1, \ldots, N$, the mean spherical constraints read

$$\langle \phi \rangle_{i.c.} \equiv \sum_{\mu=1}^{N} \langle s_\mu^2 \rangle_{i.c.} = N \,, \qquad \langle \phi' \rangle_{i.c.} \equiv \sum_{\mu=1}^{N} \langle s_\mu p_\mu \rangle_{i.c.} = 0 \,, \tag{6}$$

where $\langle \ldots \rangle_{i.c.}$ represents the average over the initial conditions, distributed according to a phase-space probability density $\rho_{i.c.}(\{s_\mu(0)\}, \{p_\mu(0)\})$. The particle is not free, but it is subject to an anisotropic quadratic potential,

$$V(\vec{s}) = -\frac{1}{2} \sum_{\mu=1}^{N} \lambda_\mu s_\mu^2 \,, \tag{7}$$

with harmonic constants, $\lambda_\mu$, drawn from a probability distribution, $\rho(\lambda_\mu)$. For concreteness, we choose $\rho(\lambda_\mu)$ to be the Wigner semi-circle law,

$$\rho(\lambda_\mu) = \frac{1}{2\pi J^2} \sqrt{(2J)^2 - \lambda_\mu^2} \qquad \text{for} \qquad \lambda_\mu \in [-2J, 2J] \,, \tag{8}$$

and zero otherwise, and we order the $\lambda_\mu$ in such a way that $\lambda_1 < \lambda_2 < \cdots < \lambda_{N-1} < \lambda_N$. The reason for this choice is the connection to the disordered Sherrington-Kirkpatrick model that we will discuss below. There are as many positive as negative $\lambda_\mu$s but the motion is not unstable since the particle is constrained to move (on average) on the sphere. Several useful properties of this distribution are summarised in App. A. Most of our qualitative results are generic, they will be recovered in very similar form for other distributions $\rho$ with finite support and no repeated values of the $\lambda_\mu$.

The potential energy landscape is very simple. Some of its important features are that its minimum is achieved for $s_N = \vec{s} \cdot \vec{v}_N = \pm\sqrt{N}$ and $s_{\mu \neq N} = 0$, and takes the value $V_{\min} = -(\lambda_N/2)N$. The first excited states have one unstable direction, it is given by $s_{N-1} = \pm\sqrt{N}$ and $s_{\mu \neq N-1} = 0$, and have potential energy $V_{1st} = -(\lambda_{N-1}/2)N$. So on and so forth one identifies all metastable points in the potential energy landscape.

With all these elements at our disposal, we define the model through the Hamiltonian,

$$\begin{aligned} H[z(t)] &= \sum_\mu \frac{p_\mu^2}{2m} + \sum_\mu \frac{1}{2}(z(t) - \lambda_\mu)s_\mu^2 - \frac{N}{2}z(t) \\ &\equiv H_{\text{quad}} + \frac{1}{2}z(t)\left( \sum_\mu s_\mu^2 - N \right) \,, \end{aligned} \tag{9}$$

where $z(t)$ is a function of time given by

$$z(t) = \sum_\mu \frac{\langle p_\mu^2 \rangle_{i.c.}}{m} + \sum_\mu \lambda_\mu \langle s_\mu^2 \rangle_{i.c.} \,, \tag{10}$$

and ensures the validity of the spherical constraint, on average over the initial conditions. Here and in the following we only write explicitly the time dependence of the Lagrange multiplier $z$ and not the one of the phase space variables $s_\mu$ and $p_\mu$, which should be assumed. There is no need to add a Lagrange multiplier for the secondary constraint. Once we introduce $z(t)$ the secondary constraint is satisfied automatically, see Section 2.1.

The equations of motion can be written as,

$$\begin{aligned} \dot{s}_\mu &= \{s_\mu, H[z(t)]\} = \frac{p_\mu}{m} \,, \\ \dot{p}_\mu &= \{p_\mu, H[z(t)]\} = -s_\mu\left(z(t) - \lambda_\mu\right) \,, \end{aligned} \tag{11}$$

where $\{\cdots\}$ denotes the Poisson bracket. The equations of motion in Eq. (11) represent a system of harmonic oscillators with time-dependent frequencies

$$\omega_\mu^2(t) = \frac{z(t) - \lambda_\mu}{m}, \tag{12}$$

coupled through the time dependent Lagrange multiplier $z(t)$. Note that the system remains non-linear due to the mode's coupling through $z$.

## 2.1  Conservation laws

Our model does not posses any strictly conserved quantity. However, the dynamics induced by $H[z(t)]$ conserve on average the same phase-space functions that the Neumann Model [25], which we introduce below and discuss in App. B, conserves exactly. To begin with, we show that the definition of $z(t)$ in Eq. (10) can be derived from the imposition of the primary constraint $\phi$ on average. To see this, let us take the second equation of motion in Eq. (11), multiply both sides by $s_\mu$, sum over all $\mu$ and take the average over $\rho_{i.c.}$. We end up with,

$$z(t) \sum_\mu \langle s_\mu^2 \rangle_{i.c.} = -m \sum_\mu \left( \langle \ddot{s}_\mu s_\mu \rangle_{i.c.} + \lambda_\mu \langle s_\mu^2 \rangle_{i.c.} \right). \tag{13}$$

Imposing $\langle \phi \rangle_{i.c.} = \sum_\mu \langle s_\mu^2 \rangle_{i.c.} - N = 0$ implies $\sum_\mu \langle \dot{s}_\mu^2 \rangle_{i.c.} + \langle s_\mu \ddot{s}_\mu \rangle_{i.c.} = 0$, as can be easily seen by differentiating twice. Inserting these conditions in Eq. (13) we find $z(t) = \sum_\mu m \langle \dot{s}_\mu^2 \rangle_{i.c.} + \lambda_\mu \langle s_\mu^2 \rangle_{i.c}$, which is exactly the definition of $z(t)$ given in Eq. (10). In conclusion, $z(t)$ is determined self-consistently to enforce $\langle \phi \rangle_{i.c.} = 0$.

The time variation of the secondary constraint is

$$\frac{d\phi'}{dt} = \left\{ \sum_\mu s_\mu p_\mu, H[z(t)] \right\} = \sum_\mu \left[ \frac{p_\mu^2}{m} - (z(t) - \lambda_\mu) s_\mu^2 \right], \tag{14}$$

and it is not conserved on a trajectory basis. However, taking the average over $\rho_{i.c.}$ and using the definition of $z(t)$ we find that $d_t \langle \phi' \rangle_{i.c.} = 0$, i.e., it is conserved on average. Moreover, if one takes $\langle \phi' \rangle_{i.c.} = 0$ at $t = 0$, we conclude that the average is zero at all times.

Let us now study the phase space functions

$$I_\mu = s_\mu^2 + \frac{1}{mN} \sum_{\nu(\neq\mu)} \frac{s_\mu^2 p_\nu^2 + p_\mu^2 s_\nu^2 - 2 s_\mu p_\mu s_\nu p_\nu}{\lambda_\nu - \lambda_\mu}, \tag{15}$$

which, we will recall later, where found to be strictly conserved in the Neumann model [46,47]. We find that,

$$\frac{m}{2} \frac{dI_\mu}{dt} = \{I_\mu, H[z(t)]\} = s_\mu p_\mu + s_\mu^2 \left( \frac{1}{N} \sum_{\nu(\neq\mu)} s_\nu p_\nu \right) - s_\mu p_\mu \left( \frac{1}{N} \sum_{\nu(\neq\mu)} s_\nu^2 \right). \tag{16}$$

It is clear from here that these functions are not conserved on each trajectory by our model. However, if we take initial conditions such that

$$\langle s_\mu^2 s_\nu p_\nu \rangle_{i.c.} = \langle s_\mu^2 \rangle_{i.c.} \langle s_\nu p_\nu \rangle_{i.c.}, \tag{17}$$

for $\mu \neq \nu$, as fulfilled, for example, if $\rho_{i.c.}$ is Gaussian in $s_\mu$ and $p_\mu$ without correlations between different modes (a case that we will analyse in detail in the rest of the paper) then

$$\frac{m}{2} \frac{d\langle I_\mu \rangle_{i.c.}}{dt} = \langle s_\mu p_\mu \rangle_{i.c.} + \langle s_\mu^2 \rangle_{i.c.} \frac{1}{N} \sum_{\nu(\neq\mu)} \langle s_\nu p_\nu \rangle_{i.c.} - \langle s_\mu p_\mu \rangle_{i.c.} \frac{1}{N} \sum_{\nu(\neq\mu)} \langle s_\nu^2 \rangle_{i.c.}. \tag{18}$$

Rearranging terms and using $\sum_\mu \langle s_\mu^2 \rangle_{i.c.} = N$ and $\sum_\mu \langle s_\mu p_\mu \rangle_{i.c.} = 0$,

$$\frac{d\langle I_\mu \rangle_{i.c.}}{dt} = 0\,, \tag{19}$$

and all $I_\mu$s are conserved on average over the initial conditions.

Finally, we analyse the conservation on average of $2H_{\text{quad}} = \sum_\mu \left( p_\mu^2/m - \lambda_\mu s_\mu^2 \right)$,

$$\frac{dH_{\text{quad}}}{dt} = \left\{ H_{\text{quad}}, H[z(t)] \right\} = -\frac{z(t)}{m} \sum_\mu s_\mu p_\mu\,. \tag{20}$$

Taking the average with respect to $\rho_{i.c.}$ and using $\langle \phi' \rangle_{i.c.} = 0$, we deduce that $H_{\text{quad}}$ is also conserved on average. This is not surprising since

$$H_{\text{quad}} = -\frac{1}{2} \sum_\mu \lambda_\mu I_\mu\,, \tag{21}$$

under the constraints.

## 2.2 Long times and large $N$ limits

In Newtonian form the equations of motion are

$$m\ddot{s}_\mu + (z(t) - \lambda_\mu)s_\mu = 0\,. \tag{22}$$

The dynamics reduce to the ones of a set of uncoupled harmonic oscillators only if $z(t)$ reaches a long-times limit with a strictly constant value. This is only possible in the large $N$ and long times limit taken in the precise order

$$\lim_{t \to \infty} \lim_{N \to \infty} z(t) \equiv z_f\,. \tag{23}$$

Indeed, in order to have a well defined long time limit, we need $N \to \infty$, given that for finite $N$ there will always be oscillations of $z(t)$ around its average.

In [10, 11] we derived Schwinger-Dyson equations in the $N \to \infty$ limit which allowed us to study the long-time evolution in the thermodynamic limit. This approach does not yield information about the behaviour of the $\mu$ modes independently. If we want to treat them one by one, we are forced to keep $N$ finite. Another formalism [87], mixing analytic and numeric methods, allowed us to reach long but also finite times in systems with finite size [10,11]. This point will be very important for the analysis of the numerical solution presented in Sec. 6.

## 2.3 The Neumann model

The celebrated Neumann Model (NM) describes the dynamics of a particle *strictly* constrained to move on the $N-1$ dimensional sphere, under the effect of fixed Hookean forces [25]. It can be formulated in two ways which we summarise in App. B.

The model we have just introduced reproduces *on average* the main characteristics of the Neumann model. We expect them to be completely equivalent in the $N \to \infty$ limit provided that the relative fluctuations of the constraints vanish. To draw an analogy with the ensembles of statistical mechanics, our model is the "grand canonical" version of the Neumann model in which the spherical constraint, playing the role of the "number of particles", is conserved only on average. We can therefore expect equivalence between both representations if the fluctuations of the quantity to be constrained vanish. We will study this point in detail in Sec. 7, adapting ideas in [72–74] to the problem at hand.

## 2.4 The spherical Sherrington-Kirkpatrick model

The choice of a quadratic potential with harmonic constants distributed with the semi-circle Wigner law, see Eq. (7), is motivated by a very well-known model of a disordered system, the spherical Sherrington-Kirkpatrick or $p = 2$ model. In App. C we recall the canonical equilibrium properties of this model when in contact with a thermal bath at temperature $T_0$. We use this probability distribution to draw the initial conditions for the dynamic evolution of our particle, with the physical motivation given in Sec. 3.

# 3 Initial conditions and quench protocol

To specify the dynamics of the system we need to choose the initial state and the Hamiltonian that drives the evolution. We address now how we implement these choices. We discuss the initial measures used: a Gaussian centered at zero with finite or diverging dispersion (Sec. 3.1.1), and a mixed two pure-state measure with the possibility of symmetry breaking induced by a vanishing pinning field (Sec. 3.1.2) [72–74]. We briefly describe the quench protocol and the energy injection or extraction it induces in Sec. 3.2. We evaluate the Uhlenbeck constants of motion in Sec. 3.3. Finally, in Sec. 3.4 we recall results found in [10] using the Schwinger-Dyson equations.

## 3.1 Initial conditions

Concerning the choice of initial conditions, we will be guided by the Boltzmann equilibrium statistical properties of the model with potential energy (7), which we call $H^{(0)}$ when the harmonic constants are $\lambda_\mu^{(0)}$. We distinguish three cases:

- The direction $\vec{v}_N$ with the largest harmonic constant $\lambda_N^{(0)}$ plays no special role and the position of the particle has no macroscopic projection on it, $s_N(t = 0) \equiv \vec{s}(t = 0) \cdot \vec{v}_N = \mathcal{O}(1)$. The fluctuations around this value are order 1. Their average vanishes and the variance is $\mathcal{O}(1)$. We call these initial configurations *extended*.

- The position of the particle is mostly aligned with this direction and $s_N(t = 0) \equiv \vec{s}(t = 0) \cdot \vec{v}_N = \mathcal{O}(N^{1/2})$. The configurations are therefore very close to the minimal energy one. We call these initial configurations *condensed with symmetry broken*.

- The position of the particle is not macroscopically aligned with the direction $\vec{v}_N$ but there are large fluctuations in the ensemble of initial conditions in such a way that $\langle s_N^2(t = 0) \rangle_{i.c.} = \mathcal{O}(N)$. We say that there is *condensation of fluctuations* in this case.

These configurations correspond to the conventional thermal equilibrium of the model defined by $H^{(0)}$ at temperatures $T_0 \geq J_0$, in the first case, or $T_0 \leq J_0$ in the second and third cases. The initial conditions are sketched in Fig. 1, more details are given below and in App. C. We then switch off any connection to the environment, change the Hamiltonian to $H$ and let the system evolve in isolation in the way described in Sec. 3.2.

### 3.1.1 Symmetric distribution

Finite $N$ initial conditions drawn from

$$\rho_{i.c.}(\{p_\mu, s_\mu\}) = \frac{1}{Z_{\text{eq}}} e^{-\frac{1}{T_0} \sum_\mu \frac{p_\mu^2}{2m} - \frac{1}{T_0} \sum_\mu \frac{(z_{\text{eq}}^{(N)} - \lambda_\mu^{(0)})}{2} s_\mu^2}, \tag{24}$$

with $Z_{\text{eq}}$ the partition function, preserve the symmetry $s_N \leftrightarrow -s_N$. The $\lambda_\mu^{(0)}$ follow the Wigner law, and each element has variance $J_0^2/N$. This distribution is equivalent to the equilibrium measure of the $p = 2$ spherical spin model at temperature $T_0$, see App. C, with the aggregate of a Maxwell distribution for the momenta. The Lagrange multiplier $z_{\text{eq}}^{(N)}$ imposes the primary constraint on average, while the average of the secondary constraint is satisfied automatically since $\langle s_\mu p_\mu \rangle_{i.c.} = 0$, $\forall \mu$. The upper-script $(N)$ indicates that $z_{\text{eq}}$ depends on the system size.

The relevant statistical averages are

$$\langle p_\mu^2 \rangle_{i.c.} = m T_0 \qquad \forall \mu, N, \tag{25}$$

$$\langle s_\mu^2 \rangle_{i.c.} = \frac{T_0}{z_{\text{eq}}^{(N)} - \lambda_\mu^{(0)}} \qquad \forall \mu, N, \tag{26}$$

where the Lagrange multiplier $z_{\text{eq}}^{(N)}$ can be obtained, numerically, as the solution of the spherical constraint equation

$$\sum_{\mu=1}^N \langle s_\mu^2 \rangle_{i.c.} = \sum_{\mu=1}^N \frac{T_0}{z_{\text{eq}}^{(N)} - \lambda_\mu^{(0)}} = N. \tag{27}$$

In a previous work (see Sec. 5.5 in Ref. [10]), we have checked that the solution $z_{\text{eq}}^{(N)}$ of Eq. (27) has correct large $N$ limits in both the high and low temperature phases. Moreover, we find that $z_{\text{eq}}^{(N)} > \lambda_N$ for any finite $N$, which ensures that the averages $\langle s_\mu^2 \rangle_{i.c.}$ are well defined.

Note that for this set of initial conditions the $s_N$ mode always has a vanishing average $\langle s_N \rangle_{i.c.} = 0$. It is its variance $\langle s_N^2 \rangle_{i.c.}$, instead, that diverges with $N$ in the *condensed* phase while it is $\mathcal{O}(1)$ in the *extended* one.

Once the finite size Lagrange multiplier is obtained, we replace it in Eq. (26) to obtain $\langle s_\mu^2(0^+) \rangle_{i.c.} = \langle s_\mu^2 \rangle_{\text{eq}}$, which together with $\langle p_\mu^2(0^+) \rangle_{i.c.} = \langle p_\mu^2 \rangle_{\text{eq}}$ and $\langle p_\mu(0^+) s_\mu(0^+) \rangle_{i.c.} = 0$, are a set of initial condition for the mode dynamics.

Some initial spin configurations of this kind are shown with red arrows in Fig. 1, in the middle and right graphs. The symmetry property is illustrated by the fact that $\vec{m}$, the average of $\vec{s}$ shown with a (green) dot, vanishes. The two concentrical spheres in the middle graph represent the fluctuations of the spherical constraint, see Sec. 7 and Refs. [72–74].

### 3.1.2 Symmetry broken configurations

In the condensed phase we can envision another kind of initial conditions, such that the last mode acquires a large average but with an $\mathcal{O}(1)$ variance,

$$\rho_{i.c.}(\{p_\mu, s_\mu\}) = \frac{1}{Z_{\text{eq}}} e^{-\frac{1}{T_0} \sum_\mu \frac{p_\mu^2}{2m} - \frac{1}{T_0} \sum_{\mu=1}^{N-1} \frac{(z_{\text{eq}}^{(N)} - \lambda_\mu^{(0)})}{2} s_\mu^2 - \frac{1}{T_0} \frac{(s_N - \bar{s}_N)^2}{2\sigma_N^2}}, \tag{28}$$

where $\sigma_N$ is an $\mathcal{O}(1)$ number. In order to fix $\langle s_N \rangle_{i.c.} = \bar{s}_N$ we use the spherical constraint,

$$N = \sum_\mu \langle s_\mu^2 \rangle_{i.c.} = \sum_{\mu=1}^{N-1} \frac{T_0}{z_{\text{eq}}^{(N)} - \lambda_\mu^{(0)}} + \bar{s}_N^2 + \sigma_N^2. \tag{29}$$

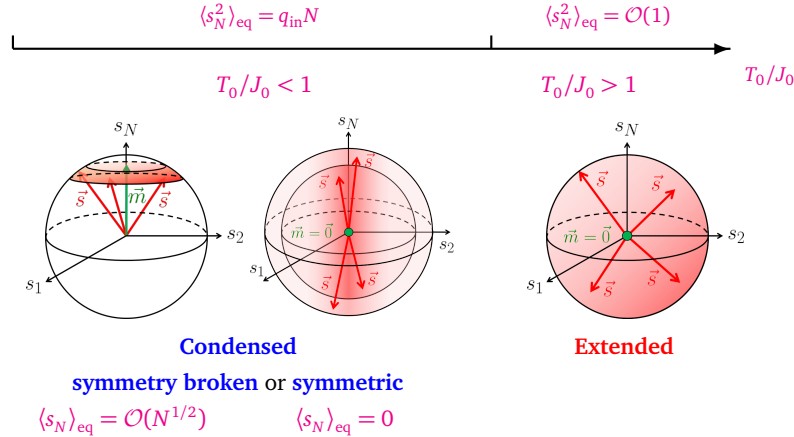

Figure 1: **The three kinds of initial conditions.** Symmetry broken and symmetric for $T_0/J_0 < 1$, which we call condensed because of $\langle s_N^2 \rangle_{\text{eq}} = q_{\text{in}}N$, and extended with $\langle s_N^2 \rangle_{\text{eq}} = \mathcal{O}(1)$ for $T_0/J_0 \geq 1$. The (green) vector $\vec{m}$ represents the symmetry broken direction, while the (red) arrows $\vec{s}$ indicate different realisations of the initial conditions.

In the large $N$ limit, $z_{\text{eq}}^{(N)} \to \lambda_{\max}^{(0)} = 2J_0$ (see App. C) and we can drop the sub-leading contribution $\sigma_N^2$. Thus,

$$\lim_{N \to \infty} \bar{s}_N^2 = N\left(1 - \frac{T_0}{J_0}\right) \equiv Nq_{\text{in}}, \tag{30}$$

and the last mode can assume two values $\bar{s}_N = \pm\sqrt{Nq_{\text{in}}}$, breaking the $s_\mu \to -s_\mu$ symmetry of the Hamiltonian. The symmetry breaking can also be generated with a small "ordering field" in the direction of the $N$-th mode, see Ref. [28].

In the large $N$ limit the symmetric and symmetry broken initial conditions give the same quadratic averages $\langle p_\mu^2 \rangle_{i.c.}$ and $\langle s_\mu^2 \rangle_{i.c.}$. Since in the numerical simulations we focus only on quadratic averages, the results for them will be the same no matter which initial conditions we take. For simplicity we choose to use the symmetric initial conditions in the numerics. The difference between both types of initial conditions becomes important, though, when we consider higher order averages such as $\langle s_\mu^4 \rangle_{i.c.}$ related to the fluctuations of the quadratic quantities. This issue will help us understanding the dynamics for some ranges of parameters and it will be relevant when we discuss the equivalency with the Neumann Model in Sec. 7.

## 3.2 Quench protocol

We perform an instantaneous quench so that at time $t = 0^+$ the averages

$$\langle p_\mu^2(0^+)\rangle_{i.c.} = mT_0, \qquad \langle s_\mu^2(0^+)\rangle_{i.c.} = \frac{T_0}{z_{\text{eq}}^{(0)} - \lambda_\mu^{(0)}}, \tag{31}$$

where $z_{\text{eq}}^{(0)}$ is the Lagrange multiplier fixed with interaction strength $J_0$, and the condensation or not of the last mode, are not altered. The further evolution is driven by the Hamiltonian

$$H[z(t)] = \sum_\mu \frac{p_\mu^2}{2m} + \sum_\mu \frac{1}{2}(z(t) - \lambda_\mu)s_\mu^2 \qquad \text{with} \qquad \lambda_\mu = \frac{J}{J_0}\lambda_\mu^{(0)}. \tag{32}$$

For $J = J_0$ the system remains in the initial thermal equilibrium.

For each initial condition, the mode energy variation is

$$\Delta e_\mu \equiv e_\mu(0^+) - e_\mu(0^-) = e_\mu^{\text{kin}}(0^+) + e_\mu^{\text{pot}}(0^+) - e_\mu^{\text{kin}}(0^-) - e_\mu^{\text{pot}}(0^-), \tag{33}$$

with the kinetic and potential energies

$$e_\mu^{\text{kin}} = \frac{1}{2m} p_\mu^2, \qquad e_\mu^{\text{pot}} = -\frac{1}{2}\lambda_\mu s_\mu^2. \tag{34}$$

Since the quench is instantaneous for each initial state, $p_\mu^2(0^+) = p_\mu^2(0^-)$, there is no variation of the modes' kinetic energy. Moreover, $s_\mu^2(0^+) = s_\mu^2(0^-)$ for all $\mu$ implies

$$\Delta e_\mu = -\frac{1}{2}(\lambda_\mu - \lambda_\mu^{(0)}) s_\mu^2(0^-) = -\frac{1}{2}\left(\frac{J}{J_0} - 1\right)\lambda_\mu^{(0)} s_\mu^2(0^-) = \left(\frac{J}{J_0} - 1\right) e_\mu^{\text{pot}}(0^-). \tag{35}$$

For positive $\lambda_\mu^{(0)}$ and $J/J_0 > 1$, or for negative $\lambda_\mu^{(0)}$ and $J/J_0 < 1$, there is potential, and hence total, energy extraction from the $\mu$th mode. Instead, for positive $\lambda_\mu^{(0)}$ and $J/J_0 < 1$ or for negative $\lambda_\mu^{(0)}$ and $J/J_0 > 1$ there is potential, and also total, energy injection in the $\mu$th mode.

Figure 2 summarises the averaged mode energy variation at the quench. We note that the modes close to the edge are softer than the rest. In particular, the $N$th one is the softest and its energy the most altered by the quench. In the figure we refer to the phases that we will find in the dynamic phase diagram, phase I for $T_0 > J_0$ and $J < J_0$, phase II for $T_0 > J_0$ and $J > J_0$, phase III for $T_0 < J_0$ and $J > J_0$, and phase IV for $T_0 < J_0$ and $J < J_0$ [12].

Concerning the total energy density variation at the quench, one needs to sum over all modes the expression above

$$\Delta e = \frac{1}{N}\sum_\mu \Delta e_\mu = -\frac{1}{2}\left(\frac{J}{J_0} - 1\right)\frac{1}{N}\sum_\mu \lambda_\mu^{(0)} s_\mu^2(0^-) = \left(\frac{J}{J_0} - 1\right) e^{\text{pot}}(0^-), \tag{36}$$

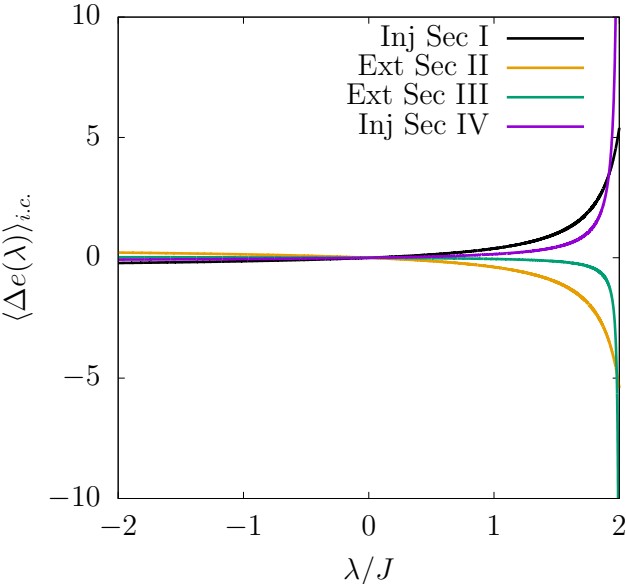

Figure 2: **Averaged mode energy variation at the instantaneous quench in the $N \to \infty$ limit.** The parameters are $T_0 = 1.5 > J_0$, $J = 0.4 < J_0$ in phase I (injection), $T_0 = 1.5 > J_0$, $J = 1.6 > J_0$ in phase II (extraction), $T_0 = 0.5 < J_0$, $J = 1.2 > J_0$ in phase III (extraction), and $T_0 = 0.5 < J_0$, $J = 0.4, J_0$ in phase IV (injection). In phases I and II $\langle \Delta e_N \rangle_{i.c.}$ is finite while in phases III (green) and IV (violet) it is proportional to $N$.

and the sign is fully determined by the prefactor since $e^{\text{pot}}(0^-)$ is negative in all cases. Whereas the quench is found to inject an extensive amount of energy if $J/J_0 < 1$, it extracts an extensive amount of energy if $J/J_0 > 1$ [10].

In the course of the evolution the mode total energies will reshuffle until, in the $N \to \infty$ limit, a stationary limit is reached in which, we will see, they remain constant.

Since the system is isolated, the total energy density in the asymptotic state, $e_f$, is the same as the one at $t = 0^+$, right after the quench, $e_f = \langle e(0^+)\rangle_{i.c.}$. The averaged kinetic energy is $\langle e_{\text{kin}}(0^+)\rangle_{i.c.} = T_0/2$ in all sectors, while the averaged potential energy is equal to $\langle e_{\text{pot}}(0^+)\rangle_{i.c.} = -JJ_0/(2T_0)$ for extended initial conditions (phases I and II) and $\langle e_{\text{pot}}(0^+)\rangle_{i.c.} = -JJ_0/(2T_0)[1 - (1 - T_0/J_0)^2]$ for condensed initial conditions (phases III and IV). The averaged kinetic and potential energies are not constant in time but the total energy is. In the asymptotic stationary limit one can use the total energy conservation and $\overline{z(t)} = 2(\overline{\langle e_{\text{kin}}(t)\rangle_{i.c.}} - \overline{\langle e_{\text{pot}}(t)\rangle_{i.c.}})$, to derive the kinetic energy parameter dependencies given in the last column of Table 1.

### 3.3 The Uhlenbeck integrals

In App. D we prove that, for the initial conditions and the Wigner density of $\lambda_\mu$ that we choose, the averaged constants of motion are given by

$$\langle I(\lambda)\rangle_{i.c.} = k_1 \frac{b - \lambda}{a - \lambda}, \qquad \text{for } \begin{cases} J_0 \geq T_0 \text{ and all } \lambda, \\ J_0 < T_0 \text{ and all } \lambda \neq \lambda_N. \end{cases} \tag{37}$$

The parameters $k_1$, $a$ and $b$ depend on $J_0, T_0, J$:

$$k_1 = \frac{T_0^2}{J_0 J} \qquad \text{and} \qquad b = \frac{J(J + J_0)}{T_0} \qquad \text{in all cases}, \tag{38}$$

while

$$a = \begin{cases} \dfrac{J}{T_0}\left(J_0 + \dfrac{T_0^2}{J_0}\right) \geq 2J & T_0 \geq J_0, \\ 2J & T_0 \leq J_0. \end{cases} \tag{39}$$

For $T_0 < J_0$ one has to single out the $N$th constant of motion:

$$\langle I_N(0^+)\rangle_{i.c.} = \left(1 - \frac{T_0}{J_0}\right)\left(1 - \frac{T_0}{J}\right)N + o(N). \tag{40}$$

In the last equation, the prefactor vanishes at $T_0 = J_0$ and $I_N$ should become $\mathcal{O}(1)$ and equal to the value in Eq. (37). One can easily verify that $\int d\lambda \, \rho(\lambda)\langle I(\lambda)\rangle_{i.c.} + \langle I_N(0^+)\rangle_{i.c.}/N = 1$.

Some relations we will use later are

$$\begin{aligned} a \pm 2J &= \frac{J}{J_0 T_0}(T_0 \pm J_0)^2, \\ a - b &= \frac{J}{J_0 T_0}(T_0^2 - J_0 J), \end{aligned} \qquad T_0 \geq J_0. \tag{41}$$

The constants $\langle I_\mu\rangle_{i.c.}$ define the allowed sub-space in phase space. We next discuss several aspects of them which will help us understand the dynamic behaviour of the particle.

The spectrum of constants of motion is shown, for several representative sets of parameters, in Fig. 3 with high temperature (a) and low temperature (b) initial conditions, respectively. We should note that in (a), though there is a strong variation close to the right edge, none of the curves diverges. Instead, in (b) the $N$th constant of motion is proportional to $N$.

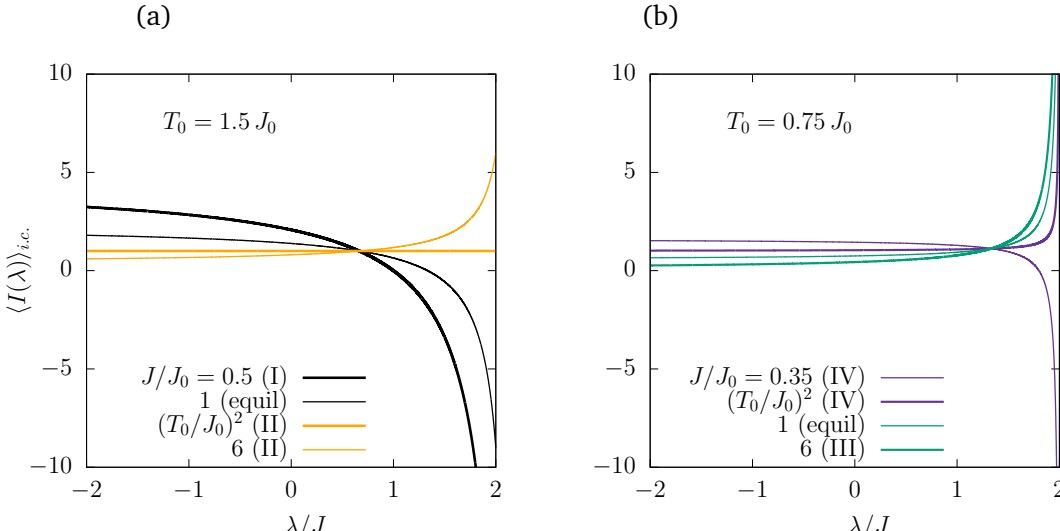

**Figure 3: The spectrum of constants of motion averaged over the initial conditions.** Extended (a) and condensed (b) initial conditions are plotted separately in the two panels for parameters given in the two keys.

One can readily check that for $T_0 > J_0$ and $T_0/J > 1$, $\langle I_N(0^+)\rangle_{i.c.} < 0$. This is what we will call phase I and is represented with a white background in Fig. 4 (a). Also for extended initial conditions, $T_0 > J_0$, within what we call phase II, $T_0/J < 1$, the sign of the largest mode averaged constant is decided by

$$\langle I_N(0^+)\rangle_{i.c.} = \begin{cases} > 0 & T_0 < (J_0 + J)/2 & y < (1+x)/2, \\ < 0 & T_0 > (J_0 + J)/2 & y > (1+x)/2, \end{cases} \quad \text{phase II}. \qquad (42)$$

This defines a straight line separating a region IIa (white background) from a region IIb (dashed blue background) both in phase II. Below this line, the sign is positive while above it, it is negative.

Another important fact is that for condensed initial conditions $T_0 < J_0$, and the first factor in $\langle I_N \rangle_{i.c.}$ is positive definite. Instead, the second factor changes sign at the transition between phase III (dashed blue background) and IV (white background):

$$\langle I_N(0^+)\rangle_{i.c.} = \begin{cases} > 0 & T_0 < J & \text{phase III}, \\ < 0 & T_0 > J & \text{phase IV}. \end{cases} \qquad (43)$$

For $T_0/J_0 < 1$, on the straight line $y = (1+x)/2$, the bulk integrals are all equal, with the $N$th one distinguishing from the rest and ensuring the validity of the sum rule $\sum_\mu \langle I_\mu \rangle_{i.c.} = N$. For $T_0/J_0 > 1$, all integrals of motion equal one on the curve $y^2 = x$ as can be observed in Fig. 3 (a). We will show in Sec. 5 that an exact solution with Gibbs-Boltzmann equilibrium properties is found for such parameters. On the continuation of this curve below $T_0/J_0 = 1$ the integrals of motion differ from each other. Still, we have also found a particularly simple exact solution for these parameters though not one of conventional equilibrium. The full curve $y = x^2$ is shown in violet in Fig. 4 (a).

### 3.4 Schwinger-Dyson equations in the $N \to \infty$ limit

In this Section we briefly summarise the phase diagram derived in the $N \to \infty$ limit from the analysis of the Schwinger-Dyson equations. See Ref. [10] for more details on these methods and Fig. 4 (b) for a recap.



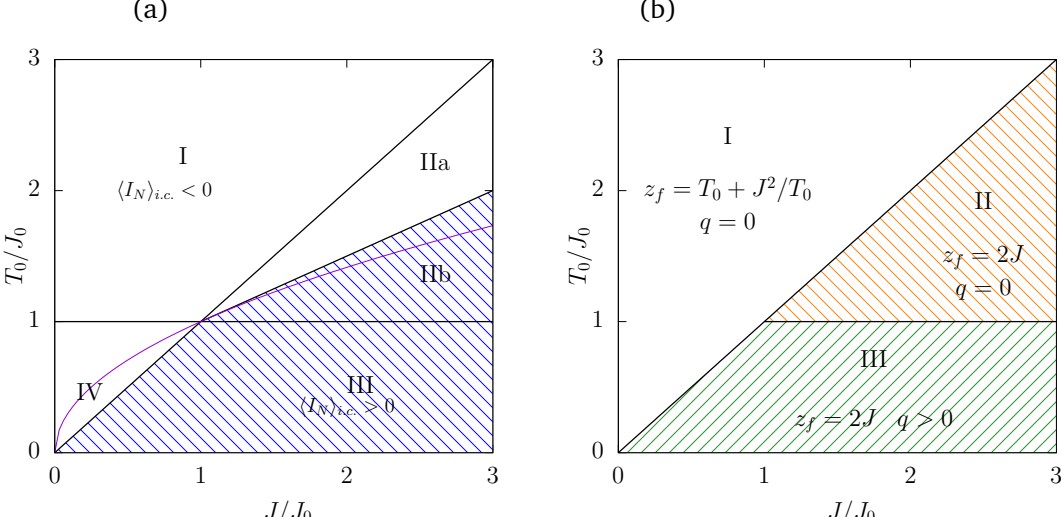

Figure 4: **Properties in parameter space.** (a) Sign of the last conserved quantity, $\langle I_N \rangle_{i.c.}$, in different sectors of the $\{T_0/J_0, J/J_0\}$ plane. Below the horizontal line $T_0 = J_0$, $\langle I_N \rangle_{i.c.} = \mathcal{O}(N)$. In the full white region $\langle I_N \rangle_{i.c.} < 0$ while in the blue dashed part $\langle I_N \rangle_{i.c.} > 0$. The violet solid line represents $T_0/J_0 = (J/J_0)^{1/2}$. $\langle I_\mu \rangle_{i.c.} = 1$ for all $\mu$ on this curve in phase II but they keep a $\mu$ dependence on the curve's continuation in phase IV. (b) The dynamical phase diagram obtained in [10] from the study of the asymptotic limit of the Schwinger-Dyson equations coupling correlation and linear response in the strict thermodynamic limit. The labels in the three phases display the values of the asymptotic Lagrange multiplier, $z_f$, and the limit of the self time-delayed correlation function $q$. The static susceptibility equals $\chi_{st} = 1/T_0$ to the left of the diagonal and $\chi_{st} = 1/J$ to the right of it. $q_0 \equiv \lim_{t \to \infty} C(t, 0) \neq 0$ in III and vanishes elsewhere. From the analysis of the scaling of $\langle s_N^2 \rangle$ and $\langle p_N^2 \rangle$ we were able to identify phase IV, see Section 5 and Fig. 5.

As common in the treatment of fully connected disordered models, one can derive closed Schwinger-Dyson equations, which couple the time-delayed disorder averaged self-correlation and linear response, in the strict $N \to \infty$ limit. These equations include the influence of the initial conditions as special terms that know about the distribution with which those have been drawn. The observables are then computed under the procedure

$$\lim_{N \to \infty} [\langle \dots \rangle_{i.c.}]_\lambda, \tag{44}$$

where $[\dots]_\lambda$ denotes an average over quenched randomness and, eventually, times are taken to diverge only after the thermodynamic limit. A detailed mixed analytical and numerical study of these equations leads to the phase diagram in Fig. 4 (b) [10]. Basically, three phases were identified, distinguished by different values of the asymptotic Lagrange multiplier, $z_f \equiv \lim_{t \to \infty} z(t)$, the susceptibility, $\chi_{st} = \tilde{R}(\omega = 0)$, the limit of the self-correlation $q = \lim_{t \to \infty} \lim_{t_w \to \infty} C(t, t_w)$ and the one of the correlation with the initial condition $q_0 = \lim_{t \to \infty} C(t, 0)$. The correlation is the position-position one, $C(t, t_w) = N^{-1} \sum_\mu [\langle s_\mu(t) s_\mu(t_w) \rangle_{i.c.}]_\lambda$. All these values, in these three phases and a new one that we recognise here (see phase IV in Section 5 and Fig. 5), are given in Table 1. The control parameter dependence of the asymptotic time-averaged kinetic energy density can also be used to distinguish the phases and are given in the last column in the same Table.

Table 1: **Control parameters and relevant measurements**: the asymptotic Lagrange multiplier $z_f$, the static susceptibility $\chi_{\rm st}$, the limit of the self-correlation $q = \lim_{t\to\infty} \lim_{t_w\to\infty} C(t, t_w)$, the asymptotic correlation with the initial condition $q_0 = \lim_{t\to\infty} C(t, 0)$, and the kinetic energy density $e^f_{\rm kin} = \overline{\langle e_{\rm kin}\rangle}_{i.c.}$, in the four phases of the phase diagram. We recall that $e_f = 2e^f_{\rm kin} - z_f/2$.

| Phase | Parameters | $z_f$ | $\chi_{\rm st}$ | $q$ | $q_0$ | $4e^f_{\rm kin}$ |
|-------|-----------|-------|-----------------|-----|-------|------------------|
| I | $T_0 > J$ | $T_0 + \dfrac{J^2}{T_0}$ | $\dfrac{1}{T_0}$ | $0$ | $0$ | $-\dfrac{J_0 J}{T_0} + 2T_0 + \dfrac{J^2}{T_0}$ |
| II | $J > T_0 > J_0$ | $2J$ | $\dfrac{1}{J}$ | $0$ | $0$ | $-\dfrac{J_0 J}{T_0} + T_0 + 2J$ |
| III | $J > T_0 \,\&\, J_0 > T_0$ | $2J$ | $\dfrac{1}{J}$ | $> 0$ | $> 0$ | $T_0\left(1 + \dfrac{J}{J_0}\right)$ |
| IV | $J < T_0 < J_0$ | $T_0 + \dfrac{J^2}{T_0}$ | $\dfrac{1}{T_0}$ | $0$ | $0$ | $\dfrac{J T_0}{J_0} - 2J + 2T_0 + \dfrac{J^2}{T_0}$ |

## 4 The Generalised Gibbs Ensemble

Local observables in the asymptotic stationary limit of the evolution of integrable systems are expected to be equal to their averages over the Generalised Gibbs Ensemble (GGE) measure. They should therefore be derived from variations of the GGE partition function

$$Z_{\rm GGE} = \int d\vec{s}\, d\vec{p}\, d\bar{z}\, d\bar{z}'\, \exp\left[-\sum_\mu \gamma_\mu I_\mu - \frac{\bar{z}}{2}\left(\sum_\mu s_\mu^2 - N\right) + \frac{\bar{z}'}{2}\left(\sum_\mu s_\mu p_\mu\right)\right], \qquad (45)$$

with respect to adequately added sources that we omit to write to lighten the notation. The $\gamma_\mu$ are Lagrange multipliers which should be implicitly fixed by the GGE equations,

$$\langle I_\mu \rangle_{i.c.} = \langle I_\mu \rangle_{\rm GGE}, \qquad (46)$$

where $\langle \cdots \rangle_{\rm GGE}$ denotes an average with respect to the measure in Eq. (45).

In Eq. (45) we propose a "soft" version of the GGE in which the constraints are imposed on average through the Lagrange multipliers $\bar{z}$ and $\bar{z}'$. (We will see that these are related but are not identical to the multipliers used in the dynamic formalism.) The relationship between this formulation of the GGE and the one involving strict constraints will be addressed in Sec. 7. We will explain how to recover the equilibrium Maxwell-Boltzmann distribution from Eq. (45), when no quench is performed, in Sec. 4.6. In the rest of this Section we evaluate the partition function.

### 4.1 The action

As a first step towards the evaluation of $Z_{\rm GGE}$, let us expand the expression of the GGE measure by using the definition of the integrals of motion,

$$Z_{\rm GGE} = \int d\vec{s}\, d\vec{p}\, d\bar{z}\, d\bar{z}'\, \exp\Big[-\sum_{\mu=1}^N \left(\gamma_\mu + \frac{\bar{z}}{2}\right)s_\mu^2 + \frac{\bar{z}'}{2}\sum_\mu s_\mu p_\mu + \frac{\bar{z}N}{2}$$
$$+ \frac{1}{mN}\sum_{\mu\neq\nu} \eta(\mu,\nu)\,(s_\mu^2 p_\nu^2 - s_\mu s_\nu p_\mu p_\nu)\Big]. \qquad (47)$$

At this stage it is important to proceed to an analytic continuation in the last term in Eq. (47). The idea is to complete the sum, adding the $\mu = \nu$ contribution, by introducing a regularised fraction

$$\eta(\mu, \nu) \equiv \left[ \frac{\gamma_\mu - \gamma_\nu}{\lambda_\mu - \lambda_\nu} \right]_R , \tag{48}$$

where $R$ stands for regularised. The only requirement for this function is to be continuous in the thermodynamic limit and to verify $\eta(\mu, \nu) = (\gamma_\mu - \gamma_\nu)/(\lambda_\mu - \lambda_\nu)$ for $\mu \neq \nu$. This continuation is simply a rewriting of the partition function and not a new definition as it can be verified that

$$\sum_{\mu \neq \nu} \frac{\gamma_\mu - \gamma_\nu}{\lambda_\mu - \lambda_\nu} (s_\mu^2 p_\nu^2 - s_\mu s_\nu p_\mu p_\nu) = \sum_{\mu, \nu} \left[ \frac{\gamma_\mu - \gamma_\nu}{\lambda_\mu - \lambda_\nu} \right]_R (s_\mu^2 p_\nu^2 - s_\mu s_\nu p_\mu p_\nu) . \tag{49}$$

In the following we will detail when this change plays a role and which key element it introduces.

Next, we define auxiliary fields,

$$A_\mu^{(s)} = s_\mu^2 , \qquad A_\mu^{(p)} = p_\mu^2 , \qquad A_\mu^{(sp)} = s_\mu p_\mu , \tag{50}$$

and we introduce factors 1

$$1 \propto \int \prod_\mu dA_\mu^{(p)} dl_\mu^{(p)} \exp\left[ \sum_\mu l_\mu^{(p)} \left( A_\mu^{(p)} - p_\mu^2 \right) \right] , \tag{51}$$

where we avoided writing numerical factors. Analogous expressions involving $A_\mu^{(s)}$ and $A_\mu^{(sp)}$ are also introduced. Inserting these identities, and after some simple steps, we find

$$\begin{aligned}
Z_{\text{GGE}} = \int & \, d\vec{s} \, d\vec{p} \, dA_\mu^{(p)} \, dl_\mu^{(p)} \, dA_\mu^{(s)} \, dl_\mu^{(s)} \, dA_\mu^{(sp)} \, dl_\mu^{(sp)} \, d\bar{z} \, d\bar{z}' \times \\
& \exp\left[ -\sum_\mu \gamma_\mu A_\mu^{(s)} + \frac{1}{mN} \sum_{\nu, \mu} \eta(\mu, \nu) \left( A_\mu^{(s)} A_\nu^{(p)} - A_\mu^{(sp)} A_\nu^{(sp)} \right) \right. \\
& \left. - \frac{\bar{z}}{2} \left( \sum_\mu A_\mu^{(s)} - N \right) + \frac{\bar{z}'}{2} \sum_\mu A_\mu^{(sp)} \right. \\
& \left. + \sum_\mu \left( l_\mu^{(s)} A_\mu^{(s)} + l_\mu^{(p)} A_\mu^{(s)} + l_\mu^{(pp)} A_\mu^{(sp)} \right) \right. \\
& \left. - \sum_\mu \left( l_\mu^{(s)} s_\mu^2 + l_\mu^{(p)} p_\mu^2 + l_\mu^{(sp)} s_\mu p_\mu \right) \right] ,
\end{aligned} \tag{52}$$

where, again, we omitted irrelevant numerical factors. With this choice of auxiliary variables the expression in the exponent is automatically organised in a group of terms that depend only on the new variables $A_\mu^{(s,p,sp)}$ and the Lagrange multipliers $l_\mu^{(s,p,sp)}$, and another group of quadratic terms in the original phase-space variables $\{s_\mu, p_\mu\}$. The Gaussian integrals over $\{s_\mu, p_\mu\}$ can then be easily computed,

$$\begin{aligned}
\prod_\mu \int & \, ds_\mu \, dp_\mu \exp\left[ -\frac{1}{2} \begin{pmatrix} s_\mu & p_\mu \end{pmatrix} \begin{pmatrix} 2l_\mu^{(s)} & l_\mu^{(sp)} \\ l_\mu^{(sp)} & 2l_\mu^{(p)} \end{pmatrix} \begin{pmatrix} s_\mu \\ p_\mu \end{pmatrix} \right] \\
& \propto \prod_\mu \left[ 4l_\mu^{(s)} l_\mu^{(p)} - \left( l_\mu^{(sp)} \right)^2 \right]^{-1/2} ,
\end{aligned} \tag{53}$$

and the GGE partition function can be rewritten as

$$Z_{\text{GGE}} \propto \int dA_\mu^{(p)} \, dl_\mu^{(p)} \, dA_\mu^{(s)} \, dl_\mu^{(s)} \, dA_\mu^{(sp)} \, dl_\mu^{(sp)} \, d\bar{z} \, d\bar{z}' \exp(-NS) , \tag{54}$$

with the action $S$ given by

$$S = \frac{1}{N} \sum_\mu \gamma_\mu A_\mu^{(s)} - \frac{1}{mN^2} \sum_{\mu,\nu} \eta(\mu,\nu) \left( A_\mu^{(s)} A_\nu^{(p)} - A_\mu^{(sp)} A_\nu^{(sp)} \right)$$
$$- \frac{1}{N} \sum_\mu \left( l_\mu^{(s)} A_\mu^{(s)} + l_\mu^{(p)} A_\mu^{(p)} + l_\mu^{(sp)} A_\mu^{(sp)} \right) + \frac{1}{2N} \sum_\mu \ln \left[ 4 l_\mu^{(s)} l_\mu^{(p)} - \left( l_\mu^{(sp)} \right)^2 \right]$$
$$+ \frac{\bar{z}}{2N} \left( \sum_\mu A_\mu^{(s)} - N \right) - \frac{\bar{z}'}{2N} \sum_\mu A_\mu^{(sp)}. \tag{55}$$

Up to now the treatment has been exact for any $N$. In order to proceed further we have to make some approximations.

## 4.2 Saddle-point evaluation

In the large $N$ limit the saddle-point values of the $A$'s will be equal to their averages under the GGE measure

$$A_\mu^{(s)}\big|_{\text{S.P.}} = \langle s_\mu^2 \rangle_{\text{GGE}}, \qquad A_\mu^{(p)}\big|_{\text{S.P.}} = \langle p_\mu^2 \rangle_{\text{GGE}}, \qquad A_\mu^{(sp)}\big|_{\text{S.P.}} = \langle s_\mu p_\mu \rangle_{\text{GGE}}, \tag{56}$$

where $\big|_{\text{S.P.}}$ indicates that the quantities are evaluated at the saddle-point. We note that $S$ is an $\mathcal{O}(1)$ object with respect to $N$ and we are making saddle-point evaluations with respect to $N$ quantities. This procedure can be justified by taking a continuum limit in which sums over $\mu$ are replaced by integrals over $\lambda$ with the adequate density. To keep the notation light, we do not make this passage explicit here and we stick to the discrete notation.

The first group of $3N$ saddle-point equations comes from differentiating the action with respect to $l_\mu^{(s,p,sp)}$,

$$A_\mu^{(s)} = \frac{2 l_\mu^{(p)}}{4 l_\mu^{(s)} l_\mu^{(p)} - \left( l_\mu^{(sp)} \right)^2}, \qquad A_\mu^{(p)} = \frac{2 l_\mu^{(s)}}{4 l_\mu^{(s)} l_\mu^{(p)} - \left( l_\mu^{(sp)} \right)^2}, \tag{57}$$

and similarly for $A_\mu^{(sp)}$. The second group of $3N$ saddle-point equations arise from differentiating the action with respect to $A_\mu^{(s,p,sp)}$,

$$l_\mu^{(s)} = \gamma_\mu + \frac{\bar{z}}{2} - \frac{1}{mN} \sum_\nu A_\nu^{(p)} \eta(\mu,\nu), \qquad l_\mu^{(p)} = -\frac{1}{mN} \sum_\nu A_\nu^{(s)} \eta(\mu,\nu),$$
$$l_\mu^{(sp)} = \frac{\bar{z}'}{2} + \frac{2}{mN} \sum_\nu A_\nu^{(sp)} \eta(\mu,\nu). \tag{58}$$

The last two equations represent the constraints and are derived from the differentiation of the action with respect to $\bar{z}$ and $\bar{z}'$,

$$\sum_\mu A_\mu^{(s)} = N, \qquad \sum_\mu A_\mu^{(sp)} = 0. \tag{59}$$

These are $6N+2$ equations for the mean-fields $A_\mu^{(s,p,sp)}$, the Lagrange multipliers $l_\mu^{(s,p,sp)}$, $\bar{z}$ and $\bar{z}'$.

## 4.3 The conserved quantities

The system of saddle-point equations obtained in the previous Subsection should be complemented with the equations that determine the GGE Lagrange multipliers $\gamma_\mu$.

The averages of the conserved quantities over the GGE distribution are

$$\langle I_\mu \rangle_{\text{GGE}} = -\frac{\partial \ln Z_{\text{GGE}}}{\partial \gamma_\mu}. \tag{60}$$

In the saddle-point approximation, at leading order in $N$,

$$\ln Z_{\text{GGE}} = -N S(A_\mu^{(s,p,sp)}, l_\mu^{(s,p,sp)}, \bar{z}, \bar{z}')\Big|_{\text{S.P.}}. \tag{61}$$

From Eq. (55) we easily obtain

$$\langle I_\mu \rangle_{\text{GGE}} = A_\mu^{(s)} + \frac{1}{mN} \sum_{\nu(\neq\mu)} \frac{1}{\lambda_\nu - \lambda_\mu} \Big[ \Big( A_\mu^{(s)} A_\nu^{(p)} - A_\mu^{(sp)} A_\nu^{(sp)} \Big) + (\mu \leftrightarrow \nu) \Big]\Big|_{\text{S.P.}}. \tag{62}$$

The contribution of the regularisation of $(\gamma_\mu - \gamma_\nu)/(\lambda_\mu - \lambda_\nu)$ for $\mu = \nu$ induces sub-leading corrections to the last expression in the large $N$ limit. The additional set of equations for the $\gamma_\mu$'s can be obtained by equating the right-hand-side of Eq. (62) with $\langle I_\mu \rangle_{i.c.}$. Equations (62) together with Eqs. (57)-(59) form a closed set which involve all the unknowns. It has the value of the conserved quantities in the initial state $\langle I_\mu \rangle_{i.c.}$ and the value of $J$ in the evolution, which is specified by the $\lambda_\mu = (J/J_0)\lambda_\mu^{(0)}$, as only inputs.

## 4.4 Simplification of the saddle-point equations

Henceforth we do not write explicitly $|_{\text{S.P.}}$, but we recall that the equations hold at the saddle-point level. We will focus on the manifold of solutions with $A_\mu^{(sp)} = 0$, $\forall \mu$, for which the secondary constraint $\sum_\mu A_\mu^{(sp)} = 0$ is satisfied automatically, and the Lagrange multiplier $\bar{z}'$ can be safely discarded, as well as the $l_\mu^{(sp)}$ which also vanish. We will see in the next Section that this particular manifold of solutions correctly captures the dynamics of the system.

With this prescription the system of saddle-point equations can be simplified to

$$A_\mu^{(p)} = \left( \frac{-2}{mN} \sum_\nu A_\nu^{(s)} \eta(\mu, \nu) \right)^{-1},$$

$$A_\mu^{(s)} = \left( 2\gamma_\mu + \bar{z} - \frac{2}{mN} \sum_\nu A_\nu^{(p)} \eta(\mu, \nu) \right)^{-1}, \tag{63}$$

$$\sum_\mu A_\mu^{(s)} = N,$$

which, together with the GGE equations,

$$\langle I_\mu \rangle_{\text{GGE}} = A_\mu^{(s)} + \frac{1}{mN} \sum_{\nu(\neq\mu)} \frac{1}{\lambda_\nu - \lambda_\mu} \Big( A_\mu^{(s)} A_\nu^{(p)} + A_\nu^{(s)} A_\mu^{(p)} \Big) = \langle I_\mu \rangle_{i.c.}, \tag{64}$$

form a closed system coupling the mean-fields $A_\mu^{(s,p)}$, the Lagrange multiplier enforcing the primary spherical constraint $\bar{z}$, and the GGE Lagrange multipliers $\gamma_\mu$.

These equations are invariant under a simultaneous change $\gamma_\mu \to \gamma_\mu + \frac{c}{2}$ and $\bar{z} \to \bar{z} - c$, where $c$ is an arbitrary number. We could choose $c = \bar{z}$ to formally eliminate $\bar{z}$ from the equations. In other words, we could "absorb" $\bar{z}$ into the Lagrange multipliers $\gamma_\mu$. This is possible

because $\sum_\mu \langle I_\mu \rangle_{\text{GGE}} = \sum_\mu A_\mu^{(s)}$, which implies that fixing the values of the conserved quantities automatically fixes the value of the primary constraint. In short, the Lagrange multiplier $\bar{z}$ is redundant. However, eliminating $\tilde{z}$ would obscure the equilibrium limit discussed in Sec. 4.6 so we keep it.

The analytic continuation leading to $\eta(\mu, \nu)$ plays an important role in scenarii in which the system condenses. As an example let us assume that we have $A_N^{(s)} = \langle s_N^2 \rangle_{\text{GGE}} = \mathcal{O}(N)$ and $A_N^{(p)} = \langle p_N^2 \rangle_{\text{GGE}} = \mathcal{O}(1)$. Then, for the $N$th mode

$$A_N^{(p)} = \left( \frac{-2}{mN} \sum_\nu A_\nu^{(s)} \eta(N, \nu) \right)^{-1} = \left( (\tilde{A}_N^{(p)})^{-1} - \frac{2}{mN} A_N^{(s)} \eta(N, N) \right)^{-1}, \tag{65}$$

$$A_N^{(s)} = \left( 2\gamma_N + \bar{z} - \frac{2}{mN} \sum_\nu A_\nu^{(p)} \eta(N, \nu) \right)^{-1} = \left( (\tilde{A}_N^{(s)})^{-1} - \frac{2}{mN} A_N^{(p)} \eta(N, N) \right)^{-1}, \tag{66}$$

where $\tilde{A}_N^{(p)}$ and $\tilde{A}_N^{(s)}$ are the mean-field solutions without the analytic continuation. In this case the difference between the two saddle-point approximations is extensive as we have

$$A_N^{(p)} = \tilde{A}_N^{(p)} + \mathcal{O}(1), \qquad A_N^{(s)} = \tilde{A}_N^{(s)} + \mathcal{O}(N). \tag{67}$$

It is yet unclear at this stage which set of solutions should be taken as a meaningful mean-field decoupling. We will explain later why the analytic continuation is the correct approach, necessary in cases with condensation.

We can extract a useful identity from Eqs. (63). First, we take the right-hand-sides of the first two equations in (63) to their left-hand-sides and we sum over $\mu$. Then we subtract the resulting equations to find

$$\sum_\mu A_\mu^{(s)} \left( 2\gamma_\mu + \bar{z} \right) = 0. \tag{68}$$

This equation will be useful in the developments of the next Subsections.

## 4.5 Harmonic *Ansatz*

We now propose a simple parametrisation of the solution of Eqs. (63),

$$A_\mu^{(s)} = \langle s_\mu^2 \rangle_{\text{GGE}} = \frac{T_\mu}{z - \lambda_\mu}, \qquad A_\mu^{(p)} = \langle p_\mu^2 \rangle_{\text{GGE}} = m T_\mu, \tag{69}$$

which corresponds to the ensemble average of a system of harmonic oscillators with frequencies $z - \lambda_\mu$ at different temperatures $T_\mu$. This parametrisation reduces the number of unknowns from $\{A_\mu^{(s)}, A_\mu^{(p)}\}$ ($2N$ in number) to $\{T_\mu\}$ and $z$ (only $N+1$). We will show that if the parameters $\{T_\mu\}$ and $z$ are such that the first equality in (63) and Eq. (68) are verified, then the second equality in Eq. (63) follows automatically, i.e., the parametrisation is consistent. The system of equations (63) and (68) can then be reduced to

$$m T_\mu = \left( \frac{-2}{mN} \sum_\nu \frac{T_\nu}{z - \lambda_\nu} \eta(\mu, \nu) \right)^{-1}, \qquad \sum_\mu \frac{T_\mu}{z - \lambda_\mu} (2\gamma_\mu + \bar{z}) = 0. \tag{70}$$

In fact, starting from the first equation above and using the identity

$$\frac{1}{z - \lambda_\nu} \frac{1}{\lambda_\nu - \lambda_\mu} = \frac{1}{z - \lambda_\mu} \left( \frac{1}{z - \lambda_\nu} - \frac{1}{\lambda_\mu - \lambda_\nu} \right), \tag{71}$$

we end up with

$$\frac{T_\mu}{z - \lambda_\mu} = \frac{N}{2}\left(\gamma_\mu \sum_\nu \frac{T_\nu}{z - \lambda_\nu} - \sum_\nu \frac{T_\nu}{z - \lambda_\nu}\gamma_\nu - \sum_\nu T_\nu \eta(\mu, \nu)\right)^{-1}. \tag{72}$$

Next we use the second equation in (70) in the last equality to finally arrive at

$$\frac{T_\mu}{z - \lambda_\mu} = \left(2\gamma_\mu + \bar{z} - \frac{2}{N}\sum_\nu T_\nu \eta(\mu, \nu)\right)^{-1}, \tag{73}$$

which is nothing but the second equation in (63) written under the harmonic *Ansatz*.

The mode temperatures $T_\mu$ are fixed by $\langle I_\mu \rangle_{i.c.} = \langle I_\mu \rangle_{\text{GGE}}$, which now reads

$$\langle I_\mu \rangle_{i.c.} = \langle I_\mu \rangle_{\text{GGE}} = \frac{T_\mu}{z - \lambda_\mu} + \frac{T_\mu}{N}\sum_{\nu(\neq\mu)}\frac{T_\nu}{\lambda_\nu - \lambda_\mu}\left(\frac{1}{z - \lambda_\mu} + \frac{1}{z - \lambda_\nu}\right), \tag{74}$$

and is independent of the $\gamma_\mu$. Thanks to the steps detailed in App. **??**, we rewrite this equation as

$$\langle I_\mu \rangle_{i.c.} = \langle I_\mu \rangle_{\text{GGE}} = \frac{2T_\mu}{z - \lambda_\mu}\left(1 + \frac{1}{N}\sum_{\nu(\neq\mu)}\frac{T_\nu}{\lambda_\nu - \lambda_\mu} - \frac{\langle s_N^2 \rangle_{\text{GGE}}}{2N}\delta_{\mu N}\right). \tag{75}$$

The last term gives a non-vanishing contribution only for $\mu = N$ and in the case of condensed initial conditions.

The parameter $z$ should be found by imposing $\sum_\mu \langle I_\mu \rangle_{\text{GGE}} = N$. It is important to recall that $z$ from the harmonic *Ansatz* is not necessarily related to the parameter $\bar{z}$ appearing in the GGE measure, which could be absorbed in the $\gamma_\mu$'s by the shift $\gamma_\mu \mapsto \gamma_\mu - \bar{z}/2$.

Equations (74) determine the mode temperatures $T_\mu$. The first equation in (70) yields the spectrum of Lagrange multipliers $\gamma_\mu$ and the second one determines $z$.

## 4.6 Equilibrium case

The constants of motion satisfy

$$\sum_\mu\left(-\frac{\lambda_\mu}{2}\right)I_\mu = \frac{1}{4mN}\sum_{\mu\neq\mu}(s_\mu p_\nu - p_\mu s_\nu)^2 - \sum_\mu \frac{\lambda_\mu}{2}s_\mu^2, \tag{76}$$

see Eq. (B.15). The right-hand-side reduces to $H_{\text{quad}}$, defined in Eq. (9), see also Eq. (B.1), provided both constraints, $\phi$ and $\phi'$, are satisfied. The last statement is easy to show, and it is the consequence of a rearrangement of the kinetic term:

$$\frac{1}{2}\sum_{\mu\neq\mu}(s_\mu p_\nu - p_\mu s_\nu)^2 = \sum_\mu p_\mu^2 \sum_{\nu(\neq\mu)}s_\nu^2 - \sum_\mu s_\mu p_\mu \sum_{\nu(\neq\mu)}s_\nu p_\nu$$
$$\approx \sum_\mu p_\mu^2\left(N - s_\mu^2\right) + \sum_\mu p_\mu^2 s_\mu^2 \approx \frac{1}{2m}\sum_\mu p_\mu^2, \tag{77}$$

where, to go from the first to the second line, we have used the primary constraint $\phi$ and the secondary constraint $\phi'$. In fact, the symbol $\approx$ denotes that the two sides are equal provided the constraints are satisfied. Considering these facts, we expect the solution of the saddle-point equations corresponding to

$$\gamma_\mu = -\lambda_\mu/(2T_0), \tag{78}$$

to be related to the equilibrium behaviour of $H_{\text{quad}}$, under the spherical constraint. In this Subsection we explore this connection.

We start by considering the first equation in (70). It is easy to show that, with the above-mentioned choice of the $\gamma_\mu$, we obtain

$$T_\mu = T_0. \tag{79}$$

This result is consistent with the equilibrium average induced by $H_{\text{quad}}$. It is important to stress that if we had not implemented the analytic continuation, Eq. (48), the result would have been

$$T_\mu = \frac{T_0}{1 - \frac{1}{N}\frac{T_\mu}{z - \lambda_\mu}} \underset{\text{Harm.}Ansatz}{=} \frac{T_0}{1 - \frac{1}{N}\langle s_\mu^2 \rangle_{\text{GGE}}}, \tag{80}$$

which departs from the equilibrium average imposed by $H_{\text{quad}}$ for modes such that $\langle s_\mu^2 \rangle_{\text{GGE}} = \mathcal{O}(N)$. This would be the case for the $N$th mode for condensed initial conditions. This remark justifies the introduction of the analytic continuation, given that having used it, the saddle-point equations reproduce the proper equilibrium results.

Additionally, if we investigate the consequences of choosing $\gamma_\mu = -\lambda_\mu/(2T_0)$ on the second equation in (70), we find that the solution reads $T_\mu/(z - \lambda_\mu) = T_0/(z_{\text{eq}} - \lambda_\mu)$ in the $N \to \infty$ limit, with $\bar{z}T + 1 = z = z_{\text{eq}}$, ensuring that the second equation in (70) is also satisfied.

In conclusion, solving the saddle-point equations with $\gamma_\mu = -\lambda_\mu/(2T_0)$ we see that, in the $N \to \infty$ limit, the GGE expectation values coincide with the ones obtained with the Maxwell-Boltzmann equilibrium measure of $H_{\text{quad}}$ at temperature $T_0$.

Finally, note that in equilibrium, Eqs. (74) have to be imposed setting $\lambda_\mu = \lambda_\mu^{(0)}$, in other words, $J = J_0$, and one should find $T_\mu = T_0$ for all $\mu$. If one compares the right-hand-side in Eq. (74) to the expressions for $\langle I_\mu \rangle_{\text{GGE}}$, it is not hard to see that they are identical with $T_\mu$ replaced by $T_0$ and $z$ by $z_{\text{eq}}$ showing that the latter are solutions to the set of $N$ coupled equations.

One can easily check that this equilibrium solution is the only one compatible with $\gamma_\mu$ being proportional to $\lambda_\mu$. To prove it, it is enough to set $\gamma_\mu = -a\lambda_\mu$ in the first equation in (70) and use the spherical constraint. The resulting equation is $T_\mu = 1/(2a)$ and one recovers a constant spectrum of temperatures, the one of equilibrium.

## 5 Analytic solution in the large $N$ limit

In this Section we derive, analytically, the spectra of mode temperatures and Lagrange multipliers in all phases of the phase diagram exposed in Fig. 5. We proceed differently in cases with $\langle I_N \rangle_{i.c.} = \mathcal{O}(1)$ (extended) and $\langle I_N \rangle_{i.c.} = \mathcal{O}(N)$ (condensed). A detailed comparison of the analytic expressions and the numerical solutions will be presented in Sec. 6.

### 5.1 Equations in the continuum limit

In the infinite $N$ limit, we can replace

$$\frac{1}{N}\sum_\mu f(\lambda_\mu) \to \int d\lambda\, \rho(\lambda)\, f(\lambda), \tag{81}$$

though in some cases we have to be careful and separate the contribution of the $N$th mode which could scale linearly with $N$. Within the harmonic *Ansatz* and in the continuum limit the

saddle-point Eqs. (70) read

$$2T(\lambda) = \left[ \fint' d\lambda' \, \rho(\lambda') \, \frac{T(\lambda')}{z - \lambda'} \left( \frac{\gamma(\lambda) - \gamma(\lambda')}{\lambda' - \lambda} \right) + q \, \frac{\gamma(\lambda) - \gamma_N}{\lambda_N - \lambda} (1 - \delta_{\lambda, \lambda_N}) \right]^{-1}$$

$$\int d\lambda \, \rho(\lambda) \, \frac{T(\lambda)}{z - \lambda} (2\gamma(\lambda) + \bar{z}) = 0 \, , \tag{82}$$

where $\fint$ indicates the principal value in the singularity at $\lambda = \lambda'$, and the prime stresses the fact that the contribution of the largest mode has been separately taken into account with the addition of the last term in the first equation. The term proportional to $q = \langle s_N^2 \rangle_{\mathrm{GGE}}/N$ can only be present for $\lambda \neq \lambda_N$ for parameters with condensation. We will also explore the possibility of having $T_N \propto N$ in which case a separate contribution to the integral in the second equation should also be considered. In short we define

$$q = \langle s_N^2 \rangle_{\mathrm{GGE}}/N \, , \qquad\qquad \tau = \langle p_N^2 \rangle_{\mathrm{GGE}}/(mN) \, , \tag{83}$$

and we see whether there are solutions with finite values of $q$ and $\tau$ in some parts of the phase diagram.

The GGE equations (75) take the form,

$$\begin{aligned}
\langle I(\lambda) \rangle_{\mathrm{GGE}} &= \frac{2T(\lambda)}{z - \lambda} \Big[ 1 - \fint' d\lambda' \, \rho(\lambda') \, \frac{T(\lambda')}{\lambda - \lambda'} \\
&\qquad\qquad - \frac{\tau}{\lambda - \lambda_N} (1 - \delta_{\lambda, \lambda_N}) - \frac{q}{2} \, \delta_{\lambda, \lambda_N} \Big] \\
&= \langle I(\lambda) \rangle_{i.c.} \, ,
\end{aligned} \tag{84}$$

for all $\lambda$ including $\lambda_N$. We used a loose notation in $\delta_{\lambda, \lambda_N}$ here and above. One can easily check that $\int d\lambda \, \rho(\lambda) \langle I(\lambda) \rangle_{\mathrm{GGE}} + \langle I_N \rangle_{\mathrm{GGE}}/N = \int d\lambda \, \rho(\lambda) \, T(\lambda)/(z - \lambda) + \langle s_N^2 \rangle_{\mathrm{GGE}}/N = 1$, where the integral run over the "bulk" and in some cases the additional terms are non-vanishing and contribute to the correct normalisation.

The $\langle I(\lambda) \rangle_{i.c.}$ and $\langle I_N \rangle_{i.c.}$ were calculated in the $N \to \infty$ limit in App. D and their parameter dependence summarised in Sec. 3.3. In this limit the saddle-point evaluations are fully justified. Taken together, Eqs. (82)-(84) constitute a closed system of integral equations for the functions $T(\lambda)$, $T_N$, $\gamma(\lambda)$, $\gamma_N$ and the parameters $z$ and $\bar{z}$. One of the numerical procedures that we employ uses Eq. (84) and the spherical constraint to fix $T(\lambda)$, $T_N$ and $z$. Then, the second equation in (82) determines $\bar{z}$, and the first one the ensemble of $\gamma$s. Surprisingly enough, these equations also admit an analytic solution which we expose in the next Subsections.

## 5.2 Temperature and multiplier spectra for $T_0 = (JJ_0)^{1/2}$

The *Ansatz*

$$T(\lambda) = J \langle I(\lambda) \rangle_{i.c.} \, , \tag{85}$$

with the explicit form of $\langle I(\lambda) \rangle_{i.c.} = k_1 (b - \lambda)/(a - \lambda)$ given in Eq. (37) and the parameter dependence of $k_1, a$ and $b$ given below this equation, solves Eqs. (82)-(84) on the special curve $T_0 = (JJ_0)^{1/2}$. Below we give some details of this solution for $T_0 \geq J_0$ and $T_0 < J_0$.

### 5.2.1 Extended cases $T_0 \geq J_0$

For the special choice of parameters $T_0 = (JJ_0)^{1/2}$ and $T_0 \geq J_0$ (phase II) there is no initial condensation, the constants of motion are all identical to one, $\langle I(\lambda) \rangle_{i.c.} = 1$, and the

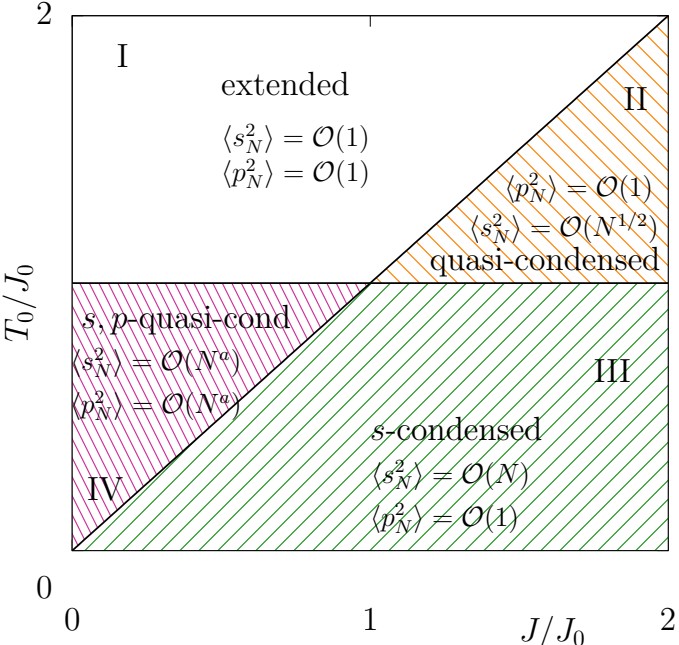

Figure 5: **The dynamic phase diagram.** The names of the phases refer to the scaling of $\langle s_N^2 \rangle$ and $\langle p_N^2 \rangle$ (both averaged over the GGE or the dynamics) with $N$ and the consequent condensation or quasi-condensation phenomena, see the explanation in the text. All transition lines are continuous.

total energy is $e_f = 0$. It turns out that the quenched system behaves as in canonical equilibrium at a single temperature, since all mode temperatures are identical, $T(\lambda) = T_f = J$ and the $\gamma(\lambda)$ become simply $-\beta_f \lambda/2$ (plus an additive constant which can be absorbed by the Lagrange multiplier $\bar{z}$). The latter identity can be checked by verifying that Eq. (82), or its discrete $\mu$ version Eq. (70), are solved by these $\gamma(\lambda)$ for any choice of $\beta_f$. One then has $\int d\lambda\, \rho(\lambda) \gamma(\lambda) I(\lambda) = -(\beta_f/2) \int d\lambda\, \rho(\lambda) \lambda\, I(\lambda) = \beta_f H_{\text{quad}}$. Therefore, the GGE measure reduces to the Gibbs-Boltzmann one. It is the constraint $\int d\lambda\, \rho(\lambda) \langle I(\lambda) \rangle_{\text{GGE}} = 1$ which imposes $T(\lambda) = T_f = J$. Moreover, although $\langle s(\lambda \to 2J) \rangle_{\text{GGE}}$ diverges, this divergence is integrable and the form

$$\langle s^2(\lambda) \rangle_{\text{GGE}} = \frac{J}{2J - \lambda} \tag{86}$$

correctly verifies the spherical constraint without any need to separate a macroscopic $\langle s_N^2 \rangle_{\text{GGE}}$. We therefore have

$$T(\lambda) = T_f = J \quad \text{and} \quad \gamma(\lambda) = -\frac{\lambda}{2J} \qquad \forall \lambda. \tag{87}$$

This spectrum of mode temperature together with $z = 2J$ yield the kinetic and potential energies $e_{\text{kin}}^f = J/2$ and $e_{\text{pot}}^f = -J/2$, consistently with the values given in Table I.

In [10] we solved the Schwinger-Dyson equations for parameters satisfying this particular relation and we found that the dynamics soon reached a stationary limit with the fluctuation-dissipation theorem holding at $T_f = J$. The results we have just derived for the GGE measure are in agreement with the system reaching conventional equilibrium at $T_f = J$ for these special parameters although we have extracted energy from the system in the quench. We also note that in the stationary regime, in which the time-dependent $z$ has reached its stationary limit $z_f = 2J$, the Hamiltonian becomes one of independent harmonic oscillators.

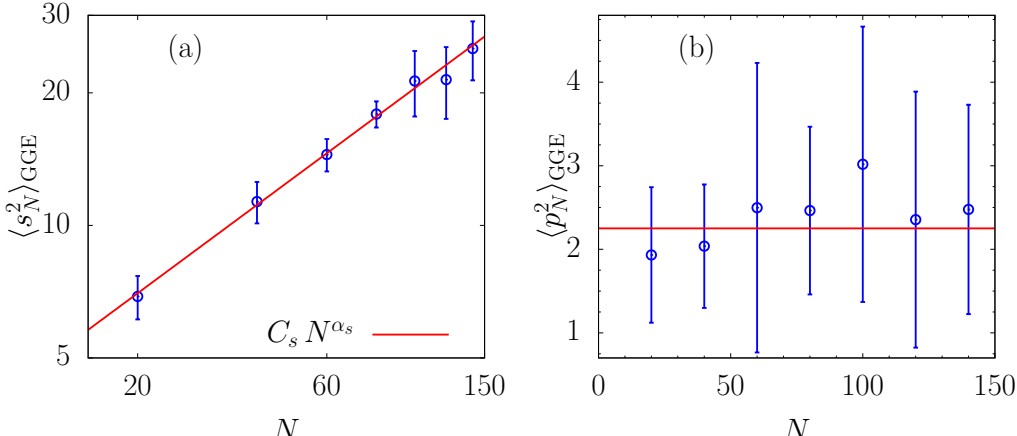

Figure 6: **Finite size dependence on the special curve $T_0 = (JJ_0)^{1/2}$ in phase II with parameters $T_0 = 1.5 J_0$ and $J = 2.25 J_0$.** (a) $\langle s_N^2 \rangle_{\text{GGE}}$, with a power law fit with parameters $C_s = 0.95$ and $\alpha_s = 0.66$. (b) $\langle p_N^2 \rangle_{\text{GGE}}$ and the horizontal line $\langle p_N^2 \rangle_{\text{GGE}} = J$. Though there are very large error bars, this constant falls well within them. The results have been averaged over up to 50 different realisations of the harmonic constants and the error bars represent the variance around the average.

In Fig. 6 we display finite $N$ data for parameters on the special curve in phase II. The data are consistent with $\langle p_N^2 \rangle_{\text{GGE}} = J$ ($m = 1$) and suggest a power law divergence of $\langle s_N^2 \rangle_{\text{GGE}}$ with $N$, with an exponent smaller than one, $\alpha_s \sim 0.66$.

### 5.2.2 Condensed cases $T_0 < J_0$

On the continuation of the curve $T_0 = (JJ_0)^{1/2}$ in phase IV, that is for $T_0 \leq J_0$, the averaged constants of motion in the bulk are not all identical. Still, the rather simple expressions

$$T(\lambda) = J \frac{z - \lambda}{2J - \lambda} \qquad \text{and} \qquad \gamma(\lambda) = \frac{1}{2J}(2J - \lambda)\frac{\lambda - J - \sqrt{JJ_0}}{\lambda - z}, \tag{88}$$

with $z = T_0 + J^2/T_0 = (J/J_0)^{1/2}(J + J_0)$ yield the exact solution of Eqs. (82) and (84) on this curve, with no need to separate the $N$th mode contributions. (We omitted the additive constant in $\gamma(\lambda)$.) A way to prove this result is to first solve for the spectrum of mode temperatures and then treat the set of equations that fix $\gamma(\lambda)$ with the *Ansatz* $\gamma(\lambda) = (2J)^{-1}(2J - \lambda)(\lambda - \nu)/(\lambda - z)$ and $\nu$ a parameter that is forced to take the form in Eq. (88). The $\lambda$ dependence of these expressions reduces to the one in (87) on the special curve in phase II. In the continuum limit, $T(\lambda)$ diverges at the edge of the spectrum but the divergence is integrable. One can check that $\int d\lambda \, \rho(\lambda) T(\lambda) = 2e_{\text{kin}}^f$ with $2e_{\text{kin}}^f$ given in the last line of Table I. Concomitantly, $\gamma(2J)$ vanishes. We note that

$$\langle s^2(\lambda) \rangle_{\text{GGE}} = \frac{J}{2J - \lambda} \tag{89}$$

on the whole special curve both in II and IV, and the spherical constraint is satisfied all along it. For these reasons it is not necessary to separate the contribution of the $N$th mode in Eqs. (82) and (84) when working on the special curve.

### 5.3 Temperature spectra in the extended phases I and II

It turns out that one can find a general solution for $T(\lambda)$ everywhere in the phase diagram. In this Section we describe the construction of this solution in phases I and II.

In cases with $T_0 \geq J_0$ we can neglect $\langle s_N^2 \rangle_{\mathrm{GGE}}/N$ and rewrite Eq. (84) in the form

$$\rho(\lambda)\frac{(z-\lambda)}{2}\langle I(\lambda)\rangle_{i.c.} = \rho(\lambda)T(\lambda)\Big[1 - \int d\lambda' \rho(\lambda')\frac{T(\lambda')}{\lambda-\lambda'}\Big], \tag{90}$$

with $\langle I(\lambda)\rangle_{i.c.}$ given in Eq. (37). We will search an exact expression for $\pi\rho(\lambda)T(\lambda)$.

Let us define the complex function $\zeta(\lambda) = \zeta_{\mathrm{R}}(\lambda) + i\zeta_{\mathrm{I}}(\lambda)$. The Kramers-Kronig relations link its imaginary and real parts, $\zeta_{\mathrm{I}}(\lambda)$ and $\zeta_{\mathrm{R}}(\lambda)$, according to

$$\zeta_{\mathrm{I}}(\lambda) = -\frac{1}{\pi}\int d\lambda' \frac{\zeta_{\mathrm{R}}(\lambda')}{\lambda'-\lambda}, \qquad\qquad \zeta_{\mathrm{R}}(\lambda) = \frac{1}{\pi}\int d\lambda' \frac{\zeta_{\mathrm{I}}(\lambda')}{\lambda'-\lambda}, \tag{91}$$

if the decay of $|\zeta(\lambda)|$ at infinity is at least as fast as $1/|\lambda|$. Identifying

$$\zeta_{\mathrm{I}}(\lambda) = \pi\rho(\lambda)T(\lambda), \tag{92}$$

Eq. (90) becomes

$$\zeta_{\mathrm{I}}(\lambda)(1 + \zeta_{\mathrm{R}}(\lambda)) = g(z,\lambda) \equiv \frac{\pi}{2}\rho(\lambda)(z-\lambda)\langle I(\lambda)\rangle_{i.c.}. \tag{93}$$

For $|\lambda| < 2J$, $g \neq 0$ and this implies

$$\zeta_R \neq -1 \qquad \text{and} \qquad \zeta_I = g/(1+\zeta_{\mathrm{R}}). \tag{94}$$

For $|\lambda| \geq 2J$, $g = 0$ and

$$\zeta_I = 0 \qquad \text{or} \qquad \zeta_R = -1. \tag{95}$$

Noticing that $\zeta^2 = (\zeta_{\mathrm{R}}^2 - \zeta_{\mathrm{I}}^2) + 2i\zeta_{\mathrm{R}}\zeta_{\mathrm{I}}$ implies $2\zeta_{\mathrm{R}}\zeta_{\mathrm{I}} = \mathrm{Im}\zeta^2$, the left-hand-side of Eq. (93) can also be written as $\mathrm{Im}(\zeta + \zeta^2/2 + f)$, with $f \in \mathbb{R}$ that is $\mathrm{Im}f = 0$. Then,

$$\mathrm{Im}(\zeta + \zeta^2/2 + f) = \mathrm{Im}(\zeta + \zeta^2/2) = g. \tag{96}$$

The (opposite) Kramers-Kronig relation applied to the complex function $\zeta + \zeta^2/2 + f$ leads to

$$\mathrm{Re}(\zeta + \zeta^2/2 + f) = \frac{1}{\pi}\int d\lambda' \frac{\mathrm{Im}(\zeta + \zeta^2/2 + f)}{\lambda'-\lambda} = \frac{1}{\pi}\int d\lambda' \frac{g(\lambda')}{\lambda'-\lambda} \equiv \frac{1}{2}G. \tag{97}$$

Using $\mathrm{Re}(\zeta + \zeta^2/2 + f) = \zeta_{\mathrm{R}} + (\zeta_{\mathrm{R}}^2 - \zeta_{\mathrm{I}}^2)/2 + f$ we can now distinguish $g \neq 0$ ($|\lambda| < 2J$) from $g = 0$ ($|\lambda| > 2J$).

In the case $g \neq 0$, $\mathrm{Re}(\zeta + \zeta^2/2 + f) = \zeta_{\mathrm{R}} + \zeta_{\mathrm{R}}^2/2 + f - g^2/[2(1+\zeta_{\mathrm{R}})^2]$ yields

$$\zeta_{\mathrm{R}} + \frac{\zeta_{\mathrm{R}}^2}{2} + f - \frac{g^2}{2(1+\zeta_{\mathrm{R}})^2} = \frac{1}{2}G, \tag{98}$$

which, after adding and subtracting $1/2$ and rearranging a little bit the various terms, becomes an equation that fixes $(1+\zeta_{\mathrm{R}})^2$:

$$0 = (1+\zeta_{\mathrm{R}}(\lambda))^4 - (1+\zeta_{\mathrm{R}}(\lambda))^2[1 + G(\lambda) - 2f(\lambda)] - g^2(\lambda), \tag{99}$$

or, in terms of $\zeta_{\mathrm{I}}^2$,

$$0 = \zeta_{\mathrm{I}}^4(\lambda) + \zeta_{\mathrm{I}}^2(\lambda)[1 + G(\lambda) - 2f(\lambda)] - g^2(\lambda). \tag{100}$$

Both are simple bi-quadratic equations, the first one for $1 + \zeta_{\mathrm{R}}$, the second one for $\zeta_{\mathrm{I}}$. The solutions read

$$2\zeta_{\mathrm{I}}^2(\lambda) = -[1 + G(\lambda) - 2f(\lambda)] + \{[1 + G(\lambda) - 2f(\lambda)]^2 + 4g^2(\lambda)\}^{1/2}, \tag{101}$$

$$2(1+\zeta_{\mathrm{R}}(\lambda))^2 = [1 + G(\lambda) - 2f(\lambda)] + \{[1 + G(\lambda) - 2f(\lambda)]^2 + 4g^2(\lambda)\}^{1/2}, \tag{102}$$

for $\lambda \in [-2J, 2J]$, where we chose the positive signs to ensure the positivity of the results. It is easy to check that these expressions verify (93) for any $f$.

In the case $g = 0$, that is, outside the interval $[-2J, 2J]$, $\zeta_I = 0$ or $\zeta_R = -1$. Then, Eq. (97) implies

$$\zeta_I = 0 \quad \Rightarrow \quad \zeta_R + \frac{\zeta_R^2}{2} + f = \frac{G}{2} \quad \Rightarrow \quad \zeta_R = -1 \pm (1 + G - 2f)^{1/2}, \tag{103}$$

$$\zeta_R = -1 \Rightarrow \quad 1 + \zeta_I^2 = 2f - G \quad \Rightarrow \quad \zeta_I = (2f - 1 - G)^{1/2}. \tag{104}$$

The first line gives a continuous function $\zeta_R$ at $|\lambda| = 2J$, where $g = 0$, if we keep the plus sign. The second line would give discontinuous $\zeta_R$ and $\zeta_I$. We therefore select

$$\zeta_I(\lambda) = 0, \qquad \zeta_R(\lambda) = -1 + (1 + G(\lambda) - 2f(\lambda))^{1/2}, \qquad \text{for } |\lambda| > 2J. \tag{105}$$

Having an explicit expression for $\zeta_I(\lambda)$ in the interval $[-2J, 2J]$, written in Eq. (101), we know the spectrum of mode temperatures $T(\lambda) = \zeta_I(\lambda)/[\pi\rho(\lambda)]$ for $|\lambda| < 2J$. With this trick, the solution is parametrized by an unknown function $f(\lambda)$.

To go further, we need the explicit forms of $g$ and $G$. The former is

$$g(\lambda) = \frac{1}{4J^2} \sqrt{4J^2 - \lambda^2} \, \theta(2J - |\lambda|)(z - \lambda) k_1 \frac{b - \lambda}{a - \lambda}, \tag{106}$$

where

$$k_1 = \frac{T_0^2}{J_0 J} \qquad \text{and} \qquad b = \frac{J(J + J_0)}{T_0} \qquad \text{in all cases,} \tag{107}$$

and

$$z = \begin{cases} T_0 + \dfrac{J^2}{T_0} & \text{if } T_0 > J, \\ 2J & \text{if } T_0 < J, \end{cases} \qquad a = \begin{cases} \dfrac{J}{T_0}\left(J_0 + \dfrac{T_0^2}{J_0}\right) & \text{if } T_0 > J_0, \\ 2J & \text{if } T_0 < J_0. \end{cases} \tag{108}$$

Note that $a > 2J$ for $T_0 \geq J_0$ and $a = 2J$ for $T_0 \leq J_0$. It will also be important to notice that $a = z$ in the full phase III and $b = z$ in phase IV. In the case $T_0 < J_0$ we have to take care of the condensed mode too.

The integral in $G$ reads

$$G(\lambda) \equiv \frac{2}{\pi} \int d\lambda' \, \frac{g(\lambda')}{\lambda' - \lambda} = -k_1 \int d\lambda' \rho(\lambda') \frac{(z - \lambda')(b - \lambda')}{(\lambda - \lambda')(a - \lambda')}, \tag{109}$$

where we used the fact that the constants of motion averaged over the initial conditions are rational functions. This integral has been discussed in App. A, see Eq. (A.6), and its result depends on whether $\lambda$ and $a$ belong to the interval $[-2J, 2J]$ or not. We know that $a$ is outside or at the border of this interval. Therefore,

$$G(\lambda) \begin{cases} = \dfrac{k_1}{2J^2}\left[\lambda^2 - \lambda(b + z - a) + (b - a)(z - a) + \dfrac{\sqrt{a^2 - 4J^2}(a - b)(a - z)}{(\lambda - a)} - 2J^2\right], \\ \xrightarrow{\lambda \to \infty} \text{finite}. \end{cases}$$

The first line refers to $|\lambda| < 2J$, where the function $G$ is regular, even at its boundary, $\lambda = 2J$. Outside the interval its expression changes, it is given in App. A, we do not need to repeat it here, and one can check that it approaches a constant in the infinite $\lambda$ limit.

Summarising,

$$2[\pi\rho(\lambda)T(\lambda)]^2 = -[1+G(\lambda)-2f(\lambda)] + \left\{[1+G(\lambda)-2f(\lambda)]^2 + 4g^2(\lambda)\right\}^{1/2}, \qquad (110)$$

with $g(\lambda)$ in Eq. (106), the parameters $a, b, z$ specified below this equation, and $G(\lambda)$ in the last unnumbered equation above, with the same parameters. More details on the functions $g$ and $G$ are given in App. E. The real function $f(\lambda)$ is, for the moment, free. For extended situations, as in I and II, we can safely set it to zero. We will see below that the same can be done in phase III, while in phase IV we need to take a special form of $f$.

We here summarise some salient features of the spectrum of mode temperatures in phases I and II which are deduced from the solution above. Their detailed derivation is given in App. E.

- In phase I, $z > 2J$. The averages $0 < T(2J) = \langle p^2(2J)\rangle_{\text{GGE}}$ and $0 < \langle s^2(2J)\rangle_{\text{GGE}} = (z-2J)\langle p^2(2J)\rangle_{\text{GGE}}$ are both finite, as well as the full spectrum of mode temperatures.

- In phase II, $z = 2J$, but one can check that $\langle p^2(2J-\epsilon)\rangle_{\text{GGE}} = T(2J-\epsilon) = \mathcal{O}(1)$ (with $\epsilon \to 0$). Consequently, $\langle s^2(2J-\epsilon)\rangle_{\text{GGE}}$ diverges at the edge. Nevertheless, the divergence is integrable and there is no need to separate an $\mathcal{O}(N)$ contribution of the last mode to ensure the validity of the spherical constraint. This is confirmed by the numerical solution of the GGE and saddle-point equations for finite $N$, see Fig. 6 (on the special curve) and Fig. 7 (away from it). Both plots allow us to confirm that $\langle s^2(\lambda_N)\rangle_{\text{GGE}}$ does not grow linearly with $N$. Away from the special curve a fit suggests $\langle s^2(\lambda_N)\rangle_{\text{GGE}} \sim N^{1/2}$ but the large error bars inhibit us from fully justifying this law. The numerical data support the finite limit of $\langle p^2(\lambda_N)\rangle_{\text{GGE}}$ as well.

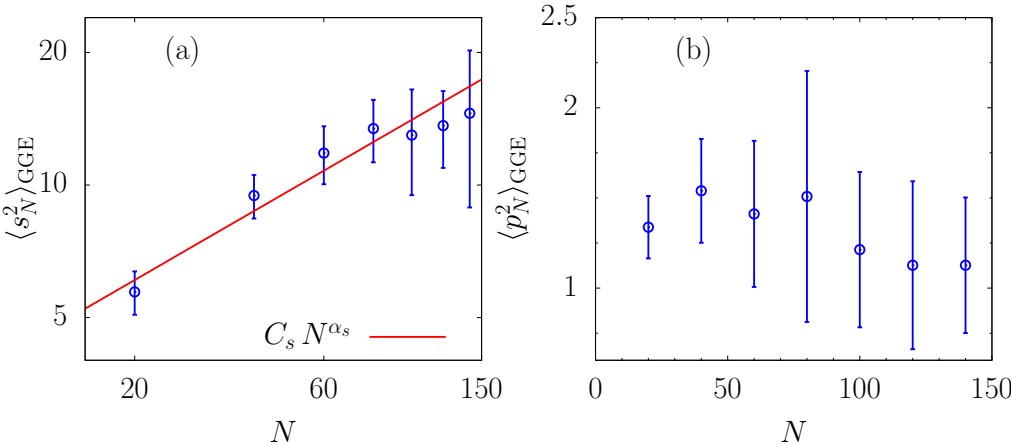

Figure 7: **Finite size dependence in phase II, away from the special curve** $T_0 = (JJ_0)^{1/2}$, **for parameters** $T_0 = 1.5J_0$ **and** $J = 1.7J_0$. (a) $\langle s_N^2\rangle_{\text{GGE}}$, with a power law fit with parameters $C_s = 1.28$, $\alpha_s = 0.52$ (b) $\langle p_N^2\rangle_{\text{GGE}}$. The results have been averaged over up to 50 different realisations of the harmonic constants and the error bars represent the variance.

## 5.4 Temperature spectra in the condensed phases

We are now in a position to treat the cases with condensation of the $N$th mode:

$$\langle s_N^2\rangle_{\text{GGE}} = qN, \qquad \langle p_N^2\rangle_{\text{GGE}} = T_N = \tau N, \qquad (111)$$

with $q$ and $\tau$ finite, but possibly vanishing, and $m = 1$.

We start by rewriting the GGE Eqs. (84) in the form

$$
\frac{(z-\lambda)}{2} \langle I(\lambda) \rangle_{i.c.} = T(\lambda) \left[ 1 - \int' d\lambda' \rho(\lambda') \frac{T(\lambda')}{\lambda - \lambda'} - \frac{\tau}{\lambda - \lambda_N} \right] \qquad \lambda \neq \lambda_N ,
$$
$$
\frac{1}{2} \langle I(\lambda_N) \rangle_{i.c.} = qN \left[ 1 - \int d\lambda' \rho(\lambda') \frac{T(\lambda')}{\lambda_N - \lambda'} - \frac{q}{2} \right] \qquad \lambda = \lambda_N ,
$$

(112)

and we search a solution for the bulk $\pi \rho(\lambda) T(\lambda)$, and the separate edge values $\tau = q(z - \lambda_N)$.

### 5.4.1 Phase III

In phase III one expects $\tau = 0$ and $\langle s_N^2 \rangle_{\text{GGE}} = qN = \mathcal{O}(N)$. Therefore, in the first equation in (112) one can neglect the last term within the square brackets and obtain the spectrum of temperatures in the bulk in the same way as we did in Sec. 5.3, leading to Eq. (110) with $f = 0$. Once the function $T(\lambda)$ for $\lambda \neq \lambda_N$ is known, one can safely continue it to $\lambda \to \lambda_N$ finding a finite value, in agreement with the no need to separate the $N$th component contribution.

The second quadratic equation in (112) fixes $q$:

$$
\frac{1}{2} \left( 1 - \frac{T_0}{J_0} \right) \left( 1 - \frac{T_0}{J} \right) = q \left[ 1 - \int d\lambda' \rho(\lambda') \frac{T(\lambda')}{2J - \lambda'} - \frac{q}{2} \right].
$$

(113)

The integral is finite (the divergence at the edge is integrable) and using the spherical constraint is simply given by $1 - q$. Thus,

$$
q^2 = \left( 1 - \frac{T_0}{J_0} \right) \left( 1 - \frac{T_0}{J} \right).
$$

(114)

Consistently, $q$ vanishes at the borders of phase III, both for $T_0 = J_0$ and $T_0 = J$, and is identical to one for $T_0 = 0$. Otherwise, it takes values in the interval $(0, 1)$. (An alternative way of fixing $q$ is to integrate Eq. (84) over $\lambda$ with the weight $\rho(\lambda)$ excluding the last mode, that is, taking $\int' d\lambda \, \rho(\lambda) \ldots$. In a few steps one recovers Eq. (113) and from it (114).)

### 5.4.2 Phase IV

In phase IV, $T_N$ could be $\mathcal{O}(N)$ and the contribution of the last term in the right-hand-side of the first equation in (112) should not be neglected *a priori*. We use the knowledge of the exact solution on the special curve $T_0 = (JJ_0)^{1/2}$, see Sec. 5.2.2, as a guideline to build the solution on the full phase IV with the current method. We thus find that there is only quasi condensation of both the $N$th coordinate and momentum in this phase. More precisely, $\langle s^2(\lambda) \rangle$ and $\langle p^2(\lambda) \rangle$ calculated in the $N \to \infty$ limit diverge at the edge of the spectrum, but $\langle s_N^2 \rangle_{\text{GGE}}$ and $\langle p_N^2 \rangle_{\text{GGE}}$ scale sub-linearly with $N$ (contrary to what we wrote in [12]).

*The special curve*

First, we verify that the already known spectrum of mode temperatures (88) solves the generic equations for parameters on the special curve. Using the $T(\lambda)$ in Eq. (88), the complex

$\zeta(\lambda)$ function reads

$$\zeta_I(\lambda) = \begin{cases} \dfrac{1}{2J}\sqrt{\dfrac{2J+\lambda}{2J-\lambda}}\,(z-\lambda)\,, & \lambda \in [-2J,2J]\,, \\[2ex] 0\,, & \lambda \notin [-2J,2J]\,, \end{cases} \tag{115}$$

$$\zeta_R(\lambda) = \begin{cases} \dfrac{1}{2J}\,(z-2J-\lambda)\,, \\[2ex] \dfrac{1}{2J}\dfrac{1}{2J-\lambda}\left\{(z-2J)2J-(z-\lambda)\left[\lambda-\sqrt{\lambda^2-(2J)^2}\right]\right\}\,, \end{cases} \tag{116}$$

where $\zeta_I$ was written from its definition and $\zeta_R$ derived from it using the Kramers-Kronig relation. Within the interval $[-2J,2J]$ the expressions above yield

$$\zeta_I(1+\zeta_R) = \pi\rho(\lambda)T(\lambda)(1+\zeta_R) = \frac{\pi}{2}\rho(\lambda)(z-\lambda)\frac{1}{J}T(\lambda)\,, \tag{117}$$

where we used $\zeta_I = \pi\rho(\lambda)T(\lambda)$. After several cancellations we recover the first expression for $\zeta_R$ in Eq. (116). If, instead, we use the just derived result in Eq. (116)

$$\pi\rho(\lambda)T(\lambda)(1+\zeta_R) = \pi\rho(\lambda)T(\lambda)\left[1+\frac{1}{2J}(z-\lambda-2J)\right] = \frac{\pi}{2}\rho(\lambda)\frac{(z-\lambda)^2}{2J-\lambda}\,, \tag{118}$$

and one recovers Eq. (88). Outside of the interval $[-2J,2J]$, Eq. (117) is just $0=0$. Thus, the $T(\lambda)$ we knew is consistent with the generic equations.

Now, we now want to obtain

$$\pi\rho(\lambda)T(\lambda) \equiv \zeta_I(\lambda) = \frac{1}{2J}\sqrt{\frac{2J+\lambda}{2J-\lambda}}\,(z-\lambda)\,, \tag{119}$$

from the generic expression (101). On the curve $T_0 = (JJ_0)^{1/2}$,

$$\begin{aligned} G(\lambda)+1 &= \frac{1}{2J^2(2J-\lambda)}\left[(z-2J)^2\,2J-(z-\lambda)^2\,\lambda\right] \\[2ex] &= \frac{1}{2J^2}\left[\lambda^2-2\lambda(z-J)+(z-2J)^2\right]\,, \end{aligned} \tag{120}$$

while

$$g(\lambda) = \frac{1}{4J^2}\left(\frac{2J+\lambda}{2J-\lambda}\right)^{1/2}(z-\lambda)^2\,. \tag{121}$$

At $\lambda = 2J$, $G(\lambda)$ is regular while $g(\lambda)$ diverges as a square root. If $f(\lambda)$ were also regular at this edge, the generic solution (101) would imply $\zeta_I \sim g^{1/2} \sim (2J-\lambda)^{-1/4}$, while we know that $\zeta_I \sim (2J-\lambda)^{-1/2}$. Therefore, the function $f$ should be different from zero and dominate the behaviour at $\lambda = 2J$. The idea is then to fix the function $f$ by looking at the behaviour close to the edge,

$$2\zeta_I^2 = -[1+G-2f]+\left\{[1+G-2f]^2+4g^2\right\}^{1/2} \sim 2f+|-2f| = 4f\,, \tag{122}$$

which, using the expected form of $\zeta_I^2$ with $T(\lambda) \sim J(z-2J)/(2J-\lambda)$ in this same limit, is solved by

$$f(\lambda) = \frac{1}{2J}\frac{(z-2J)^2}{2J-\lambda} \qquad \text{for } \lambda \simeq 2J\,. \tag{123}$$

We note that $f(\lambda)$ is identical to zero for $J = J_0$, where $z \to 2J$ and phase IV joins phase III.

Introducing now this $f$ in the second member of Eq. (122), with $G$ from Eq. (120) and $g$ from Eq. (121), one recovers the correct solution $T(\lambda) = J(z - \lambda)/(2J - \lambda)$ for all $\lambda$ on the special curve $T_0 = (JJ_0)^{1/2}$. We conclude that the addition of a function $f$ with the properties underlined above is instrumental to find the correct temperature spectrum on the special curve in IV.

Finally, we need to check that the second equation in (112), after replacing $\lambda_N = 2J$, is compatible with vanishing $q$ and $\tau$ on the special curve in IV. This equation reads

$$\iota \equiv \frac{\langle I(\lambda_N)\rangle_{i.c.}}{N} = 2q\left[1 - \fint d\lambda' \rho(\lambda') \frac{T(\lambda')}{2J - \lambda'} - \frac{q}{2}\right], \tag{124}$$

and determines $q$. (Alternatively, one can take the first equation in (112) and integrate it over $\lambda$ with the weight $\rho(\lambda)$ excluding the last mode to find this same equation.) The integral can be estimated from the already known $T(\lambda)$ for $\lambda \neq \lambda_N$ and it diverges. Indeed, since $T(\lambda) = J(z - \lambda)/(2J - \lambda)$ the integrand has a factor $1/(2J - \lambda)^2$ which makes it divergent. Thus, $q$ must vanish on this curve, as well as $\tau$, in the $N \to \infty$ limit, to let the left-hand-side be finite.

*Away from the special curve*

Going back to Eq. (112) with the inclusion of the last non-vanishing term in the square brackets for generic parameters in phase IV, the steps detailed in the previous Subsection are only slightly modified to yield

$$0 = \zeta_I^4(\lambda) + \zeta_I^2(\lambda)\left[\left(1 - \frac{\tau}{\lambda - 2J}\right)^2 + G(\lambda) - 2f(\lambda)\right] - g^2(\lambda), \tag{125}$$

for $\lambda \neq \lambda_N$. Our guess now is that in the full phase IV, and close to the edge,

$$2\tilde{f}(\lambda) \equiv \frac{2\tau}{\lambda - 2J} - \frac{\tau^2}{(\lambda - 2J)^2} + 2f(\lambda) = \frac{2\kappa(T_0/J_0, J_0/J)}{2J - \lambda}. \tag{126}$$

If this were so,

$$(\pi\rho T)^2 = \zeta_I^2 \sim 2\tilde{f} \sim \frac{2\kappa}{(2J - \lambda)} \implies T(\lambda \sim 2J) \sim \frac{(2\kappa J^3)^{1/2}}{2J - \lambda}, \tag{127}$$

with $\kappa$ to be determined as a function of the control parameters. We know already that it should satisfy $\kappa = 0$ on the horizontal line $T_0 = J_0$ and on the diagonal $T_0 = J$. So one could expect the numerator to be proportional to $z - 2J$ from the second condition. The first condition is achieved by another factor $(T_0 - J_0)^2/J_0^{3/2}$ which equals $z - 2J$ on the special line. Moreover, one should recover the expression in (123) for parameters on the special curve. Hence we propose

$$\kappa(T_0/J_0, J_0/J) = \frac{1}{2}\sqrt{\frac{T_0}{J^2}}(z - 2J)^{3/2}\left(1 - \frac{T_0}{J_0}\right). \tag{128}$$

We note that $T(\lambda \sim 2J)$ diverges as $(2J - \lambda)^{-1}$ and is integrable over the interval with semi-circle law weight. After some replacements and simplifications, once written in terms of adimensional parameters the numerator in Eq. (127) reads

$$\kappa = \frac{J_0^2}{2J}\left|\frac{T_0}{J_0} - \frac{J}{J_0}\right|^3\left(\frac{J_0}{T_0} - 1\right). \tag{129}$$

Introducing the $2\tilde{f}$ given in Eq. (126) with this $\kappa$ in Eq. (125), we find the full parameter dependence of $\zeta_I$ and hence $T(\lambda)$ for all $\lambda$.

We then checked numerically that this form leads to results which are in excellent agreement with what we get for the bulk temperatures from the direct solution of the saddle-point and GGE equations. Moreover, we verified that the integral of $T(\lambda)$ coincides with twice the kinetic energy density and that $\langle s^2(\lambda) \rangle$ is normalised to one with no need to consider separate contributions from $q$ and $\tau$. Therefore they vanish not only on the special line but in the full phase IV. This is once again justified by the fact that the integral in Eq. (124) diverges.

*Summary*

Let us now summarise the salient features of the spectrum of mode temperatures and the observables in phases III and IV of the phase diagram.

- In phase III, $T(2J) = 1/2 \sqrt{T_0 J/J_0} (J_0 + J - 2T_0)/\sqrt{T_0 - J - J_0}$ is finite as well as the full spectrum of temperatures. Instead, from the bulk solution we obtain that $\langle s^2(2J - \epsilon) \rangle_{\text{GGE}}$ is inversely proportional to $z - 2J + \epsilon$ and diverges for $\epsilon \to 0$. Moreover, to ensure the validity of the averaged spherical constraint one has to treat the $N$th mode contribution separately, and fix $\langle s_N^2 \rangle_{\text{GGE}} = qN$ from Eq. (114).

- In phase IV both bulk $T(\lambda)$ and $\langle s^2(\lambda) \rangle$ diverge close to the border with $(2J - \lambda)^{-1}$ but we do not need to separate the contributions of the $N$th mode. Indeed, the $N$th mode averages $\langle s_N^2 \rangle_{\text{GGE}}$ and $\langle p_N^2 \rangle_{\text{GGE}}$ are sub-linear in $N$ and do not contribute to the macroscopic values of, e.g., the averaged kinetic energy and spherical constraint. Numerical evaluations of the $N$ dependencies of $\langle p_N^2 \rangle_{\text{GGE}}$ and $\langle s_N^2 \rangle_{\text{GGE}}$ are shown in Fig. 8 away from the special curve. The relative values are in good agreement with the harmonic relation $\langle p_N^2 \rangle_{\text{GGE}} = (z - \lambda_N) \langle s_N^2 \rangle_{\text{GGE}}$ and the sub-linear growth with $N$ is exhibited by the red lines.

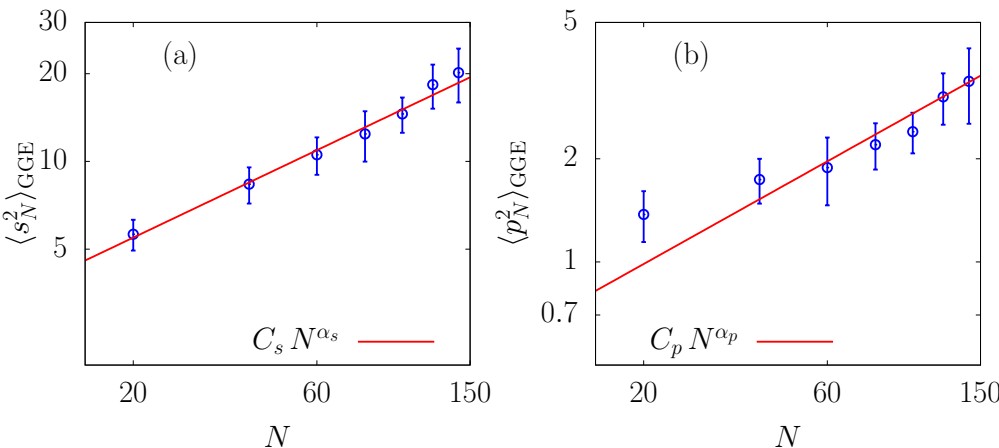

Figure 8: **Finite size dependence in phase IV for parameters $T_0 = 0.8 J_0$ and $J = 0.5 J_0$, which lie away from the special curve $T_0 = (J J_0)^{1/2}$.** (a) $\langle s_N^2 \rangle_{\text{GGE}}$ against $N$ and a power law fit to all data points with $\alpha_s = 0.62 \pm 0.03$. (b) $\langle p_N^2 \rangle_{\text{GGE}}$ against $N$, together with the power $C_p N^{\alpha_s}$ with $\alpha_s = 0.62$ (in red) which describes the data quite satisfactorily, apart from the first data point at $N = 20$. (A fit to all points including the $N = 20$ one yields $\alpha_p = 0.4 \pm 0.05$ which is not consistent with the harmonic *Ansatz*.) The results have been averaged over up to 50 disorder realisations and the error bars represent the variance around the average.

## 5.5 The multipliers $\gamma(\lambda)$

The equations that fix the multipliers $\gamma(\lambda)$, Eqs. (82), multiplied by $\pi\rho(\lambda)/(z-\lambda)$ become

$$
\begin{aligned}
\frac{\pi\rho(\lambda)}{z-\lambda} = &-2\pi\frac{\gamma(\lambda)\rho(\lambda)T(\lambda)}{z-\lambda}\fint d\lambda'\,\frac{\rho(\lambda')\,T(\lambda')}{(z-\lambda')(\lambda-\lambda')} \\
&+2\pi\frac{\rho(\lambda)T(\lambda)}{z-\lambda}\fint d\lambda'\,\frac{\rho(\lambda')\,T(\lambda')\gamma(\lambda')}{(z-\lambda')(\lambda-\lambda')} \\
&+2K_I(\lambda),
\end{aligned}
\tag{130}
$$

with

$$
K_I(\lambda)\equiv\pi\frac{\rho(\lambda)T(\lambda)}{z-\lambda}\,q\,\frac{\gamma(\lambda)-\gamma_N}{\lambda_N-\lambda}(1-\delta_{\lambda N}),
\tag{131}
$$

for $\lambda\neq\lambda_N$. Note that $K_I(\lambda)$ depends on $\gamma(\lambda)$. There is also the integral constraint in the second Eq. (82) to be taken into account. We define the real and imaginary parts of two complex functions $\phi$ and $\Xi$

$$
\phi_I(\lambda)=\pi\frac{\rho(\lambda)T(\lambda)}{z-\lambda}, \qquad \phi_R(\lambda)=-\fint d\lambda'\,\frac{\rho(\lambda')T(\lambda')}{(z-\lambda')(\lambda-\lambda')},
\tag{132}
$$

$$
\Xi_I(\lambda)=\pi\frac{\rho(\lambda)T(\lambda)\gamma(\lambda)}{(z-\lambda)}, \qquad \Xi_R(\lambda)=-\fint d\lambda'\,\frac{\rho(\lambda')T(\lambda')\gamma(\lambda')}{(z-\lambda')(\lambda-\lambda')},
\tag{133}
$$

and we use them to rewrite Eq. (130) as

$$
\begin{aligned}
h_I(\lambda)-K_I(\lambda) &\equiv \frac{\pi\rho(\lambda)}{2(z-\lambda)}-K_I(\lambda) \\
&= \Xi_I(\lambda)\phi_R(\lambda)-\Xi_R(\lambda)\phi_I(\lambda)=\mathrm{Im}[\Xi(\lambda)\phi^*(\lambda)+l(\lambda)],
\end{aligned}
\tag{134}
$$

where we defined two real functions $h_I$ and $l(\lambda)$, the first one with a known expression and the second one to be fixed below. Concomitantly,

$$
\begin{aligned}
h_R(\lambda)-K_R(\lambda) &= -\fint d\lambda'\,\frac{\rho(\lambda')}{2(z-\lambda')(\lambda-\lambda')}-\frac{1}{\pi}\fint d\lambda'\,\frac{K_I(\lambda')}{(\lambda-\lambda')} \\
&= \mathrm{Re}[\Xi(\lambda)\phi^*(\lambda)]+l(\lambda) \\
&= \Xi_R(\lambda)\phi_R(\lambda)+\Xi_I(\lambda)\phi_I(\lambda)+l(\lambda).
\end{aligned}
\tag{135}
$$

We choose $l(\lambda)=-K_R(\lambda)+a$ to get rid of contributions to Eqs. (134) and (135) that would have the unknowns within an integration (ensured by the first term $-K_R$ in $l$) and the correct result in the equilibrium limit (the addition of the term $a$ to be fixed below) Equations (134)-(135) form a set of two linear equations for the unknown $\Xi$:

$$
\begin{pmatrix}h_I-K_I\\h_R-a\end{pmatrix}=\begin{pmatrix}-\phi_I & \phi_R\\\phi_R & \phi_I\end{pmatrix}\begin{pmatrix}\Xi_R\\\Xi_I\end{pmatrix},
\tag{136}
$$

with solution

$$
\begin{pmatrix}\Xi_R\\\Xi_I\end{pmatrix}=\frac{1}{\phi_R^2+\phi_I^2}\begin{pmatrix}-\phi_I & \phi_R\\\phi_R & \phi_I\end{pmatrix}\begin{pmatrix}h_I-K_I\\h_R-a\end{pmatrix},
\tag{137}
$$

which implies

$$
\Xi_I=\frac{1}{\phi_R^2+\phi_I^2}\left[\phi_R(h_I-K_I)+\phi_I(h_R-a)\right].
\tag{138}
$$

Using now the definitions of $\Xi_I$ and $K_I$, with $qN=\langle s_N^2\rangle_{\mathrm{GGE}}$, and $a=K_I\,\phi_R/\phi_I$

$$
(\phi_R^2+\phi_I^2)\gamma(\lambda)+\phi_R\,2q\,\frac{\gamma(\lambda)-\gamma_N}{\lambda_N-\lambda}=\frac{(z-\lambda)}{\pi\rho(\lambda)T(\lambda)}(\phi_R h_I+\phi_I h_R).
\tag{139}
$$

The choice of $a$ will be clear below, when comparing the generic result to the expected one in standard equilibrium ($J=J_0$). This equation fixes the spectrum $\gamma(\lambda)$.

### 5.5.1 Useful identities

Before making explicit the parameter dependence of $\gamma(\lambda)$ in the various phases of the phase diagram, we present two identities that will be useful to evaluate the right-hand-side of Eq. (139):

$$-\phi_R(\lambda) = \int d\lambda' \frac{\rho(\lambda')T(\lambda')}{(z-\lambda')(\lambda-\lambda')} = -\frac{1}{z-\lambda}\left[1 - q - \fint d\lambda' \frac{\rho(\lambda')T(\lambda')}{\lambda-\lambda'}\right], \quad (140)$$

$$-2h_R(\lambda) = \int d\lambda' \frac{\rho(\lambda')}{(z-\lambda')(\lambda-\lambda')} = -\frac{1}{z-\lambda}\left[\fint d\lambda' \frac{\rho(\lambda')}{z-\lambda'} - \fint d\lambda' \frac{\rho(\lambda')}{\lambda-\lambda'}\right]$$

$$= -\frac{1}{z-\lambda}\frac{1}{2J^2} \times \begin{cases} (z-\lambda) & \text{for } z = 2J \text{ (II-III)}, \\ \left(z - \sqrt{z^2-(2J)^2} - \lambda\right) & \text{for } z \geq 2J \text{ (I-IV)}. \end{cases} \quad (141)$$

We can rewrite $\phi_R$, for $\lambda \neq \lambda_N$, with the help of Eq. (112)

$$\langle I(\lambda)\rangle_{\text{GGE}} = \frac{2T(\lambda)}{z-\lambda}\left[1 - \fint d\lambda' \frac{\rho(\lambda')T(\lambda')}{\lambda-\lambda'}\right] = \langle I(\lambda)\rangle_{i.c.}, \quad (142)$$

as

$$\phi_R(\lambda) = -\frac{q}{z-\lambda} + \frac{\langle I(\lambda)\rangle_{i.c.}}{2T(\lambda)}. \quad (143)$$

Then,

$$\phi_I^2 + \phi_R^2 = \left[\frac{\pi\rho(\lambda)T(\lambda)}{z-\lambda}\right]^2 + \left[-\frac{q}{z-\lambda} + \frac{\langle I(\lambda)\rangle_{i.c.}}{2T(\lambda)}\right]^2$$

$$= \frac{1}{(z-\lambda)^2}\left\{[\pi\rho(\lambda)T(\lambda)]^2 + \left[-q + \frac{\langle I(\lambda)\rangle_{i.c.}(z-\lambda)}{2T(\lambda)}\right]^2\right\}. \quad (144)$$

We also have

$$\phi_R h_I + \phi_I h_R =$$
$$= \left[-\frac{q}{z-\lambda} + \frac{\langle I(\lambda)\rangle_{i.c.}}{2T(\lambda)}\right]\frac{\pi\rho(\lambda)}{2(z-\lambda)} + \frac{\pi\rho(\lambda)T(\lambda)}{2(z-\lambda)^2}\left[\frac{z-\lambda-\sqrt{z^2-(2J)^2}}{2J^2}\right], \quad (145)$$

which reads, in a slightly more compact form,

$$\phi_R h_I + \phi_I h_R =$$
$$= \frac{\pi\rho(\lambda)}{2(z-\lambda)^2}\left[-q + \frac{\langle I(\lambda)\rangle_{i.c.}(z-\lambda)}{2T(\lambda)} + \frac{T(\lambda)(z-\lambda)}{2J^2} - \frac{T(\lambda)\sqrt{z^2-(2J)^2}}{2J^2}\right]. \quad (146)$$

### 5.5.2 Special cases

*Extended phases I and II*

In cases with no condensation (phases I and II) we do not have to worry about the function $K$ since it vanishes. Moreover, $q = 0$. Equation (139) simplifies to

$$(\phi_R^2 + \phi_I^2)\gamma(\lambda) = \frac{(z-\lambda)}{\pi\rho(\lambda)T(\lambda)}(\phi_R h_I + \phi_I h_R), \quad (147)$$

and replacing $(\phi_R^2 + \phi_I^2)$ and $(\phi_R h_I + \phi_I h_R)$ using Eqs. (144) and (146), respectively, we find the following expression for $\gamma(\lambda)$:

$$\gamma(\lambda) = (z-\lambda)\frac{J^2\langle I(\lambda)\rangle_{i.c.}(z-\lambda) + T^2(\lambda)[(z-\lambda) - \sqrt{z^2-(2J)^2}]}{[J\langle I(\lambda)\rangle_{i.c.}(z-\lambda)]^2 + [2\pi J\rho(\lambda)T^2(\lambda)]^2}, \quad (148)$$

which holds for all $\lambda$.

*Equilibrium in phase I*

In equilibrium in phase I, at $T_0 \geq J_0 = J$, $z = T_0 + J_0^2/T_0$, $T(\lambda) = T_0$, and

$$\langle I(\lambda) \rangle_{i.c.} = \frac{T_0^2}{J_0^2} \frac{\frac{2J_0^2}{T_0} - \lambda}{\frac{J_0^2}{T_0} + T_0 - \lambda} = \frac{T_0^2}{J_0^2} \frac{\frac{2J_0^2}{T_0} - \lambda}{z - \lambda}. \tag{149}$$

Replacing in Eq. (148) one finds

$$\gamma(\lambda) = \frac{J_0^2}{T_0^2} - \frac{\lambda}{2T_0}. \tag{150}$$

The constant should be irrelevant and the $\lambda$ dependence is the correct one.

*On the special curve in phase II*

On the special curve $T_0 = (J_0 J)^{1/2}$ in phase II, $\langle I(\lambda) \rangle_{i.c.}^2 = 1$ for all $\lambda$, all temperatures are equal, $T(\lambda) = J$, and $z = 2J$. Equation (148) yields

$$\gamma(\lambda) = 1 - \frac{\lambda}{2J}. \tag{151}$$

*Equilibrium in phase III*

In equilibrium in phase III, $T_0 < J_0 = J$, $z = 2J_0$, $T(\lambda) = T_0$, $q = 1 - T_0/J_0$,

$$\langle I(\lambda) \rangle_{i.c.} = \begin{cases} \dfrac{T_0^2}{J_0^2} \dfrac{\frac{2J_0^2}{T_0} - \lambda}{2J_0 - \lambda} = \dfrac{T_0^2}{J_0^2} \dfrac{\frac{2J_0^2}{T_0} - \lambda}{z - \lambda} & \text{for } \lambda < 2J_0, \\[4mm] \left(1 - \dfrac{T_0}{J_0}\right)^2 N & \text{for } \lambda = 2J_0, \end{cases} \tag{152}$$

and

$$\gamma(\lambda) = c - \frac{\lambda}{2T_0} \qquad \text{for all } \lambda, \tag{153}$$

with $c$ an arbitrary constant. We can then check the validity of Eqs. (82) with the $N$th mode contribution explicitly separated,

$$2T_0 \!\!\!\fint d\lambda' \, \rho(\lambda') \frac{T_0}{z - \lambda'} \frac{\gamma(\lambda') - \gamma(\lambda)}{\lambda - \lambda'} + 2T_0 q \frac{\gamma_N - \gamma(\lambda)}{\lambda - \lambda_N}$$
$$= 2T_0 \!\!\!\fint d\lambda' \, \rho(\lambda') \frac{T_0}{(z - \lambda')} \frac{1}{2T_0} + 2T_0 q \frac{1}{2T_0} = 1,$$

which is just the spherical constraint.

We can now try to check the generic form in equilibrium. Under such conditions we can replace $T(\lambda) = T_0$, $z = 2J_0$, $J = J_0$ and evaluate $\phi_R$ and $\phi_I$

$$\phi_R^2 = \left(\frac{T_0}{2J_0^2}\right)^2, \qquad \phi_I^2 = \frac{T_0^2}{4J_0^4}\left(\frac{2J_0 + \lambda}{2J_0 - \lambda}\right). \tag{154}$$

On the other hand,

$$(\phi_R h_I + \phi_I h_R)\frac{(z-\lambda)}{\pi\rho(\lambda)} = \frac{T_0}{2J_0^2} \,. \tag{155}$$

Going back to Eq. (139) and replacing $\gamma(\lambda) = c - \lambda/(2T_0)$ for all $\lambda$ including $\lambda_N$, we get

$$T_0 \frac{T_0^2}{J_0^3} \frac{1}{2J_0 - \lambda}\left(c - \frac{\lambda}{2T_0}\right) + 2T_0 \frac{T_0}{2J_0^2}\left(1 - \frac{T_0}{J_0}\right)\frac{1}{2T_0} = \frac{T_0}{2J_0^2} \,. \tag{156}$$

We see that the linear terms in $\lambda$ cancel while the constant term then fixes $c$ to $c = J_0/T_0$. For $J_0 = T_0$, parameters for which we join equilibrium in phase I and the beginning of the special curve in phase II, $c = 1$, consistently with the results found above.

*On the special curve in phase IV*

Here,

$$\phi_R = \frac{1}{2J} \quad \text{and} \quad \phi_I = \frac{\pi\rho(\lambda)J}{2J - \lambda} \quad \implies \quad \phi_R^2 + \phi_I^2 = \frac{1}{J}\frac{1}{2J - \lambda} \,,$$
$$(\phi_R h_I + \phi_I h_R)\frac{(z-\lambda)}{\pi\rho(\lambda)} = \frac{1}{4J}\left[1 + \frac{z - \sqrt{z^2 - (2J)^2} - \lambda}{2J - \lambda}\right] , \tag{157}$$

so that Eq. (139) yields

$$\gamma(\lambda) = \frac{1}{4J}\frac{2J - \lambda}{z - \lambda}\left[2J - \lambda + z - \lambda - \sqrt{z^2 - (2J)^2}\right] , \tag{158}$$

which, after replacing $z$ with the parameters on the special line, becomes

$$\gamma(\lambda) = \frac{1}{2J}(2J - \lambda)\frac{J + \sqrt{J_0 J} - \lambda}{\sqrt{\frac{J}{J_0}(J_0 + J) - \lambda}} \,. \tag{159}$$

We note that $\gamma(\lambda \to 2J) = 0$.

# 6 Comparison between static and dynamic results

We now present a thorough comparison between the GGE predictions and the dynamic behaviour. We focus on the square coordinates $s_\mu^2$, the square momenta $p_\mu^2$ which are equivalent to the temperatures $T_\mu$, and the GGE Lagrange multipliers $\gamma_\mu$. Concerning the static calculations, we either work with finite $N$ or in the infinite $N$ limit. In the former case, we solve Eqs. (62) together with the saddle-point Eqs. (63) and the spherical constraint, without making any assumption on the form of the solution. Next, we apply the harmonic *Ansatz* that allows us to take the $N \to \infty$ limit. We then either use Eqs. (75) to determine the temperature spectrum $T(\lambda)$ numerically or we simply use the analytic $T(\lambda)$ derived in Secs. 5.3 and 5.4 – consistently, they are indistinguishable. With this spectrum, we then construct the GGE averaged $s_\mu^2$. We present data for the Lagrange multipliers $\gamma_\mu$ for finite systems and we compare them to the analytic expressions derived in Sec. 5.5. Finally, we use the method sketched in Sec. 6.1 to derive the dynamic results. The comparison of both sets of results, besides allowing us to test the GGE hypothesis, Eq. (5), will provide more information on the four phases in the phase diagram. (In this Section we set $m = 1$.)

## 6.1 Mode dynamics for finite $N$ systems

The dynamics of the mode averages $\langle s_\mu^2(t)\rangle_{i.c.}$ and $\langle \dot{s}_\mu^2(t)\rangle_{i.c.}$ can be solved conveniently using an approach described in detail in Ref. [10]. In this section we introduce the method briefly and highlight some subtle points in its implementation.

The method is based on an amplitude-phase *Ansatz* [84–87] for the mode trajectories,

$$s_\mu(t) = s_\mu(0)\sqrt{\frac{\Omega_\mu(0)}{\Omega_\mu(t)}} \cos \int_0^t dt'\, \Omega_\mu(t') + \frac{\dot{s}_\mu(0)}{\sqrt{\Omega_\mu(t)\Omega_\mu(0)}} \sin \int_0^t dt'\, \Omega_\mu(t'). \tag{160}$$

The main ingredient of this formulation is the function $\Omega_\mu(t)$, which depends on the mode $\mu$ but not on the initial conditions for $s_\mu$ and $\dot{s}_\mu$. This is very convenient, because the averages over initial conditions pass through $\Omega_\mu(t)$ and only act over factors containing $s_\mu(0)$ and $\dot{s}_\mu(0)$. The auxiliary function $\Omega_\mu(t)$ satisfies the equation,

$$\frac{1}{2}\frac{\ddot{\Omega}_\mu(t)}{\Omega_\mu(t)} - \frac{3}{4}\left(\frac{\dot{\Omega}_\mu(t)}{\Omega_\mu(t)}\right)^2 + \Omega_\mu^2(t) = \omega_\mu^2(t) = \frac{(z(t)-\lambda_\mu)}{m}. \tag{161}$$

Of course, initial conditions $\Omega_\mu(0)$ and $\dot{\Omega}_\mu(0)$ should be supplied. However, it turns out that the initial conditions for $\Omega_\mu$ can be arbitrarily chosen, i.e., any real initial condition for $\Omega_\mu$ will generate exactly the same dynamics for the physically relevant observables related to $s_\mu$ and $p_\mu$. In other words, even if the time-dependence of $\Omega_\mu(t)$ is modified by choosing different initial conditions, the dynamics of the physical observables, which typically depend on combinations of the form $\Omega_\mu(0)/\Omega_\mu(t) \cos^2\left[\int_0^t dt'\, \Omega_\mu(t')\right]$, are independent of the initial conditions chosen for $\Omega_\mu(t)$.

As we mentioned earlier, for the same quench, i.e., the same values of $T_0$, $J_0$ and $J$, symmetric and symmetry broken initial conditions produce the same averages for phase-space functions which are quadratic in $\{s_\mu, p_\mu\}$. Given that $z(t)$ depends only on quadratic averages, this means that $z(t)$ and consequently $\Omega_\mu(t)$ are the same for both sets of initial conditions.

The question remains about the dynamics of $\langle s_N(t)\rangle_{i.c.}$ for symmetry broken initial conditions. Coming back to Eq. (160) we get,

$$\langle s_N(t)\rangle_{i.c.} = \langle s_N(0)\rangle_{i.c.}\sqrt{\frac{\Omega_N(0)}{\Omega_N(t)}} \cos \int_0^t dt'\, \Omega_N(t'), \tag{162}$$

where, for symmetry broken initial conditions, $\langle s_N(0)\rangle_{i.c.} = \bar{s}_N = \mathcal{O}(N^{1/2})$, see Sec. 3.1.2. On the other hand,

$$\langle s_N^2(t)\rangle_{i.c.} = \frac{\langle s_N^2(0)\rangle_{i.c.}\,\Omega_N(0)}{\Omega_N(t)} \cos^2 \int_0^t dt'\, \Omega_N(t')$$
$$+ \frac{\langle \dot{s}_N^2(0)\rangle_{i.c.}}{\Omega_N(0)\Omega_N(t)} \sin^2 \int_0^t dt'\, \Omega_N(t'). \tag{163}$$

Given that $\langle s_N^2(0)\rangle_{i.c.} = \bar{s}_N^2 + \sigma_N^2$, where $\sigma_N = \mathcal{O}(1)$, and the fact that $\langle \dot{s}_N^2(0)\rangle_{i.c.} = \mathcal{O}(1)$, we conclude that,

$$\langle s_N^2(t)\rangle_{i.c.} = \langle s_N(t)\rangle_{i.c.}^2 + \mathcal{O}(1), \tag{164}$$

which implies that both averages coincide in the large $N$ limit. In conclusion, the phase-amplitude *Ansatz* is able to accommodate the symmetry broken situation in which $\langle s_N(t)\rangle_{i.c.}$ acquires a non-vanishing and extensive value.

## 6.2 Check of the harmonic *Ansatz*

We start with two checks of the harmonic *Ansatz*. The first one tests its accuracy within the dynamic formalism. The second one confronts the GGE predictions to the exact asymptotic steady state parameter dependence of the (time-averaged) kinetic energy.

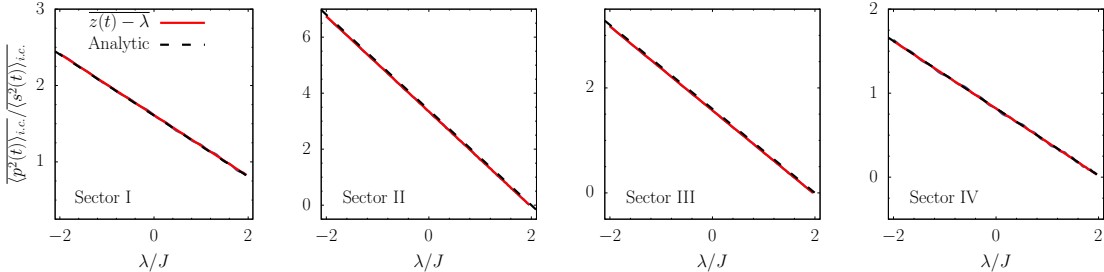

Figure 9: **Test of the harmonic *Ansatz* with dynamic data.** $N = 1024$ system with parameters in the four phases of the phase diagram: in phase I, $T_0 = 1.5 J_0$ and $J = 0.4 J_0$; in phase II, $T_0 = 1.5 J_0$ and $J = 1.7 J_0$; in phase III, $T_0 = 0.5 J_0$ and $J = 0.8 J_0$; and finally in phase IV, $T_0 = 0.5 J_0$ and $J = 0.4 J_0$. $\lambda$ are the post-quench harmonic constants and they are normalised by $J$ in the horizontal axes letting them all vary between $-2$ and $2$. The dashed black lines are the analytic predictions $z_f - \lambda = T_0 + J^2/T_0 - \lambda$ (I and IV) and $z_f - \lambda = 2J - \lambda$ (II and III), while the red lines are the numerical solution to the dynamic equations.

The most direct test of the harmonic *Ansatz* we could think of is to compare $\overline{\langle p_\mu^2(t) \rangle}_{i.c.} / \overline{\langle s_\mu^2(t) \rangle}_{i.c.}$ to $\overline{z(t) - \lambda_\mu}$, all computed with the dynamic formalism. Within the harmonic hypothesis, these two quantities should be equal. We plot them for parameters in the four phases of the phase diagram in Fig. 9. The agreement is perfect in all phases. We also compare with the analytic prediction for $\overline{z(t) - \lambda_\mu}$, given by $z_f - \lambda_\mu$ with $z_f = 2J$ in phases II and III, and $z_f = T_0 + J^2/T_0$ in phases I and IV.

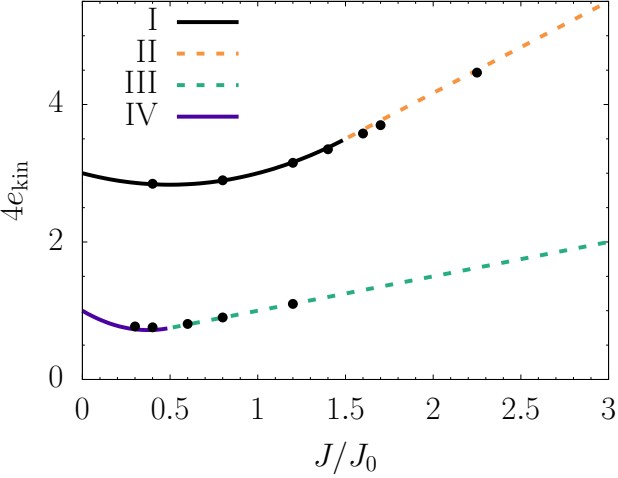

Figure 10: **Test of the harmonic *Ansatz* within the GGE calculation.** The kinetic energy density as a function of $J/J_0$ for two values of the initial temperature $T_0/J_0$ of the initial conditions. The black dots are numerical results obtained with the harmonic *Ansatz* for the evaluation of the GGE and the curves represent the exact values for $T_0/J_0 = 1.5$ (above) and $T_0/J_0 = 0.5$ (below).

The second test of the accuracy of the harmonic *Ansatz* consists in comparing the parameter dependence of the kinetic energy density that it predicts, to the exact one. This is represented in Fig. 10, where the black dots are the numerical evaluation of $\langle e_{kin}\rangle_{GGE}$ in a system with $N = 100$ and the solid curves represent the exact values recalled in Table 1 evaluated at $T_0/J_0 = 1.5$ (above) and $T_0/J_0 = 0.5$ (below). There is perfect agreement.

## 6.3 GGE and dynamic averages

### 6.3.1 Phase I

In the whole phase I, both with energy injection or extraction, the asymptotic $z_f$ is larger than $\lambda_N$, there is no condensation of modes, and the constants of motion are all $\mathcal{O}(1)$ including the $N$-th one.

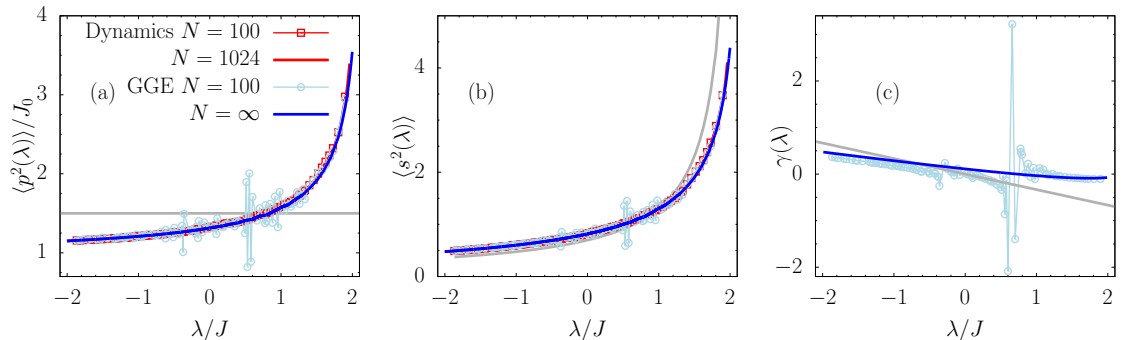

Figure 11: **Comparison of the dynamic and GGE results in phase I with energy injection,** $T_0 = 1.5J_0$ **and** $J = 0.4J_0$**.** (a) The averaged $p^2(\lambda)$, (b) the averaged $s^2(\lambda)$ and (c) the Lagrange multipliers $\gamma(\lambda)$. Finite size dynamic data for $N = 100$ and $N = 1024$ compared to the GGE analytic results. Here and in all following plots we measure $p^2$ in units of $J_0$, which is set to 1 as well as $m$. All grey curves represent the pre-quench equilibrium values; in (a) they are the initial temperature, $T_0/J_0 = 1.5$, in (c) $\gamma_\mu = -\beta'\lambda_\mu^{(0)}/2 = -J_0\lambda_\mu/(2T_0J) = -0.33\lambda_\mu/J$. All quantities are finite at the edge of the spectrum.

In Fig. 11 we show numerical results for parameters such that there is energy injection in this phase. We compare the solution of the GGE equations for $\langle s_\mu^2\rangle_{GGE}$ and $\langle p_\mu^2\rangle_{GGE}$ for finite $N$, the analytic expressions for infinite $N$, and the numerical integration of the mode equations for finite $N$. The GGE results for finite $N$ show some oscillations in the middle of the spectrum which can be ascribed to the finite system size. In panel (c) we plot the spectrum of $\gamma_\mu$ for the $N = 100$ system. The outlier data points in the middle of the spectrum could well be finite size effects, since they correspond to the same modes for which the $\langle s_\mu^2\rangle_{GGE}$ and $\langle p_\mu^2\rangle_{GGE}$ deviate from their more regular trend. Ignoring these points, the rest of the data display a rather linear dependence on the mode index, though the slope is different from the equilibrium one at the pre-quench parameters, which is plotted with a grey inclined thin line. Having said this, the $N \to \infty$ behaviour of $\gamma_\mu$ is not completely linear. (As a side comment, in this phase the instantaneous steady state approximation introduced in [10] was very accurate, see Fig. 13 in this reference.)

Parameters with energy extraction in this phase lead to equivalent perfect agreement between GGE and dynamics. The only difference is the bending downwards of the temperature spectrum close to the right edge.

### 6.3.2 Phase II

In Fig. 12 we show numerical results in phase II. In general, the dynamic behaviour is in very good agreement with the GGE predictions both on the special line and away from it. The accord deteriorates a bit when moving far away from the transition. This is linked to the fact that the integration of the dynamic equations in cases in which $z = \lambda_N$ is hard close to the edge of the spectrum. We give more details on the reason for this when treating cases in phase III, which suffer from the same problems.

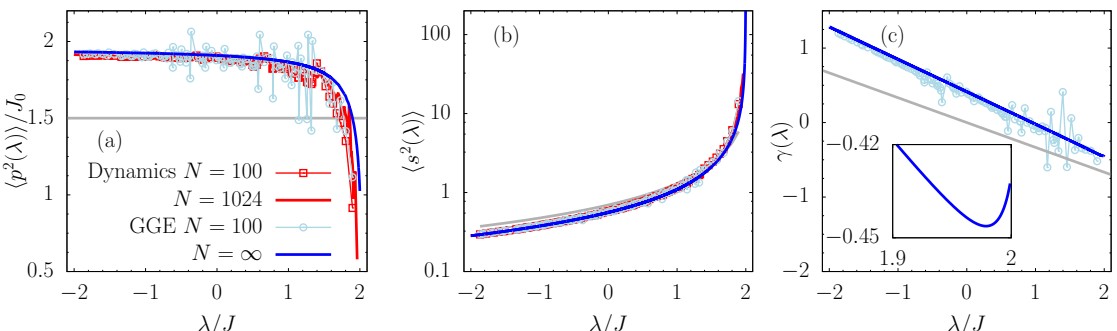

Figure 12: **Comparison of the dynamic and GGE results in phase II.** The quench parameters are $T_0 = 1.5 J_0$ and $J = 1.7 J_0$. (a) $\langle p^2(\lambda) \rangle$, (b) $\langle s^2(\lambda) \rangle$ and (c) GGE Lagrange multipliers $\gamma(\lambda)$. The $N \to \infty$ GGE results can be confronted to the dynamic ones with finite $N$. Note the fast grow of $\langle s^2 \rangle_{\mathrm{GGE}}$ at the edge of the spectrum in (b) accompanied by a vanishing $\langle p^2 \rangle_{\mathrm{GGE}}$ in (a). The grey horizontal line in (a) is at the initial temperature $T_0 = 1.5 J_0$, the grey curve in (b) is the initial average $\langle s_\mu^2(0^+) \rangle_{i.c.}$ and in (c) the grey straight line represents the pre-quench equilibrium values $\gamma(\lambda) = -\beta' \lambda^{(0)}/2 = -J_0 \lambda/(2 T_0 J) = -0.333 \lambda/J$. The zoom highlights the fact that $\gamma$ deviates from the straight line close to the edge of the spectrum.

### 6.3.3 Phase III

In Fig. 13 we show numerical results in phase III, with parameters such that there is energy injection. The static data are in very good agreement with the dynamic ones in the bulk of the spectrum but there are significant deviations at the edge. In fact, the solution of the dynamic mode equations gets tricky for $\lambda$ close to $\lambda_N$ in phases in which $z_f = \lim_{N \to \infty} \lambda_N$ (II and III). More concretely, at finite $N$ the last mode cannot be considered to be in a stationary state, and to take numerically $N \to \infty$ together with the corresponding large time limit is impossible.

In Fig. 14 we show the magnitude of the relative temporal fluctuations, as quantified with the dispersion from the mean, as a function of $\mu$ for all phases in the phase diagram. It is clear that in phases II and III the fluctuations are large and the modes near the edge of the spectrum are not yet stationary.

### 6.3.4 Phase IV

In Fig. 15 we display numerical results for parameters in phase IV, The agreement between dynamic and GGE averages is extremely good. Note the divergencies of $\langle s^2 \rangle$ and $\langle p^2 \rangle$ at the edge of the spectrum singled out and discussed in the analytic Section, which are not proportional to system size though.

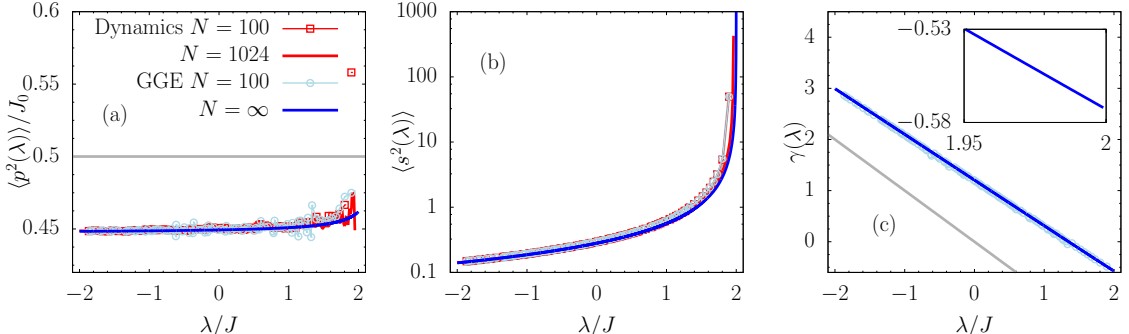

Figure 13: **Comparison of the dynamic and GGE results in phase III, with energy injection.** The quench parameters are $T_0 = 0.5 J_0$ and $J = 0.8 J_0$. (a) $\langle p^2(\lambda) \rangle$, (b) $\langle s^2(\lambda) \rangle$ and (c) GGE Lagrange multipliers $\gamma(\lambda)$. Note the divergence of $\langle s^2(\lambda) \rangle_{\text{GGE}}$ at the edge of the spectrum and the finite value that $\langle p_N^2 \rangle_{\text{GGE}}$ takes for $N \to \infty$. The reason for the deviating dynamic point in (a) is discussed in the text. The grey curves represent the initial values. The inset in (c) highlights the behaviour of $\gamma(\lambda)$ close to the edge.

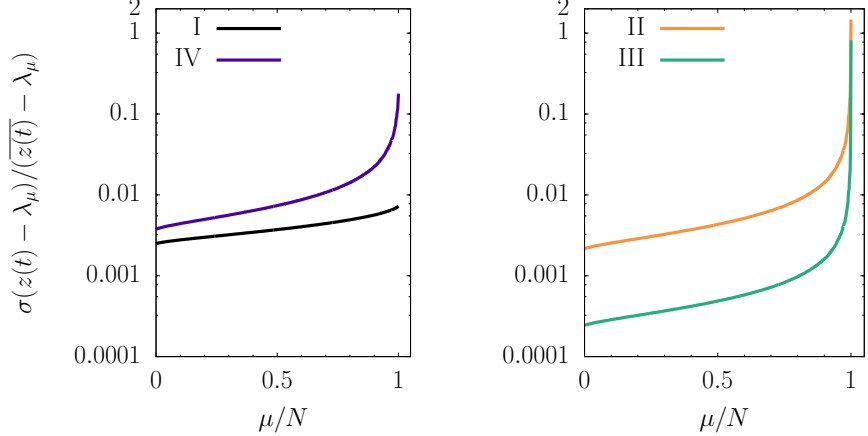

Figure 14: **Temporal fluctuations of the mode frequencies in all phases of the phase diagram** in a system with $N = 1024$. The temperature of the initial conditions are $T_0 = 1.5 J_0$ in phases I and II and $T_0 = 0.5 J_0$ in phases III and IV. The post-quench interaction is $J = 0.4 J_0$ (phase I), $J = 1.7 J_0$ (phase II), $J = 1.2 J_0$ (phase III), and $J = 0.5 J_0$ (phase IV).

## 6.4 Balance of mode energies

For $J/J_0 < 1$ the system gets energy from the quench while for $J/J_0 > 1$ it releases energy. Right after the instantaneous quench, each positive (negative) mode receives (releases) energy for $J/J_0 < 1$, and does the opposite for $J/J_0 > 1$, see Sec. 3.2. Although in the further evolution the energy of the modes are not individually conserved, the initial heating or cooling of the edge modes is maintained in the quenches we showed. In Fig. 11 in phase I, Fig. 13 in phase III, and Fig. 15 in phase IV, results for quenches with energy injection are studied, and the right end modes get hotter. On the contrary, with global energy extraction, one heats the negative modes close to the left edge, and cools the positive ones close to the right edge, see e.g. Fig. 12 in phase II.

A direct consequence of the validity of the harmonic *Ansatz* is that, asymptotically, each

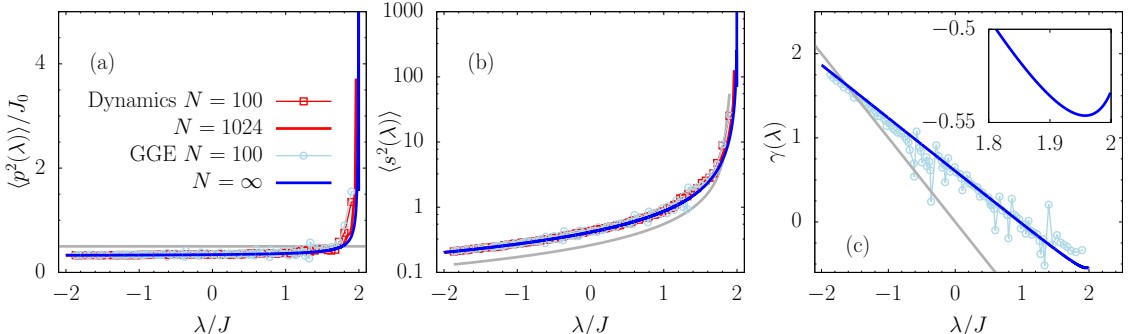

Figure 15: **Comparison of the dynamic and GGE results in phase IV, with energy injection.** The quench parameters are $T_0 = 0.5J_0$ and $J = 0.4J_0$. (a) $\langle p^2(\lambda) \rangle$, (b) $\langle s^2(\lambda) \rangle$ and (c) GGE Lagrange multipliers $\gamma(\lambda)$. The grey curves are the initial profiles.

mode should satisfy energy equipartition, with a modified spring constant $z_f - \lambda_\mu$,

$$\frac{1}{m} \overline{\langle p_\mu^2 \rangle}_{i.c.} = (z_f - \lambda_\mu) \overline{\langle s_\mu^2 \rangle}_{i.c.} = T_\mu, \tag{165}$$

and its total energy be constant

$$\overline{\langle e_\mu \rangle}_{i.c.} = 2T_\mu, \tag{166}$$

and equal to twice the mode temperature. The spectra of $T_\mu$ which we derive analytically in this paper comply with these relations.

# 7 Fluctuations

In this Section we focus on the analytic study of fluctuations within the dynamic and GGE approaches. As mentioned in Sec. 2.3 the fluctuations of the constraints with respect to the initial conditions decide whether our model is equivalent to the Neumann one in the large $N$ limit. In the statistical realm, the fluctuations of the constraints calculated with the GGE measure determine the equivalency between canonical (with strict spherical constraints) and grand-canonical (constraints on average as in Eq. (45)) formulations of the GGE. We will show that in both dynamical and statistical calculations there are no relevant fluctuations in phases I, II and IV while there are in phase III making the equivalence of the NM and SNM models arguable for parameters in this part of the phase diagram.

## 7.1 Symmetric initial conditions

In Sec. 3.1.1 we introduced initial conditions which are in equilibrium at low and high temperature with respect to the canonical phase diagram and do not break any symmetry. At low temperatures, in this kind of initial configurations, the fluctuations of $s_N$ scale with $N$ in a way that ensures the spherical constraint on average, but the average of this mode is still zero. In this Section, we first evaluate the fluctuations of the primary and secondary constraints $\phi$ and $\phi'$ along the trajectories generated by these initial conditions. We also use the corresponding GGE measure to evaluate the fluctuations of $\phi$ and $\phi'$. Then we compare.

### 7.1.1 Fluctuations in the dynamics

Our objective in this section is to address the scaling of the fluctuations of the primary and secondary constraint *at all times*. To do it, we check the scaling of the fluctuations at the initial

time (initial conditions) and in the asymptotic state (long time averages). These two checkpoints will give us a complete picture about the dynamics of the fluctuations of the constraints.

Before we proceed to the main analysis, it would be useful to recall the scaling of the averages $\langle s_N^2(t)\rangle_{i.c.}$ and $\langle p_N^2(t)\rangle_{i.c.}$ in the different phases of the parameter space.

We start with the scaling in the initial conditions, i.e., at $t=0$. In phases I and II, $T_0 > J_0$, the initial conditions are not condensed, hence $\langle s_N^2(0)\rangle_{i.c.}$ and $\langle p_N^2(0)\rangle_{i.c.}$ do not scale with $N$. In phases III and IV, $T_0 < J_0$, the initial conditions are condensed, i.e., $\langle s_N^2(0)\rangle_{i.c.}$ scales linearly with $N$ [73, 74].

On the other hand, as was established in the previous section, the scaling of the *long time averages* of these quantities is the same as the corresponding statistical averages in the GGE. In phases I, II and IV there is no condensation. In particular, there is no $N$ dependent scaling in phase I while, from the numerics we see sublinear scaling of $\overline{\langle s_N^2\rangle}_{i.c.}$ in phase II and of both $\overline{\langle s_N^2\rangle}_{i.c.}$ and $\overline{\langle p_N^2\rangle}_{i.c.}$ in phase IV (for reference, the corresponding finite $N$ GGE averages are shown in Figs. 6, 7 and 8).) In phase III the long time average of $\langle s_N^2(t)\rangle_{i.c.}$ scales linearly with $N$, which indicates condensation.

As an interesting example, in Fig. 16 we show the scaling of the long time averages for a point in sector IV. Even though the initial conditions are condensed, the scaling of the long-time average of $\langle s_N^2(t)\rangle_{i.c.}$ is clearly sublinear, which is compatible with the predictions of the GGE, see Sec. 5.4.2. We can also observe that the long time average of $\langle p_N^2(t)\rangle_{i.c.}$ develops a sublinear but non-trivial scaling with $N$ that was not present in its initial conditions. In a similar fashion, in sector II, $\langle s_N^2(t)\rangle_{i.c.}$ picks up a sublinear but non-trivial scaling even if the initial conditions show no scaling with $N$ (not shown).

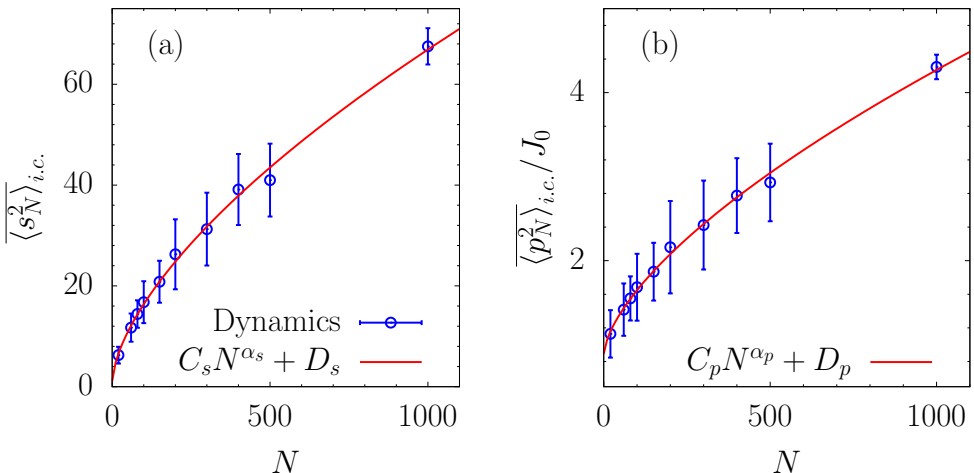

Figure 16: **Scaling of the averages at the edge of the spectrum for a phase IV point**, $T_0 = 0.8J_0$ and $J = 0.6J_0$. The exponents of the fits are $\alpha_s = 0.63$ and $\alpha_p = 0.65$.

We first calculate the fluctuations of $\phi = \sum_\mu s_\mu^2(t) - N$. For symmetric initial conditions, $\langle s_\mu(0)\rangle_{i.c.} = \langle p_\mu(0)\rangle_{i.c.} = 0$, $\langle s_\mu(0)s_\nu(0)\rangle_{i.c.} = \delta_{\mu\nu}\langle s_\mu^2(0)\rangle_{i.c.}$, $\langle p_\mu(0)p_\nu(0)\rangle_{i.c.} = \delta_{\mu\nu}\langle p_\mu^2(0)\rangle_{i.c.}$, and $\langle s_\mu(0)p_\nu(0)\rangle_{i.c.} = 0$, for all $\mu, \nu$, where $\langle\cdots\rangle_{i.c.}$ denotes an average over initial conditions. Moreover, the dynamics are given by the phase-amplitude *Ansatz*, see Sec. 6.1:

$$s_\mu(t) = s_\mu(0)a_\mu(t) + \dot{s}_\mu(0)b_\mu(t), \tag{167}$$

where, in order to ease the notation, we defined

$$a_\mu(t) \equiv \sqrt{\frac{\Omega_\mu(0)}{\Omega_\mu(t)}} \cos \int_0^t dt' \, \Omega_\mu(t'), \quad b_\mu(t) \equiv \frac{1}{\sqrt{\Omega_\mu(t)\Omega_\mu(0)}} \sin \int_0^t dt' \, \Omega_\mu(t').$$

We now have to calculate higher order averages involving products of time-dependent four phase space variables, that is averages of the kind $\langle s_\mu^2(0) s_\nu^2(0) \rangle_{i.c.}$. In order to do it, we exploit that the initial distribution is Gaussian for all modes, including the last one, even in cases in which $s_N$ condenses. Then, according to Isserli's theorem,

$$\langle s_\mu^2(0) s_\nu^2(0) \rangle_{i.c.} = (1 - \delta_{\mu\nu}) \langle s_\mu^2(0) \rangle_{i.c.} \langle s_\nu^2(0) \rangle_{i.c.} + 3\delta_{\mu\nu} \langle s_\mu^2(0) \rangle_{i.c.}^2. \tag{168}$$

Similar relations apply to averages involving $p_\mu(0)$. Putting all these identities together and after some simple algebra we obtain

$$\left\langle \left( \sum_\mu s_\mu^2(t) \right) \left( \sum_\nu s_\nu^2(t) \right) \right\rangle_{i.c.} - N^2 = 2 \sum_\mu \langle s_\mu^2(0) \rangle_{i.c.}^2 a_\mu^4(t)$$

$$+ 4 \sum_\mu \langle s_\mu^2(0) \rangle_{i.c.} \langle \dot{s}_\mu^2(0) \rangle_{i.c.} a_\mu^2(t) b_\mu^2(t) + 2 \sum_\mu \langle \dot{s}_\mu^2(0) \rangle_{i.c.}^2 b_\mu^4(t). \tag{169}$$

Using the form of the dynamical *Ansatz* in Eq. (167) and the mean values in Eq. (168)

$$\frac{1}{N^2} \left\langle \left( \sum_\mu s_\mu^2(t) \right) \left( \sum_\nu s_\nu^2(t) \right) \right\rangle_{i.c.} - 1 = \frac{2}{N^2} \sum_\mu \langle s_\mu^2(t) \rangle_{i.c.}^2. \tag{170}$$

This expression is valid at all times and even for finite $N$. Given this expression, we have three different scenarios. If all averages $\langle s_\mu^2(t) \rangle_{i.c.}$ are $\mathcal{O}(1)$ then $\sum_\mu \langle s_\mu^2(t) \rangle_{i.c.}^2$ is $\mathcal{O}(N)$ and the variance vanishes as $N^{-1}$ in the large $N$ limit. Instead, if there is condensation $\langle s_N^2(t) \rangle_{i.c.} = \mathcal{O}(N)$, which implies $\langle s_N^2(t) \rangle_{i.c.}^2 = \mathcal{O}(N^2)$, and the variance remains $\mathcal{O}(1)$ even in the large $N$ limit. An intermediate case appears whenever we have sublinear scaling of $\langle s_N^2(t) \rangle_{i.c.}$ with a power $\alpha_N < 1$. In such case the fluctuations vanish, but eventually slower than $N^{-1}$.

Bearing this in mind, we see that the fluctuations of $\phi$ vanish in the large $N$ limit in phases I and II, both for the initial conditions and in the long-time limit. In phases III and IV, the initial conditions are condensed, and that implies that the fluctuations of $\phi$ do not vanish in the large $N$ limit. However, the situation can be easily corrected if, instead of symmetric initial conditions, we use symmetry broken ones, see Sec. 3.1.2. In such case, the fluctuations of $\phi$ are well behaved both for phase III and IV, see Sec. 7.2.1. Regarding the long times limit in the initially condensed phases, the fluctuations of $\phi$ remain condensed in phase III and vanish in phase IV. Notice that we are estimating the scaling of the average $\overline{\langle s_N^2(t) \rangle_{i.c.}^2}$ using the known scalings of $\overline{\langle s_N^2(t) \rangle_{i.c.}}$. We have numerically checked that such estimation is correct.

Next we study the variance of the secondary constraint $\phi' = \sum_\mu s_\mu p_\mu$. To perform the calculation we recall that

$$\dot{s}_\mu(t) = s_\mu(0) c_\mu(t) + \dot{s}_\mu(0) d_\mu(t), \tag{171}$$

with

$$c_\mu(t) \equiv -\frac{1}{2} \sqrt{\frac{\Omega_\mu(0)}{\Omega_\mu(t)} \frac{\dot{\Omega}_\mu(t)}{\Omega_\mu(t)}} \cos \int_0^t dt' \, \Omega_\mu(t') - \sqrt{\Omega_\mu(0)\Omega_\mu(t)} \sin \int_0^t dt' \, \Omega_\mu(t'),$$

$$d_\mu(t) \equiv \sqrt{\frac{\Omega_\mu(t)}{\Omega_\mu(0)}} \cos \int_0^t dt' \, \Omega_\mu(t') - \frac{1}{2} \frac{1}{\sqrt{\Omega_\mu(0)\Omega_\mu(t)}} \frac{\dot{\Omega}_\mu(t)}{\Omega_\mu(t)} \sin \int_0^t dt' \, \Omega_\mu(t').$$

As for the analysis of the primary constraint, we use Gaussian decouplings to calculate the higher order averages over initial conditions:

$$\frac{1}{N^2}\left\langle\left(\sum_\mu s_\mu(t)p_\mu(t)\right)\left(\sum_\nu s_\nu(t)p_\nu(t)\right)\right\rangle_{i.c.} =$$
$$\frac{1}{N^2}\sum_\mu \langle s_\mu^2(t)\rangle_{i.c.}\langle p_\mu^2(t)\rangle_{i.c.} + \frac{1}{N^2}\sum_\mu \langle s_\mu(t)p_\mu(t)\rangle_{i.c.}^2. \tag{172}$$

The first term can give a non-vanishing contribution in the large $N$ limit only if both $\langle s_N^2(t)\rangle$ and $\langle p_N^2(t)\rangle$ have amplitudes that scale linearly with $N$, which is not verified in any phase. This implies that the first term does not poses any threat to the vanishing of fluctuations of the secondary constraint in the large $N$ limit neither for the initial conditions, nor for the long-time limit. Again, note that we are estimating the scaling of $\overline{\langle s_N^2(t)\rangle\langle p_N^2(t)\rangle}$ using the known scaling of $\overline{\langle s_N^2(t)\rangle}$ and $\overline{\langle p_N^2(t)\rangle}$. In this case we have also checked that the estimation is correct. On the other hand, we have checked that the second term in (172) is similarly innocuous.

These observations have a deep meaning regarding the equivalence between the dynamics under the averaged or the strict spherical constraints. In phases I and II the two are equivalent in the large $N$ limit, since the constraints are fulfilled on average and their variances vanish with increasing $N$. In phases III and IV the primary constraint fails, and the dynamics of the two models are not completely equivalent. However, the scaling of the fluctuations of the primary constraint can be corrected if we chose symmetry broken initial conditions, see Sec. 7.2.1.

Similar conclusions are deduced from the study of the fluctuations in the GGE, which will be analysed in the next Subsection for the same kind of initial conditions.

### 7.1.2 Fluctuations in the Generalised Gibbs Ensemble

In this Section we study the fluctuations of the two constraints in the GGE formalism. For simplicity, we use the discrete $\mu$ form of the GGE action and saddle-point equations. The continuous limit, with $N \to \infty$, can be easily obtained at every step of the calculations. We introduce sources $\mathcal{J}_\mu^{(s^2)}$ and $\mathcal{J}_\mu^{(p^2)}$ coupled to $s_\mu^2$ and $p_\mu^2$, and we thus transform the GGE partition function into a generating functional, from which averages can be readily calculated. For example,

$$-\frac{\partial \ln Z_{GGE}[\mathcal{J}]}{\partial \mathcal{J}_\mu^{(s^2)}}\bigg|_{\mathcal{J}=0} = \langle s_\mu^2\rangle_{GGE}, \qquad -\frac{\partial \ln Z_{GGE}[\mathcal{J}]}{\partial \mathcal{J}_\mu^{(p^2)}}\bigg|_{\mathcal{J}=0} = \langle p_\mu^2\rangle_{GGE},$$
$$\frac{\partial^2 \ln Z_{GGE}[\mathcal{J}]}{\partial \mathcal{J}_\mu^{(s^2)}\mathcal{J}_\nu^{(s^2)}}\bigg|_{\mathcal{J}=0} = \langle s_\mu^2 s_\nu^2\rangle_{GGE} - \langle s_\mu^2\rangle_{GGE}\langle s_\nu^2\rangle_{GGE}. \tag{173}$$

After simple manipulations which involve the saddle-point equations, we find

$$\langle s_\mu^2 s_\nu^2\rangle_{GGE} - \langle s_\mu^2\rangle_{GGE}\langle s_\nu^2\rangle_{GGE} = 2\,\delta_{\mu\nu}\,\langle s_\mu^2\rangle_{GGE}^2,$$
$$\langle p_\mu^2 p_\nu^2\rangle_{GGE} - \langle p_\mu^2\rangle_{GGE}\langle p_\nu^2\rangle_{GGE} = 2\,\delta_{\mu\nu}\,\langle p_\mu^2\rangle_{GGE}^2, \tag{174}$$
$$\langle s_\mu^2 p_\nu^2\rangle_{GGE} = \langle s_\mu^2\rangle_{GGE}\langle p_\nu^2\rangle_{GGE},$$

which correspond to averages over independent Gaussian ensembles with zero mean for all $\mu$.

We can observe that whenever $\langle s_N^2\rangle_{GGE}$ or $\langle p_N^2\rangle_{GGE}$ are order $N$, as in the condensed phase III, the fluctuations of $s_N^2$ and $p_N^2$ are proportional to $N^2$. As already explained in the description of the canonical equilibrium of the spherical Sherrington-Kirkpatrick model, this phenomenon

is known as condensation of fluctuations [73, 74], does not involve symmetry breaking, but has a deep impact on the fluctuations of the constraints.

The fluctuations of the primary constraint are

$$\frac{1}{N^2}\left\langle\left(\sum_\mu s_\mu^2\right)\left(\sum_\nu s_\nu^2\right)\right\rangle_{\text{GGE}} - 1 = \frac{2}{N^2}\sum_\mu \langle s_\mu^2\rangle_{\text{GGE}}^2. \tag{175}$$

In phase I, all the averages $\langle s_\mu^2\rangle_{\text{GGE}}$ are order 1, which means that the fluctuations are order $N^{-1}$. In phase II, all the averages $\langle s_\mu^2\rangle_{\text{GGE}}$ are order 1, except for $\langle s_N^2\rangle_{\text{GGE}}$ which is order $N^\alpha$ with $\alpha \sim 0.5$. This implies, again, that the fluctuations vanish in the large $N$ limit. In phase III, $\langle s_N^2\rangle_{\text{GGE}}$ is order $N$, and the fluctuations do not vanish but turn up to be order 1 in the large $N$ limit. In phase IV, the situation is similar to phase II, all the averages $\langle s_\mu^2\rangle_{\text{GGE}}$ are order 1, except for $\langle s_N^2\rangle_{\text{GGE}}$ which is order $N^a$ with $a < 1$. This implies that the fluctuations vanish in the large $N$ limit.

Regarding the secondary constraint, we have,

$$\frac{1}{N^2}\left\langle\sum_{\mu\nu} s_\mu p_\mu s_\nu p_\nu\right\rangle_{\text{GGE}} = \frac{1}{N^2}\sum_\mu \langle s_\mu^2\rangle_{\text{GGE}}\langle p_\mu^2\rangle_{\text{GGE}}. \tag{176}$$

In phases I, II and III these fluctuations vanish as $N^{-1}$ for large $N$. In phase IV $\langle s_N^2\rangle_{\text{GGE}} \propto N^{\alpha_s}$ and $\langle p_N^2\rangle_{\text{GGE}} \propto N^{\alpha_p}$ with $\alpha_s < 1$ and $\alpha_p < 1$, see Sec. 5.4.2, which implies that the relative fluctuations in the secondary constraint also vanish, but slower than $N^{-1}$.

These observations have an impact on the equivalence of the ensembles defined by imposing the constraints exactly or on average. Whenever the relative fluctuations of both constraints vanish on average, the results obtained with the "spherically averaged" ensemble are completely equivalent to those obtained with the strictly spherical one. We can conclude that in phases I, II and IV both ensembles are equivalent in the $N \to \infty$ limit, whereas in phase III they are not since the fluctuations of the primary constraint do not vanish in such limit. The situation is similar to the one studied by Kac and Thompson [72] but with the difference that in our case we have additional momenta and, consequently, one additional constraint.

As a brief summary, for symmetric initial conditions the two ways of imposing the constraint are equivalent in phases I, II and IV, but they are not in phase III.

The situation in phase III can be fixed if we introduce symmetry breaking in the GGE, which will be done in the next Section.

## 7.2 Symmetry broken initial conditions

In phase 3.1.2 we discussed the initial conditions, at low temperature with respect to the equilibrium phase diagram, that break rotational symmetry by attributing an $N$-dependent value to $\langle s_N\rangle_{i.c.}$. The spherical constraint is also satisfied with this choice. We now evaluate the fluctuations of these configurations in the dynamical and GGE formalisms.

### 7.2.1 Fluctuations in the dynamics

Separating the contribution of the $N$th mode from the terms involving only the bulk variables, and performing the Gaussian averages (with zero mean) over the bulk variables,

$$\left\langle\left(\sum_\mu s_\mu^2(t)\right)\left(\sum_\nu s_\nu^2(t)\right)\right\rangle_{i.c.} = \left(\sum_{\mu(\neq N)}\left\langle s_\mu^2(t)\right\rangle_{i.c.}\right)^2 + 2\sum_{\mu(\neq N)}\left\langle s_\mu^2(t)\right\rangle_{i.c.}^2$$
$$+ 2\left\langle s_N^2(t)\right\rangle_{i.c.}\sum_{\nu(\neq N)}\left\langle s_\nu^2(t)\right\rangle_{i.c.} + \left\langle s_N^4(t)\right\rangle_{i.c.}. \tag{177}$$

In phase III, in the large $N$ limit, the $N$th mode is a non-fluctuating condensate with

$$\left\langle s_N^4(t)\right\rangle_{i.c.} = \left\langle s_N^2(t)\right\rangle_{i.c.}^2 , \tag{178}$$

(note the absence of factor 3 meaning that this is not the consequence of the Wick factorisation but the one of $s_N \propto (qN)^{1/2} + o(N^{1/2})$). Equation (177) simplifies to

$$\frac{1}{N^2}\left\langle \left(\sum_\mu s_\mu^2(t)\right)\left(\sum_\nu s_\nu^2(t)\right)\right\rangle_{i.c.} = \frac{1}{N^2}\left(\sum_\mu \left\langle s_\mu^2(t)\right\rangle_{i.c.}\right)^2 + o(1) = 1 + o(1).$$

In phase IV the $N$th mode, in the long-time limit, scales as $\left\langle s_N^2(t)\right\rangle_{i.c.} \propto N^{\alpha_s}$ with $\alpha_s < 1$. We numerically checked that the second, third and fourth terms in the r.h.s of Eq. (177) are negligible, i.e. they scale slower than $N^2$. Thus the equation for the primary constraint simplifies to

$$\frac{1}{N^2}\left\langle \left(\sum_\mu s_\mu^2(t)\right)\left(\sum_\nu s_\nu^2(t)\right)\right\rangle_{i.c.} = \frac{1}{N^2}\left(\sum_\mu \left\langle s_\mu^2(t)\right\rangle_{i.c.}\right)^2 + o(1) = 1 + o(1).$$

In a nutshell, with symmetry broken initial conditions, in phases III and IV our model verifies the primary constraint as the fluctuations vanish in the thermodynamic limit.

Let us now consider the fluctuations of the secondary constraint. Separating the $N$th mode contribution from the rest of the terms, and using the independent harmonic oscillator *Ansatz* (in the bulk) which implies $\langle s_\mu(t)p_\mu(t)\rangle_{i.c.} = 0$, we find

$$\frac{1}{N^2}\left\langle \left(\sum_\mu s_\mu(t)p_\mu(t)\right)\left(\sum_\nu s_\nu(t)p_\nu(t)\right)\right\rangle_{i.c.} = \frac{1}{N^2}\left\langle s_N^2(t)p_N^2(t)\right\rangle_{i.c.} .$$

For both phases III and IV we have $\left\langle s_N^2(t)p_N^2(t)\right\rangle_{i.c.} = o(N^2)$. Thus the secondary constraint – as well as the first one – is strictly verified in the thermodynamic limit.

These results imply that with the use of symmetry broken initial conditions the relative fluctuations of both the primary and secondary constraint vanish in the thermodynamic limit. This, in turn, implies that, if we use these initial conditions, there is no difference in imposing the constraints on average or exactly for large $N$. We should also recall that the symmetric or symmetry broken initial conditions produce different results only for observables which involve a product of three or more phase-space variables $\{s_\mu, p_\mu\}$, see Sec. 3.1.2. In particular, for the observables considered in this work, averages of quadratic functions of $s_\mu$ and $p_\mu$, both sets of initial conditions give the same results.

### 7.2.2 Fluctuations in the Generalised Gibbs Ensemble

In this Section we develop a formulation of the GGE that includes a symmetry breaking pinning field, by virtue of which the last mode can acquire a non-vanishing average. In parts we use the language of the spin Sherrington-Kirkpatrick model; more precisely, we name the field a magnetic one and "magnetized" means $\langle s_N\rangle_{GGE} = qN$.

The introduction of a magnetic field in the equilibrium partition function breaks the $\mathbb{Z}_2$ symmetry ($s_\mu \to -s_\mu$) and introduces a "magnetised" state as the new thermal equilibrium in the low temperature phase. We follow similar steps in the GGE formulation. We introduce a resolution of identity with a delta function which fixes the average of $s_N$ to a value $m_s$, which can eventually be taken to scale with $N$ or vanish. We express the Dirac delta with the help of its Fourier representation, with an auxiliary (imaginary) field $h_s$ acting on $s_N$ as pinning field:

$$1 \propto \int dm_s\, dh_s\, e^{h_s(m_s - s_N)} . \tag{179}$$

$m_s$ represents $\langle s_N \rangle_{\text{GGE}}$ at the saddle point level.

Two reasons can be evoked to justify this approach. First, in the case in which the spring forces are not rescaled ($J = J_o$) the GGE partition function should give the same results as the equilibrium measure. This means that in the low temperature phase if the $\mathbb{Z}_2$ symmetry is broken in the initial conditions we need to obtain the same symmetry breaking in the GGE measure. However, the GGE partition function in Sec. 4 yields $\langle s_N \rangle_{\text{GGE}} = 0$ for all parameters. Indeed, focusing on Eq. (53) the absence of linear term with respect to $s_N$ prevents the system from getting magnetised. Thus, the auxiliary field $h_s$ should correct this problem. Secondly, in previous papers [10, 11] the dynamics were studied using the Martin-Siggia-Rose generating functional and the Schwinger-Dyson equations derived in the thermodynamic limit taken before switching off the pinning fields. These equations couple self-correlation and linear response defined as

$$R(t, t') = \lim_{\vec{h} \to \vec{0}} \lim_{N \to \infty} \frac{1}{N} \sum_\mu \Big\langle \frac{\partial s_\mu^{(h)}(t)}{\partial h_\mu(t')} \Big\rangle, \tag{180}$$

where the superscript $(h)$ indicates that the trajectory $s_\mu^{(h)}(t)$ is calculated under the field. Therefore, if we want to match the Schwinger-Dyson dynamic results with a GGE calculation, we need to take the same convention for the order of limits, meaning we shall take the thermodynamic limit first.

The full study of the GGE partition function with this extra auxiliary field $h_s$ is similar to the one presented in Sec. 4, and can be found in App. F. Here we just give the relevant definitions and we stress some key steps in the derivation. First, the full set of auxiliary variables, which we gather under one vector,

$$\vec{\varphi} \equiv \Big( \{A_\mu^{(p^2)}\}_{\mu \in [\![1,N]\!]}, \{A_\mu^{(sp)}\}_{\mu \in [\![1,N]\!]}, \{l_\mu^{(p^2)}\}_{\mu \in [\![1,N]\!]}, \{l_\mu^{(sp)}\}_{\mu \in [\![1,N]\!]}, z \Big), \tag{181}$$

(where the $A$s, $l$s and $z$ have the same meaning as in the GGE construction already presented, see Eq. (F.19) and its derivation) can be split into a component acting on the $N$th mode – $\vec{\varphi}_N$ – and the other ones acting on the rest of the modes – $\vec{\varphi}_{\text{bulk}}$. After integrating over $\vec{s}$ and $\vec{p}$ the GGE measure takes the form

$$Z_{\text{GGE}} \propto \int d\vec{\varphi} \, e^{-S(\vec{\varphi})} \propto \int d\vec{\varphi} \, e^{-S_N(\vec{\varphi}_N) - S_{\text{bulk}}(\vec{\varphi})}, \tag{182}$$

see the development in App. F. Written in this form $S_N$ depends only on $\vec{\varphi}_N$, $S_{\text{bulk}}$ involving all other modes, and the modes are coupled through $z$. The end point in this Appendix is that the calculation returns the harmonic *Ansatz* with

$$\langle s_\mu^2 \rangle_{\text{GGE}} = \frac{T_\mu}{z - \lambda_\mu} \qquad \text{and} \qquad \langle p_\mu^2 \rangle_{\text{GGE}} = m T_\mu \qquad \text{for} \quad \mu \neq N, \tag{183}$$

and slightly different conditions on the $N$th mode

$$\langle p_N^2 \rangle_{\text{GGE}} \;\; = \;\; m(z - \lambda_N) \langle m_s^2 \rangle_{\vec{\varphi}_N}. \tag{184}$$

The last mode action – up to subextensive contributions – then reads

$$2S_N(\vec{\varphi}_N) = \frac{m_s^2}{\langle s_N^2 \rangle_{\vec{s},\vec{p}} - m_s^2}, \tag{185}$$

with

$$\langle \ldots \rangle_{\vec{s},\vec{p}} = \int d\vec{s} \, d\vec{p} \, e^{-S(\vec{s},\vec{p},\vec{\varphi})} \ldots. \tag{186}$$

In phase III, this action describes two magnetised states with opposite magnetisation $\pm m_s$, the fluctuation $\langle s_N^2 \rangle_{\vec{s}, \vec{p}} - m_s^2$ being independent of the magnetisation considered. In phases I, II and IV this saddle-point approach for the GGE measure also describes the correct stationary measure. Indeed, in these regions we trivially have $m_s = 0$ in the thermodynamic limit.

We conclude that the fluctuations of the primary constraint in the symmetry broken GGE vanish in the thermodynamic limit, as well as those of the secondary constraint. This, in turn, implies that for the symmetry broken GGE the formulations imposing the constraints exactly or on average give the same results for large $N$.

### 7.3 Summary

In short, we can extract the following conclusions on the identity or differences in the system's behaviour, depending on whether the constraint is imposed on average or strictly, for both symmetric (Sec. 7.1.1) or symmetry broken (Sec. 7.2.1) initial conditions and in the static calculation (Secs. 7.1.2 and 7.2.2).

- Phases I and II. Imposing the spherical constraint on average or strictly yield equivalent results in the large $N$ limit. First, the fluctuations of the primary and secondary constraints with respect to the *initial condition* measure vanish for large $N$, meaning that the initial conditions are indeed of Neumann form. Moreover, the fluctuations of the constraints maintain their scaling properties throughout the dynamics, and then, the dynamics are also of Neumann type.

- Phase III. The fluctuations of the primary constraint with symmetric initial conditions do not vanish in the large $N$ limit, while the ones of the secondary constraint do vanish in the same limit. The dynamics preserve these scaling properties. The initial conditions are not of Neumann form, but the dynamics do conserve the primary constraint. A simple picture of what is going on is that we are averaging over trajectories that live on a sphere, but a different sphere for each initial condition. However, if we consider symmetry broken initial conditions, the divergence in the fluctuations of the primary constraint are cured and the dynamics of the two models are equivalent even in this sector.

- Phase IV. The properties of the initial conditions are the same as for sector III. The difference is that the scaling in the asymptotic state does respect both constraints (there is no condensation in the long-time averages).

Turning now to the statistical mechanics realm, in Secs. 7.1.2 and 7.2.2 we calculated the scaling of the constraint fluctuations in the GGE formalism without and with symmetry breaking, respectively. This allow us to draw conclusions about the equivalency between the soft GGE with partition function given by Eq. (45) and the strictly constrained GGE, with partition function given by

$$Z_{\text{GGE}} = \int d\vec{s} \, d\vec{p} \, \exp\left[ -\sum_\mu \gamma_\mu I_\mu \right] \delta\left( \sum_\mu s_\mu^2 - N \right) \delta\left( \sum_\mu s_\mu p_\mu \right). \qquad (187)$$

The symmetry broken formulation includes the imaginary fields that generate a non-vanishing average of $s_N$ (phase III). The conclusions are similar to those obtained for the dynamical calculation. In phases I, II and IV the two formulations are equivalent. In phase III, with $\langle s_N \rangle_{\text{GGE}} = 0$ the fluctuations of the primary constraint do not vanish in the large $N$ limit and the soft and strict GGEs are not equivalent. Instead, the symmetry broken formulation of the soft GGE fixes the scaling of the primary constraint rendering the soft and strict GGEs equivalent in the large $N$ limit.

(a)                                           (b)

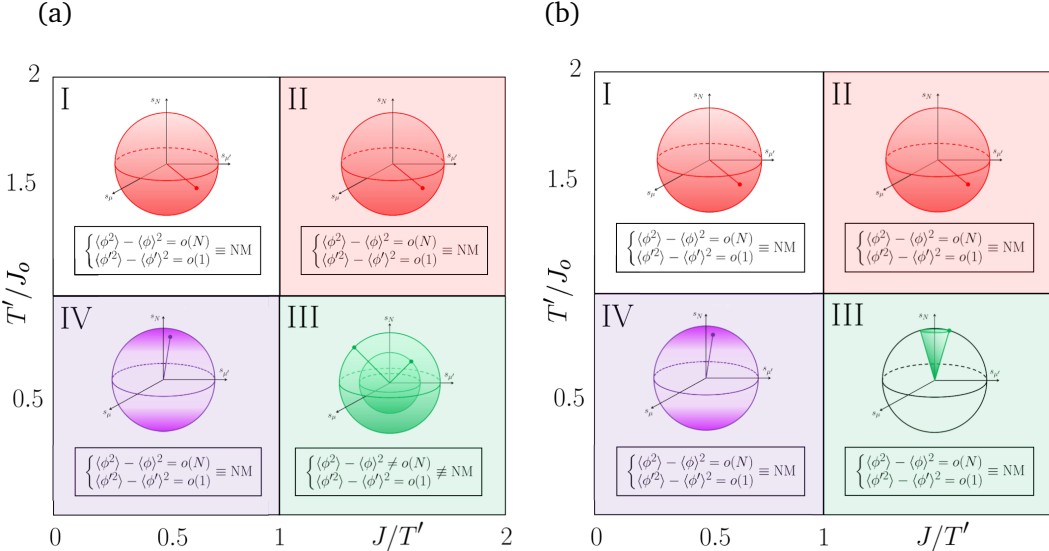

Figure 17: **Dynamic phase diagrams and typical trajectories.** (a) Symmetric initial conditions. The four phases appear as squares in this representation. The scaling with $N$ of the fluctuations of the primary and secondary constraints in the dynamics of the SNM are written in the four boxes. The spheres represent real space and the location of the typical trajectories are represents by the shaded surfaces in I, II, III and with a curve in IV. In phase III all trajectories are embedded on a sphere but their radius is not always $\sqrt{N}$. (b) Symmetry broken initial conditions.

## 8   Conclusions

Our results contribute to the characterisation of macroscopic classical integrable systems, and the understanding of their asymptotic properties in statistical physics terms.

The asymptotic dynamics of the Soft Neumann Model after instantaneous quenches can be rationalised in terms of a rich dynamic phase diagram. Based on the analysis of global quantities like the auto correlation function and the linear susceptibility with the Schwinger-Dyson approach, in [10] we established a phase diagram, which we reproduce in Fig. 4 (b).

This paper completes and gives a much more detailed description of these phases *via* a static and dynamic mode resolved analysis which characterises all $\langle s_\mu^2 \rangle$ and $\langle p_\mu^2 \rangle$ and, in particular, yields the scaling of $\langle s_N^2 \rangle$ and $\langle p_N^2 \rangle$ with system size. Our central result is the calculation of the GGE partition sum and the averages of mode dependent observables which we could express as (implicit) functions of the control parameters. We then successfully compared the GGE averages to the dynamic ones computed numerically over sufficiently long time windows in large systems. In the stationary limit and within our numerical accuracy dynamic and static averages coincide.

In Figs. 17 we illustrate the system's behaviour in the four phases using a representation of the phase diagram, in terms of $(J/T_0, T_0/J_0)$, which renders the phases rectangular. In Fig. 18 we show the plane corresponding to the $N$th mode and we use the notation $\langle \dots \rangle$ to represent both the dynamic and GGE averages. The behaviour of the averaged trajectories in the $N$th plane of phase space are summarised below.

- In phases I, II and IV a typical trajectory moves on the sphere and does not have a macroscopic projection on any of the coordinates, not even the $N$th one, which is singled-out as the vertical direction of the sketch. The GGE averages $\langle p_N \rangle_{\text{GGE}}$ and $\langle s_N \rangle_{\text{GGE}}$ vanish

as also do the time averages $\overline{\langle \ldots \rangle}_{i.c.}$ of the same observables, see Fig. 18. $\langle p_\mu^2 \rangle_{\text{GGE}}$ and $\langle s_\mu^2 \rangle_{\text{GGE}}$ are all $o(N)$. The exact solution that we found in this paper allowed us to prove that $s_N$ and $p_N$ are only quasi-condensed in phase IV, as sketched by the fluctuations sketched in Fig. 18 (d) (contrary to what we claimed in [12]).

- In phase III, a typical trajectory starting from a symmetry broken initial condition with a macroscopic projection on the direction of the $N$th coordinate keeps this projection in the course of time and, typically, precedes around it. The trajectory does not leave the sphere. The GGE as well as the dynamic averages of the momentum in this same $N$th direction, $\langle p_N \rangle_{\text{GGE}} = \overline{\langle p_N \rangle}_{i.c.}$, vanish. Instead, $\langle s_N \rangle_{\text{GGE}} = \overline{\langle s_N \rangle}_{i.c.}$ are proportional to $\pm N^{1/2}$. The sign depends on the sense of the initial condition and the power and prefactor ensure that the spherical constraint is satisfied. This is indicated by the two green dots in Fig. 18 (c). If, instead, symmetric initial conditions are used, $\langle s_N \rangle_{\text{GGE}} = \overline{\langle s_N \rangle}_{i.c.} = 0$ and the large fluctuations of the primary constrained are represented in the $N$th plane in Fig. 18 (b). For both kinds of initial conditions $\langle s_N^2 \rangle = qN$.

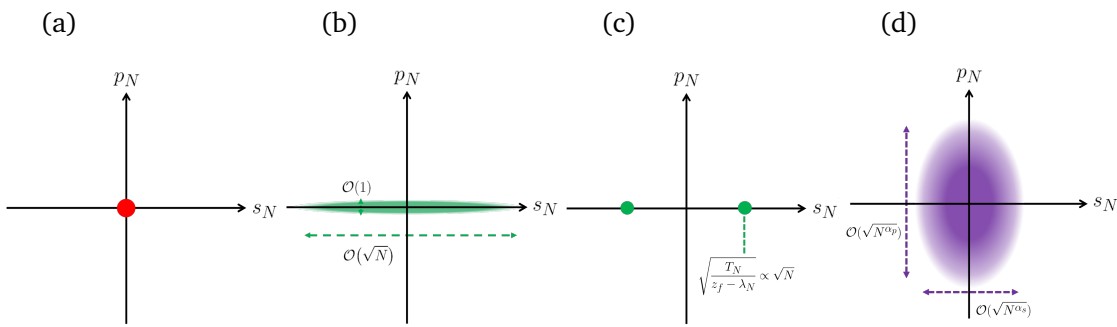

Figure 18: **Sketches of the $s_N$ and $p_N$ dependencies with $N$.** The colour code follows the one of the four phases. (a) Phases I and II. (b) Phases III with symmetric initial conditions. (c) Phases III with symmetry broken initial conditions. (d) Phase IV.

The equivalence between the Soft Neumann Model and the original model in which the constraint is imposed strictly is another issue that deserved our attention. We addressed it by studying the fluctuations of the primary and secondary constraint, which amounts to computing averages of quartic functions of the phase space variables.

- In phase I, II and IV the two models are equivalent since the fluctuations of both constraints vanish in the thermodynamic limit.

- In phase III with symmetric initial conditions all dynamical trajectories of the SNM are embedded on a sphere but their radius is not always equal to $N^{1/2}$. For symmetry broken initial conditions the fluctuations of the primary constraint vanish and the two models become equivalent again.

Still, beyond the possible differences between the SNM and NM, the equivalence between dynamic and stationary averages calculated with the GGE still holds in all phases.

Interesting paths to extend our study could be to consider the effect of weak integrability breaking perturbations [20,21], and a particularly attractive way to do it would be to connect the Hamiltonian dynamics of the Neumann model to the relaxational one of the stochastic open system [62–68]. Another intriguing issue is whether a similar approach can be adapted to treat the ferromagnetic finite dimensional O(N) model, with an explicit space structure.

Let us end with a short comment of other studies of classical integrable models. Several authors have recently developed a Generalised Hydrodynamic Theory of quantum integrable many-body finite dimensional systems with an extensive number of coupled conservation laws [92–95]. The assumption of local Gibbs-Boltzmann equilibrium, at the heart of usual hydrodynamic theories, is replaced in these models by an assumption of local equilibrium in a Generalised Gibbs Ensemble. Following these papers, applications to classical field theories and lattice models were considered, for example, to the sinh-Gordon model [15] and the Toda system [16, 17]. The main difference between the model we treated and the ones studied in these papers is its "mean-field" character or, in other terms, the fact that in terms of interactions, the spherical constraint can be interpreted as a long-range one. This simplification allowed us to obtain exact results in the large $N$ limit with no approximation scheme.

## A   The Wigner semi-circle law

The Wigner semi-circle law is

$$\rho(\lambda) = \frac{1}{2\pi J^2} \sqrt{(2J)^2 - \lambda^2} \qquad \text{for} \qquad \lambda \in [-2J, 2J], \qquad (A.1)$$

and zero otherwise. We recall here, for future reference, a number of integrals of this density. Its normalization and symmetry ensure

$$\int d\lambda \, \rho(\lambda) = 1, \qquad \int d\lambda \, \rho(\lambda) \lambda = 0, \qquad \int d\lambda \, \rho(\lambda) \lambda^2 = J^2. \qquad (A.2)$$

Then,

$$\text{Int}_0(a) \equiv \fint d\lambda \, \rho(\lambda) \frac{1}{a - \lambda} = \begin{cases} \dfrac{a}{2J^2} & a \in [-2J, 2J], \\[2mm] \dfrac{1}{2J^2} \left[ a - \text{sgn}(a) \sqrt{a^2 - (2J)^2} \right] & a \notin [-2J, 2J], \end{cases} \qquad (A.3)$$

with $\fint$ the principal part. With simple recursions one finds

$$
\begin{aligned}
\text{Int}_1(a) &\equiv \fint d\lambda \, \rho(\lambda) \frac{\lambda}{a - \lambda} = \fint d\lambda \, \rho(\lambda) \left[ \frac{\lambda - a}{a - \lambda} + \frac{a}{a - \lambda} \right] = -1 + a \, \text{Int}_0(a), \\
\text{Int}_2(a) &\equiv \fint d\lambda \, \rho(\lambda) \frac{\lambda^2}{a - \lambda} = \fint d\lambda \, \rho(\lambda) \frac{\lambda(\lambda - a + a)}{a - \lambda} = a \, \text{Int}_1(a).
\end{aligned}
\qquad (A.4)
$$

We now use these expressions to evaluate

$$
\begin{aligned}
\fint d\lambda \, \rho(\lambda) \frac{(c - \lambda)(d - \lambda)}{(a - \lambda)(b - \lambda)} = 1 + \frac{1}{b - a} \big[ & (cd - a(c + d) + a^2) \, \text{Int}_0(a) \\
& - (cd - b(c + d) + b^2) \, \text{Int}_0(b) \big],
\end{aligned}
\qquad (A.5)
$$

for $a \neq b$. At this level the expression is symmetric under $a \leftrightarrow b$ and $c \leftrightarrow d$. Some checks are the following. For $b = c = d = 0$ or $a = c = d = 0$ one recovers $-\text{Int}_1(a) = 1 - a \, \text{Int}_0(a)$. For $b = d$ and $a = c$, or $b = c$ and $a = d$, the result reduces to 1, by normalization.

Now we need to distinguish different cases depending on $a \in [-2J, 2J]$ and $b \in [-2J, 2J]$ or not. Multiplying the integral in (A.5) by $2J^2(b-a)$ and calling the result Int:

$$
\begin{cases}
-(b-a)\big[cd-(c+d)(b+a)+a^2+b^2+ab-2J^2\big] & a \& b \in [-2J, 2J], \\[1em]
\big[cd-a(c+d)+a^2\big]a & a \in [-2J, 2J] \& \\
-\big[cd-b(c+d)+b^2\big]\big[b-\sqrt{b^2-(2J)^2}\big]+(b-a)2J^2 & b \notin [-2J, 2J], \\[1em]
\big[cd-a(c+d)+a^2\big]\big[a-\sqrt{a^2-(2J)^2}\big] & \\
-\big[cd-b(c+d)+b^2\big]\big[b-\sqrt{b^2-(2J)^2}\big]+(b-a)2J^2 & a \& b \notin [-2J, 2J].
\end{cases}
\tag{A.6}
$$

(To avoid writing signs, we took $a, b$ positive in the cases in which they are outside the interval $[-2J, 2J]$.) Consistently, all expressions are symmetric with respect to $c \leftrightarrow d$. The first and third cases are also anti-symmetric with respect to $a \leftrightarrow b$ (recall that we multiplied the integral by $b-a$). A particular case, valid for $c = a$ and $d \mapsto a$, is

$$
\fint d\lambda\, \rho(\lambda) \frac{a-\lambda}{b-\lambda} =
\begin{cases}
\dfrac{1}{2J^2}\big[(a-b)b+2J^2\big] & b \in [-2J, 2J], \\[1em]
\dfrac{1}{2J^2}\big[(a-b)\big(b-\sqrt{b^2-(2J)^2}\big)+2J^2\big] & b \notin [-2J, 2J].
\end{cases}
\tag{A.7}
$$

If we look at $c = d$, which is what we have in the integral defining $G$ on the special line $T_0 = (JJ_0)^{1/2} > J_0$,

$$
\fint d\lambda\, \rho(\lambda) \frac{(c-\lambda')^2}{(a-\lambda')(b-\lambda')} = \frac{1}{(b-a)}\big[(c-a)^2 \operatorname{Int}_0(a) - (c-b)^2 \operatorname{Int}_0(b) + (b-a)\big]. \tag{A.8}
$$

One also has

$$
\fint d\lambda\, \rho(\lambda) \frac{a-\lambda}{(b-\lambda)(\lambda-c)} = \frac{b-a}{b-c}\big[-\operatorname{Int}_0(b) + \operatorname{Int}_0(c)\big] - \operatorname{Int}_0(c), \tag{A.9}
$$

for $a \neq b \neq c$.

# B The Neumann Model

The Neumann Model (NM) describes the dynamics of a particle strictly constrained to move on the $N-1$ dimensional sphere under the effect of harmonic forces [25]. It can be formulated in two ways that we summarise below.

## B.1 Constrained formulation

In the constrained formulation the NM is given by a harmonic Hamiltonian

$$
H = \sum_\mu \frac{p_\mu^2}{2m} - \sum_\mu \frac{\lambda_\mu s_\mu^2}{2} \equiv H_{\text{quad}}, \tag{B.1}
$$

with $\lambda_\mu \neq \lambda_\nu$ for $\mu \neq \nu$, under the primary and secondary constraints

$$
\phi = \sum_\mu s_\mu^2 - N = 0, \qquad\qquad \phi' = \frac{1}{N} \sum_\mu s_\mu p_\mu = 0, \tag{B.2}
$$

respectively. The Greek indices $\mu$, $\nu$ run from 1 to $N$. The second equation is a consistency condition that follows from imposing

$$
\dot{\phi} \equiv \frac{d\phi}{dt} = \{\phi, H\} = 0, \tag{B.3}
$$

where the curly brackets are the conventional Poisson ones. Note that we use a notation oriented towards the formulation of the so-called $p = 2$ disordered model that we introduce in Sec. C. In particular, the minus sign in the potential energy implies that the modes with higher/lower energy are the those with lower/higher $\lambda$. In order to calculate the evolution of various quantities, the constraints can be taken into account by transforming the Poisson brackets into Dirac brackets. In the specific case of the spherical constraint the Dirac bracket is given by

$$\{f, g\}_D = \{f, g\} + \frac{1}{2} \{f, \phi\} \{\phi', g\} - \frac{1}{2} \{f, \phi'\} \{\phi, g\} . \tag{B.4}$$

The form of the Dirac brackets is determined solely by the constraints. In this way, the dynamics of the phase space function $f$ under the Hamiltonian $H$ and subject to the constraints $\phi$ and $\phi'$ is given by its Dirac bracket with the Hamiltonian $H$:

$$\dot{f} = \{f, H\}_D . \tag{B.5}$$

Using the Dirac brackets we then derive the equations of motion for the constrained model:

$$
\begin{aligned}
\dot{s}_\mu &= \{s_\mu, H\}_D = \frac{p_\mu}{m}, \\
\dot{p}_\mu &= \{p_\mu, H\}_D = -s_\mu \left[ \frac{1}{N} \sum_\nu \left( \frac{p_\nu^2}{m} + \lambda_\nu s_\nu^2 \right) - \lambda_\mu \right],
\end{aligned} \tag{B.6}
$$

where we have used $\phi' = 0$, and which can be condensed as

$$m\ddot{s}_\mu = -s_\mu(\bar{z}(\{s_\mu\}, \{p_\mu\}) - \lambda_\mu), \tag{B.7}$$

with the phase-space function

$$\bar{z}(\{s_\mu\}, \{p_\mu\}) \equiv \frac{1}{N} \sum_\nu \left( \frac{p_\nu^2}{m} + \lambda_\nu s_\nu^2 \right). \tag{B.8}$$

This function represents the restoring force that keeps the particle on the sphere, and will play an important role when we define the "soft" version of the model in Sec. **??**.

## B.2 Unconstrained formulation

The equations of motion (B.6) can be obtained from a different Hamiltonian involving canonical variables that vary freely in phase space. Following [48], we introduce new momentum variables $r_\mu$ through the canonical transformation

$$p_\mu = r_\mu \left( \frac{1}{N} \sum_\nu s_\nu^2 \right) - s_\mu \sum_\nu s_\nu r_\nu . \tag{B.9}$$

The variables $s_\mu$ and $r_\mu$ are canonical, $\{s_\mu, r_\nu\} = \delta_{\mu\nu}$, and it can be verified that the induced Poisson brackets between the $s_\mu$ and $p_\mu$ variables reproduce the Dirac brackets of the constrained version. Basically,

$$\{s_\mu, p_\nu\}_D = \{s_\mu, p_\nu(s, r)\} , \qquad \{p_\mu, p_\nu\}_D = \{p_\mu(s, r), p_\nu(s, r)\} . \tag{B.10}$$

Moreover, $N^{-1} \sum_\mu s_\mu r_\mu = 0$. The equation of motion for $s_\mu$ and $r_\mu$ can be obtained from the Hamiltonian:

$$H' = \frac{1}{4N} \sum_{\mu \neq \nu} L_{\mu,\nu}^2 - \frac{1}{2} \sum_\mu \lambda_\mu s_\mu^2, \tag{B.11}$$

where the $L_{\mu,\nu}$ are the elements of an angular momentum anti-symmetric matrix

$$\sqrt{m}\, L_{\mu,\nu} = s_\mu r_\nu - r_\mu s_\nu. \tag{B.12}$$

The equations of motion are now obtained in the canonical way

$$\dot{s}_\mu = \frac{\partial H'}{\partial r_\mu}, \qquad\qquad \dot{r}_\mu = -\frac{\partial H'}{\partial s_\mu}, \tag{B.13}$$

and are equivalent to Eqs. (B.6).

## B.3 The constants of motion

Most importantly, K. Uhlenbeck found that the equations of motion of the unconstrained formulation for $s_\mu$ and $r_\mu$ lead to the existence of conserved quantities in involution with respect to the Poisson bracket [47]:

$$
\begin{aligned}
I_\mu &= s_\mu^2 + \frac{1}{mN}\sum_{\nu(\neq\mu)}\frac{1}{\lambda_\nu - \lambda_\mu}L_{\mu,\nu}^2(s,r) \\
&= s_\mu^2 + \frac{1}{mN}\sum_{\nu(\neq\mu)}\frac{s_\mu^2 p_\nu^2 + p_\mu^2 s_\nu^2 - 2s_\mu p_\mu s_\nu p_\nu}{\lambda_\nu - \lambda_\mu}.
\end{aligned} \tag{B.14}
$$

These expressions satisfy two constraints for any choice of the $\lambda_\mu$,

$$\sum_\mu \lambda_\mu I_\mu = -2H', \qquad\qquad \sum_\mu I_\mu = \sum_\mu s_\mu^2. \tag{B.15}$$

Since $\sum_\mu \lambda_\mu I_\mu = -2H'(s,r)$, the dynamics of the $\{s_\mu, r_\mu\}$ system are integrable in the sense of Liouville. The dynamics of the constrained $\{s_\mu, p_\mu\}$ are also integrable, since $L_{\mu,\nu}(s,r) = L_{\mu,\nu}(s,p)$, and the Poisson brackets for $\{s_\mu, r_\mu\}$ imply the Dirac brackets for $\{s_\mu, p_\mu\}$.

# C The spherical Sherrington-Kirkpatrick model

The statistical mechanics of the SNM is directly related to the so called $p = 2$ spherical spin model or spherical Sherrington-Kirkpatrick (SSK) model, for which the $\lambda_\mu$'s are taken to be the eigenvalues of a random real symmetric matrix in which each element is drawn from a Gaussian distribution conveniently normalised, that is to say, a matrix in the GOE ensemble.

In the context of disordered spin systems, the $p = 2$ spherical spin model or SSK has "potential energy" [28]

$$H_{\text{pot}} = -\frac{1}{2}\sum_{i\neq j}J_{ij}s_i s_j = -\frac{1}{2}\sum_\mu \lambda_\mu s_\mu^2. \tag{C.1}$$

The variables are the real "spins" $s_i$, $i = 1,\dots,N$, which interact through the coupling strengths $J_{ij} = J_{ji}$, that can be thought of as being the elements of a real symmetric matrix with eigenvalues $\lambda_\mu$. The last equality is the result of a diagonalisation, and $s_\mu = \vec{s}\cdot\vec{v}_\mu$ with $\vec{s} = (s_1,\dots,s_N)$ and $\vec{v}_\mu$ the $\mu$-th eigenvector or the matrix $J_{ij}$. The sum runs over $\mu = 1,\dots,N$ and we will order the eigenvalues in such as way that $\lambda_1 < \cdots < \lambda_N$. The units are such that $[H_{\text{pot}}] = [J_{ij}] = J$ and $[\lambda_\mu] = [J_{ij}] \Rightarrow [s_\mu^2] = 1$.

In order to constrain the range of variation of the real spins, a global spherical constraint is introduced in the definition of the partition function

$$Z_{\text{eq}} = \int\prod_\mu ds_\mu \int_{c-i\infty}^{c+i\infty} dz_{\text{eq}}\, e^{-\beta H_{\text{pot}}}\, e^{-\frac{\beta z_{\text{eq}}}{2}\left(\sum_\mu s_\mu^2 - N\right)}. \tag{C.2}$$

Working at fixed $z_{\text{eq}}$, the Gaussian integration over the $s_\mu$ yields

$$\langle s_\mu^2 \rangle_{\text{eq}} = \frac{T}{z_{\text{eq}} - \lambda_\mu}, \tag{C.3}$$

with $[z_{\text{eq}}] = J$. The Lagrange multiplier $z_{\text{eq}}$ is then fixed by imposing the spherical constraint on average:

$$1 = \frac{1}{N} \sum_\mu \langle s_\mu^2 \rangle_{\text{eq}} \rightarrow 1 = T \int d\lambda \frac{\rho(\lambda)}{z_{\text{eq}} - \lambda}. \tag{C.4}$$

Using the semi-circle law for the eigenvalue density in the $N \to \infty$ limit, valid for random elements $J_{ij}$ with variance $J^2/N$,

$$\rho(\lambda) = \frac{1}{2\pi J^2} \sqrt{4J^2 - \lambda^2}, \tag{C.5}$$

one finds

$$z_{\text{eq}} - \sqrt{z_{\text{eq}}^2 - (2J)^2} = 2\beta J^2, \tag{C.6}$$

as long as $z_{\text{eq}} > 2J$, leading to the temperature dependent function

$$z_{\text{eq}} = T + \frac{J^2}{T} \equiv z_+ \quad \text{for} \quad T > T_c = J. \tag{C.7}$$

Below $T_c = J$, Eq. (C.6) ceases to have a real solution. The Lagrange multiplier is then fixed to its minimal value

$$z_{\text{eq}} = 2J = \lambda_N \equiv z_-, \tag{C.8}$$

and the spherical constraint (C.4) is no longer satisfied since

$$T \int d\lambda \frac{\rho(\lambda)}{z_{\text{eq}} - \lambda} = T \int d\lambda \frac{\rho(\lambda)}{2J - \lambda} = \frac{T}{J}. \tag{C.9}$$

There are two ways to solve the latter conundrum. The simplest one is to propose that the $N$th mode condenses:

$$s_{\mu=N} = \pm\sqrt{N} \left( 1 - \frac{T}{J} \right)^{1/2} + \delta s_{\mu=N}, \tag{C.10}$$

ensuring the validity of the spherical constraint

$$\frac{1}{N} \sum_\mu \langle s_\mu^2 \rangle_{\text{eq}} = \left( 1 - \frac{T}{J} \right) + \frac{T}{J} = 1. \tag{C.11}$$

In this case, the spin vector $\vec{s}$ has a macroscopic projection on the eigenvector associated to the largest eigenvalue, $\vec{s} \cdot \vec{v}_N \propto \sqrt{N}$. As shown in [28], the introduction of a magnetic field in the equilibrium partition function breaks the $\mathbb{Z}_2$ symmetry ($s_\mu \to -s_\mu$) and introduces such a magnetised state as in as the thermal equilibrium in the low temperature phase.

Another possibility is that the fluctuations of the $N$-th mode are the ones that condense, meaning that

$$\langle s_N^2 \rangle_{\text{eq}} = \left( 1 - \frac{T}{J} \right) N, \tag{C.12}$$

with no macroscopic projection of the spin vector in the direction of $\vec{v}_N$,

$$\langle s_N \rangle_{\text{eq}} = 0. \tag{C.13}$$

One can interpret the types of low temperature thermal equilibrium as arising from two orders of limits. The $\mathbb{Z}_2$ symmetric solution is obtained when the magnetic field is taken to zero before the thermodynamic limit ($\lim_{N\to\infty}\lim_{h\to 0}$) while the $\mathbb{Z}_2$ symmetry broken one is obtained when the thermodynamic limit is taken first ($\lim_{h\to 0}\lim_{N\to\infty}$). We will see the influence of these two kinds of initial states, and how similar condensation of fluctuations arise in the dynamics of the SNM model in Sec. 7. For more details on the difference of the two kinds of equilibrium at $T < J$ and how this is related to inequivalence of equilibrium ensembles see [72–74].

The equilibrium linear susceptibility to a field that couples globally to the spins, $-h\sum_i s_i = -h\sum_\mu s_\mu$ is

$$\chi_{\text{eq}} \equiv \frac{1}{N}\sum_\mu \frac{\delta\langle s_\mu\rangle^h_{\text{eq}}}{\delta h}\bigg|_{h=0} = \frac{\beta}{N}\sum_\mu\left(\langle s_\mu^2\rangle_{\text{eq}} - \langle s_\mu\rangle^2_{\text{eq}}\right) = \begin{cases} 1/T & T > J\,, \\ 1/J & T < J\,, \end{cases} \tag{C.14}$$

where in the first identity we used the static fluctuation-dissipation theorem.

The static properties of this model and, especially, its fluctuations, have called a recent surge of interest in the mathematical physics community [52–59], see also [60,61].

In the statistical physics context, the relevant dynamics to be considered are of Langevin type, with dissipation and noise induced by the coupling to a bath. Many studies of the relaxation dynamics of this model after quenches across the critical temperature into the low temperature phase demonstrated that, in the course of time, the spin configuration tends to align with the eigenvector associated to the largest eigenvalue without being able to do it if the $N \to \infty$ limit is taken from the outset [62–68]. If, instead, one lets time scale with $N$ three relaxation regimes are clearly distinguished in the approach to thermal equilibrium [69,70].

# D   Averaged constants of motion

We evaluate two types of average of the Uhlenbeck constants of motion. In App. D.1 we use the canonical equilibrium Gibbs Boltzmann distribution at temperature $T_0$ for the initial conditions, and we take the mean over it. In App. **??** we average over the GGE measure using the harmonic *Ansatz*.

## D.1   Thermal initial conditions

On average over the equilibrium initial measure $\rho_0$ the constants of motion are

$$\begin{aligned}
\langle I_\mu(0^+)\rangle_{i.c.} &= \langle s_\mu^2(0^+)\rangle_{i.c.} \\
&+ \frac{1}{mN}\sum_{\nu(\neq\mu)}\frac{1}{\lambda_\nu - \lambda_\mu}\Big[\langle s_\mu^2(0^+)p_\nu^2(0^+)\rangle_{i.c.} + \langle p_\mu^2(0^+)s_\nu^2(0^+)\rangle_{i.c.} \\
&\qquad\qquad\qquad\qquad -2\langle s_\mu(0^+)p_\mu(0^+)s_\nu(0^+)p_\nu(0^+)\rangle_{i.c.}\Big] \\
&= \langle s_\mu^2(0^+)\rangle_{i.c.} + \frac{1}{mN}\sum_{\nu(\neq\mu)}\frac{\langle s_\mu^2(0^+)\rangle_{i.c.}\langle p_\nu^2(0^+)\rangle_{i.c.} + \langle p_\mu^2(0^+)\rangle_{i.c.}\langle s_\nu^2(0^+)\rangle_{i.c.}}{\lambda_\nu - \lambda_\mu}\,.
\end{aligned} \tag{D.1}$$

The factorisation of the average of the four factors in the last term, which eventually makes its contribution vanish, is justified because of the Gaussian character of the measure and because the sum runs over $\nu(\neq\mu)$. The factorisation is safe even in cases in which the fluctuations of the $N$th mode are macroscopic. Apart from sub-leading corrections, these averages satisfy the

two global constraints:

$$\sum_{\mu} \langle I_{\mu}(0^+) \rangle_{i.c.} = \sum_{\mu} \langle s_{\mu}^2(0^+) \rangle_{i.c.} = N \,, \tag{D.2}$$

$$\sum_{\mu} \lambda_{\mu} \langle I_{\mu}(0^+) \rangle_{i.c.} = \sum_{\mu} \lambda_{\mu} \langle s_{\mu}^2(0^+) \rangle_{i.c.} - \frac{1}{m} \sum_{\mu} \langle p_{\mu}^2(0^+) \rangle_{i.c.}$$

$$+ \frac{1}{mN} \sum_{\mu} \langle s_{\mu}^2(0^+) \rangle_{i.c.} \langle p_{\mu}^2(0^+) \rangle_{i.c.}$$

$$= -2 \langle H(0^+) \rangle_{i.c.} + \mathcal{O}(1) \,. \tag{D.3}$$

### D.1.1 $T_0 > J_0$ (extended initial condition)

In this case

$$\langle I_{\mu}(0^+) \rangle_{i.c.} = \frac{T_0}{z_{eq}(T_0) - \lambda_{\mu}^{(0)}} + \frac{T_0^2}{N} \sum_{\nu(\neq\mu)} \frac{1}{\lambda_{\nu} - \lambda_{\mu}} \left[ \frac{1}{z_{eq}(T_0) - \lambda_{\mu}^{(0)}} + (\mu \leftrightarrow \nu) \right] \,, \tag{D.4}$$

and the sums can be readily rearranged in such a way that they can be calculated analytically:

$$\langle I_{\mu}(0^+) \rangle_{i.c.} = \left[ 1 + \frac{2T_0 J_0}{J} \int_{-2J_0}^{2J_0} d\lambda' \frac{\rho(\lambda')}{\lambda' - \lambda_{\mu}^{(0)}} + \frac{T_0 J_0}{J} \int_{-2J_0}^{2J_0} d\lambda' \frac{\rho(\lambda')}{z_{eq}(T_0) - \lambda'} \right] \,.$$

The kind of integrals in the last two terms appear in the derivation of the Wigner semi-circle law using the Coulomb gas approach and are sometimes called Tricomi's Theorem. They are recalled in App. A as well. When $\lambda'$ lies within the interval of variation of the integration variable, as in the second term between the square brackets,

$$\int_{-2J_0}^{2J_0} \frac{d\lambda'}{2\pi J_0^2} \frac{\sqrt{(2J_0)^2 - \lambda'^2}}{\lambda' - \lambda_{\mu}^{(0)}} = -\frac{\lambda_{\mu}^{(0)}}{2J_0^2} \,, \tag{D.5}$$

see Eq. (5.24) in Ref. [90]. The result for the second integral is different, since $T_0 > J_0$ and the Lagrange multiplier $z_{eq}(T_0)$ is larger than $2J_0$ (or equal to this value at $T_0 = J_0$). The integral yields

$$\int_{-2J_0}^{2J_0} d\lambda' \frac{\rho(\lambda')}{z_{eq}(T_0) - \lambda'} = \frac{1}{2J_0^2} \left[ z_{eq}(T_0) - \sqrt{(z_{eq}(T_0))^2 - 4J_0^2} \right] \,. \tag{D.6}$$

Then

$$\langle I_{\mu}(0^+) \rangle_{i.c.} = \frac{T_0}{z_{eq}(T_0) - \lambda_{\mu}^{(0)}} \left\{ 1 - \lambda_{\mu}^{(0)} \frac{T_0}{J_0 J} + \frac{T_0}{2J_0 J} \left[ z_{eq}(T_0) - \sqrt{(z_{eq}(T_0))^2 - 4J_0^2} \right] \right\} \,.$$

We can now use $z_{eq}(T_0) = T_0 + J_0^2/T_0$ to simplify a bit this form

$$\langle I_{\mu}(0^+) \rangle_{i.c.} = \frac{T_0^2}{J_0 J} \frac{(J_0^2 + J_0 J)/T_0 - \lambda_{\mu}^{(0)}}{(J_0^2 + T_0^2)/T_0 - \lambda_{\mu}^{(0)}} = \frac{T_0^2}{J_0 J} \frac{(J_0^2 + J_0 J)/T_0 - J_0 \lambda_{\mu}/J}{(J_0^2 + T_0^2)/T_0 - J_0 \lambda_{\mu}/J} \,. \tag{D.7}$$

We have verified that the two constraints, $\sum_{\mu} \langle I_{\mu} \rangle_{i.c.} = N$ and $\sum_{\mu} \lambda_{\mu}^{(0)} \langle I_{\mu} \rangle_{i.c.} = -2 \langle H \rangle_{i.c.} = -\sum_{\mu} \langle p_{\mu}^2 \rangle_{i.c.}/m + \sum_{\mu} \lambda_{\mu}^{(0)} \langle s_{\mu}^2 \rangle_{i.c.}$, are satisfied by these expressions.

For future reference, we can see which is the condition on the parameters imposed by $\langle I_\mu(0^+)\rangle_{i.c.} > 0$. Focusing on the largest eigenvalue, $\mu = N$, for which $\lambda_N^{(0)} = 2J_0$, one easily checks that the denominator is a perfect square and hence always positive. Concerning the numerator, the condition for positivity is

$$1 < y < \frac{1+x}{2} \qquad \text{with} \qquad x = \frac{J}{J_0} \text{ and } y = \frac{T_0}{J_0}. \tag{D.8}$$

In the full phase I and in a part of phase II, delimited by the two straight lines in the inequality, $\langle I_N(0^+)\rangle_{i.c} < 0$. To the right of this line, in phase II, $\langle I_N\rangle_{i.c.} > 0$.

A special case is the one in which the numerator and denominator in the second factor in Eq. (D.7) are equal and the $\langle I_\mu\rangle_{i.c.}$ lose their $\mu$ dependence. This is achieved for $J_0^2 + J_0 J = J_0^2 + T_0^2$ which implies $J/J_0 = (T_0/J_0)^2$ and is shown with a blue line within phase II in Fig. 4. On this curve, all $\langle I_\mu\rangle_{i.c.}$ equal one and satisfy the two constraints, $\sum_\mu \langle I_\mu\rangle_{i.c.} = N$ and $\sum_\mu \lambda_\mu \langle I_\mu\rangle_{i.c.} = -2\langle H(0^+)\rangle_{i.c.} = -2e_f = 0$, see Table 1.

Several typical cases are plotted in Fig. 3 (a).

### D.1.2 $T_0 < J_0$ (condensed initial condition)

For $\mu = N$ we can focus on the leading $\mathcal{O}(N)$ contribution

$$
\begin{aligned}
\langle I_N(0^+)\rangle_{i.c.} &= \langle s_N^2\rangle_{i.c.}\left(1 + \frac{T_0}{N}\sum_{\nu(\neq N)}\frac{1}{\lambda_\nu - \lambda_N}\right) + \langle p_N^2\rangle_{i.c.}\frac{1}{mN}\sum_{\nu(\neq N)}\frac{\langle s_\nu^2\rangle_{i.c.}}{\lambda_\nu - \lambda_N} \\
&\mapsto \left(1 - \frac{T_0}{J_0}\right)\left(1 - \frac{T_0}{J}\right)N - \frac{1}{N}\sum_{\nu(\neq N)}\frac{T_0^2}{z^{(0)} - \lambda_\nu^{(0)}}\frac{1}{\lambda_N - \lambda_\nu},
\end{aligned} \tag{D.9}
$$

where we took the continuum limit in the calculation of the integral in the first term. The last term is more delicate to handle. In the infinite $N$ limit we know that $z^{(0)} \to 2J_0$ and $\lambda_N \to 2J$ so we can expect this full second term to diverge with $N$. One can check numerically that $N^{-1}\sum_{\mu\neq N}(\lambda_N - \lambda_\mu)^{-2} = o(N)$. A way to confirm the sub-linear scaling with $N$ is to check that the first term is enough to ensure the normalisation of the constants of motion once the other $\langle I_\mu\rangle_{i.c.}$ with $\mu \neq N$ as computed. Indeed, for $\mu \neq N$

$$
\begin{aligned}
\langle I_\mu(0^+)\rangle_{i.c.} &= \langle s_\mu^2\rangle_{i.c.}\left(1 + \frac{T_0}{N}\sum_{\nu(\neq\mu)}\frac{1}{\lambda_\nu - \lambda_\mu}\right) + \langle p_\mu^2\rangle_{i.c.}\frac{1}{mN}\sum_{\nu(\neq\mu,N)}\frac{\langle s_\nu^2\rangle_{i.c.}}{\lambda_\nu - \lambda_\mu} \\
&+ \langle p_\mu^2\rangle_{i.c.}\frac{1}{mN}\frac{\langle s_N^2\rangle_{i.c.}}{\lambda_N - \lambda_\mu} = \frac{T_0^2}{J_0^2 J}\frac{(JJ_0 + J_0^2)/T_0 - J_0\lambda_\mu/J}{2 - \lambda_\mu/J}.
\end{aligned} \tag{D.10}
$$

This expression coincides with the one in Eq. (D.7) for $T_0 > J_0$ if we identify $z_{eq}(T_0)$ there with $2J_0$ here. We can also check that the normalised sum of the $I_\mu$s, including the $\mathcal{O}(1)$ contribution of the $N$-th mode, equals 1. Therefore,

$$\langle I_N(0^+)\rangle_{i.c.} = \left(1 - \frac{T_0}{J_0}\right)\left(1 - \frac{T_0}{J}\right)N. \tag{D.11}$$

Since $T_0/J_0 < 1$ the first factor is positive. In phase III $J/J_0 > T_0/J_0$ and, therefore, $J > T_0$ implying that $\langle I_N(0^+)\rangle_{i.c.}$ is positive. Instead, in phase IV $J/J_0 < T_0/J_0$ and one has that $\langle I_N(0^+)\rangle_{i.c.}$ is negative. Summarising

$$\langle I_N(0^+)\rangle_{i.c.} = \begin{cases} > 0 & \text{in phase III,} \\ < 0 & \text{in phase IV.} \end{cases} \tag{D.12}$$

In phase IV one can identify the straight line $2T_0/J_0 = (1 + J/J_0)$ on which the bulk constants of motion are all equal, $\langle I_{\mu \neq N} \rangle_{i.c.} = T_0{}^2/(J_0 J)$, and $\langle I_N \rangle_{i.c.} = -(1-J/J_0)^2 J_0/(4J)N$, complying with the two constraints in the large $N$ limit. The straight line $2T_0/J_0 = (1 + J/J_0)$ is shown in Fig. 4 and the bulk $\langle I_{\mu \neq N} \rangle_{i.c.}$ in some representative cases are depicted in Fig. 3 (b).

The conservation of the integrals imposes constraints on the stationary state reached after the quench. Four different regions of the phase diagram, see Fig. 4, are easily identified according to the sign and scaling with $N$ of $\langle I(\lambda_N) \rangle_{i.c.}$ although do not coincide exactly with the dynamic phases.

### D.2 The constants of motion in the GGE

Separating the sums as we did in Eq. (D.9),

$$\langle I_\mu \rangle_{\text{GGE}} = \langle s_\mu^2 \rangle_{\text{GGE}} \left( 1 + \frac{1}{N} \sum_{\nu(\neq \mu)} \frac{\langle p_\nu^2 \rangle_{\text{GGE}}}{\lambda_\nu - \lambda_\mu} \right) + \langle p_\mu^2 \rangle_{\text{GGE}} \frac{1}{mN} \sum_{\nu(\neq \mu)} \frac{\langle s_\nu^2 \rangle_{\text{GGE}}}{\lambda_\nu - \lambda_\mu} . \tag{D.13}$$

In the second sum we have to consider separately the case $\mu \neq N$ and $\mu = N$, and the last one is the tricky one, since in phases III and IV, $\langle I_N \rangle_{\text{GGE}}$ should be $\mathcal{O}(N)$.

Let us first look at phase III, where $\langle s_N^2 \rangle_{\text{GGE}} = \mathcal{O}(N)$ and $\langle p_N^2 \rangle_{\text{GGE}} = \mathcal{O}(1)$ while all the other averages are also $\mathcal{O}(1)$. The averaged $N$th constant of motion is

$$\begin{aligned}
\langle I_N \rangle_{\text{GGE}} &= \langle s_N^2 \rangle_{\text{GGE}} \left( 1 + \frac{1}{mN} \sum_{\nu(\neq N)} \frac{\langle p_\nu^2 \rangle_{\text{GGE}}}{\lambda_\nu - \lambda_N} \right) + \langle p_N^2 \rangle_{\text{GGE}} \frac{1}{mN} \sum_{\nu(\neq N)} \frac{\langle s_\nu^2 \rangle_{\text{GGE}}}{\lambda_\nu - \lambda_N} \\
&= qN \left( 1 + \frac{1}{N} \sum_{\nu(\neq N)} \frac{T_\nu}{\lambda_\nu - \lambda_N} \right) - T_N \frac{1}{N} \sum_{\nu(\neq N)} \frac{T_\nu}{(\lambda_\nu - \lambda_N)^2} \\
&\to qN \left( 1 - \frac{1}{N} \sum_{\nu(\neq N)} \frac{T_\nu}{\lambda_N - \lambda_\nu} \right) = q^2 N ,
\end{aligned} \tag{D.14}$$

where we dropped the sub-leading contribution of the last term in the second line, and we obtained a result which is consistent with Eq. (D.11).

In phase IV we know $\langle p_\mu^2 \rangle_{\text{GGE}} = (z - \lambda_\mu) \langle s_\mu^2 \rangle_{\text{GGE}}$ for all $\mu$ including $\mu = N$, and $z > \lambda_N$. Moreover, the analytic solution explained in the main body of the paper indicates that $\langle s_\mu^2 \rangle_{\text{GGE}} \propto (\lambda_N - \lambda_\mu)^{-1}$. Thus, the two sums in the first line of Eq. (D.14) have a power of $(\lambda_N - \lambda_\nu)^2$ in the denominator with a finite numerator. We already know that these sums, in the large $N$ limit, go as a power of $N$ which is smaller than one, say $N^a$. Thus, they dominate the right hand side and

$$\langle I_N \rangle_{\text{GGE}} \propto -\langle s_N^2 \rangle_{\text{GGE}} N^a . \tag{D.15}$$

Then, the result can be proportional to $N$ if $\langle s_N^2 \rangle_{\text{GGE}} \propto N^{1-a}$. Note that the sign is correct, since $\langle I_N \rangle_{\text{GGE}}$ is negative in phase IV. $\langle p_N^2 \rangle_{\text{GGE}}$ is also proportional to $N^{1-a}$ in this phase because of the harmonic relation. In short we have

$$\langle s_N^2 \rangle_{\text{GGE}} = (z - \lambda_N) \langle p_N^2 \rangle_{\text{GGE}} = \mathcal{O}(N^{1-a}) . \tag{D.16}$$

## E  Details of the exact solution

In this Appendix we give more details on the exact solution. In particular, we focus on the behaviour close to the edge of the spectrum of harmonic constants, that is to say, on the coordinates $\mu \sim N$.

Both $g^2$ and $G$ take some special forms in certain phases of the phase diagram or for special relations of the parameters. They can also be simply expanded for $0 < \epsilon = 2J - \lambda \ll 1$. We review some of these useful properties here.

### E.1 Properties of $g(\lambda)$

Two simple cases are

$$[4J^2 g(\lambda)]^2 = [(2J)^2 - \lambda^2] \times \begin{cases} (z - \lambda)^2 & \text{for } a = b \text{ in II}, \\ (b - \lambda)^2 k_1^2 & \text{in III since } z = a. \end{cases} \tag{E.1}$$

In equilibrium $J_0 = J$ and this implies $b = 2J^2/T_0$ and $a = z$ at all $T_0$. These parameters fall in phases I and III. The latter relation induces some simplifications and

$$g_{\text{eq}}(\lambda) = \frac{\pi T_0^2}{2J^2} \rho(\lambda) \left( \frac{2J^2}{T_0} - \lambda \right). \tag{E.2}$$

For $a \neq 2J$, that is, in phases I and II,

$$[4J^2 g(\epsilon)]^2 = k_1^2 (z - 2J + \epsilon)^2 \left[ 4J \frac{(b - 2J)^2}{(a - 2J)^2} \epsilon \right.$$
$$\left. + \frac{(-ab^2 + 12abJ - 6b^2J - 20aJ^2 + 8bJ^2 + 8J^3)}{(a - 2J)^3} \epsilon^2 + \mathcal{O}(\epsilon^3) \right]. \tag{E.3}$$

In phase I, the leading order is order $\epsilon$. In phase II, $z = 2J$, and the lowest order is $\epsilon^3$. On the special curve $T_0 = (JJ_0)^{1/2}$ in phase II

$$[4J^2 g(\epsilon)]^2 = \epsilon^3 [4J - \epsilon]. \tag{E.4}$$

In phase III, $a = z = 2J$ and

$$[4J^2 g(\epsilon)]^2 = k_1^2 \epsilon (4J - \epsilon)(b - 2J + \epsilon). \tag{E.5}$$

Instead, in IV, $a = 2J$, $z \neq 2J$ and

$$[4J^2 g(\epsilon)]^2 = 4J k_1^2 (z - 2J)^2 (b - 2J)^2 \frac{1}{\epsilon}, \tag{E.6}$$

to leading order. On the special curve, $T_0 = (JJ_0)^{1/2}$, $b = z$ and the two factors in the numerator combine into a fourth power.

### E.2 Properties of $G(\lambda)$

In equilibrium $J_0 = J$, $b = 2J^2/T_0$, and $a = z$ at all $T_0$. One has

$$G_{\text{eq}}(\lambda) = \frac{T_0^2}{2J^4} \left( \lambda^2 - \frac{2J^2}{T_0} \lambda - 2J^2 \right). \tag{E.7}$$

Working a little bit with these expressions one recovers $\chi_I = \pi \rho T_0$, with $T_0$ constant, from our general solution of the quartic equation above, setting $f = 0$.

On the special curve in II, on which $k_1 = 1$, $a = b$, and $\langle I_\mu \rangle_{i.c.} = 1$,

$$1 + G(\lambda) = \frac{\lambda}{2J^2} (\lambda - z) < 0 \qquad \text{if } a = b \text{ in II}. \tag{E.8}$$

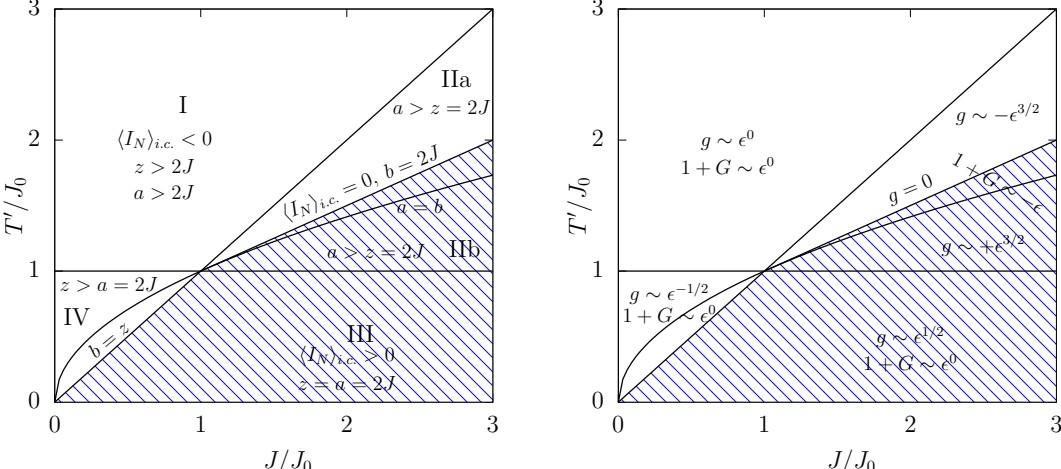

Figure 19: Parameters helping the calculation of the limits

Since $z = 2J$ in this phase, one easily sees from there that the expansion close to the edge starts at order $\epsilon$ on this line. In the rest of phase II, the $\mathcal{O}(\epsilon^0)$ also brings some special consequences, since it vanishes identically, and one has

$$G(2J - \epsilon) = -1 \qquad \text{in the full phase II}. \tag{E.9}$$

In the full phase III, one finds a similar expression for $G(\lambda)$ since $a = z$,

$$G(\lambda) = \frac{T_0^2}{2J_0 J^3} \left( \lambda^2 - b\lambda - 2J^2 \right) \qquad \text{in the full phase III}. \tag{E.10}$$

For the particular equilibrium case $J_0 = J$, one recovers Eq. (E.7). In phase IV, $a = 2J$ but $z \neq a > 2J$. In particular, on the special line in IV, $T_0^2 = J_0 J$, $k_1 = 1$, $b = z = (J/J_0)^{1/2}(J + J_0)$ and

$$1 + G(\lambda) = \frac{1}{2J^2} \left[ \lambda^2 - 2\lambda(z - J) + (z - 2J)^2 \right] \qquad \text{if } b = z \text{ in IV}. \tag{E.11}$$

In complete generality, close to the edge of the $[-2J, 2J]$ interval, for $\epsilon = 2J - \lambda$, $G$ behaves as

$$
\begin{aligned}
\frac{2J^2}{k_1} G(\epsilon) =\ & 2J(a - b - z + J) - (a - b)\sqrt{a^2 - 4J^2} \frac{a - z}{a - 2J} + (b - a)(z - a) \\
& + \left[ -a + b - 4J + z + \frac{(a - b)\sqrt{a^2 - 4J^2}(a - z)}{(a - 2J)^2} \right] \epsilon \\
& + \left[ 1 - \frac{(a - b)\sqrt{a^2 - 4J^2}(a - z)}{(a - 2J)^3} \right] \epsilon^2 \\
& + \mathcal{O}(\epsilon^3).
\end{aligned}
\tag{E.12}
$$

### E.3 The solution close to the edge of the spectrum

At the right edge of the $[-2J, 2J]$ interval, if $f = 0$ and $1 + G \sim +\epsilon^0$ (finite and positive) the situation in phases I and III,

$$2\chi_I^2 = 2(\pi\rho T)^2 \simeq \frac{2g^2}{1 + G} = \frac{2\pi^2}{4} \frac{\rho^2 (z - \lambda)^2 |\langle I \rangle_{i.c.}|^2}{1 + G}, \tag{E.13}$$

for $\lambda = 2J - \epsilon$, and this implies

$$
\begin{aligned}
T(2J) &= \frac{1}{2}(z - 2J)|\langle I(2J)\rangle_{i.c.}| \frac{1}{\sqrt{1 + G(2J)}} \\
&= \frac{(z - 2J)}{(a - 2J)} \frac{T_0}{2J_0} \frac{|J_0 + J - 2T_0|}{\sqrt{1 + G(2J)}}.
\end{aligned}
\tag{E.14}
$$

- In phase I, $z > 2J$ and $a > 2J$ and $0 < T(2J) = \langle p^2(2J)\rangle_{\text{GGE}}$ and $0 < \langle s^2(2J)\rangle_{\text{GGE}} = (z - 2J)\langle p^2(2J)\rangle_{\text{GGE}}$ are also finite.

- In phase III, $z = a = 2J$, two factors cancel and $T(2J) = 1/2 \sqrt{T_0 J/J_0}(J_0 + J - 2T_0)/\sqrt{T_0 - J - J_0}$ is finite. Instead, $\langle s^2(2J)\rangle_{\text{GGE}}$ is inversely proportional to $2J - \lambda$ and diverges at $\lambda \to 2J$.

In phase II, $z = 2J$ and $a > 2J$. Right at the edge of the spectrum $1 + G(2J) = 0$. The first order correction in $\epsilon$ becomes the leading one and it is proportional to $(T_0 - J)/(T_0 - J_0)$ (with positive proportionality constant) and hence negative in the full phase, $1 + G(2J - \epsilon) \sim -\epsilon$. Concerning $g$, $g(2J - \epsilon) = \mathcal{O}(\epsilon^{3/2})$ changing sign on another special straight line, on which it vanishes identically. It is therefore smaller than $1 + G$ in the full phase.

- In phase II one cannot apply the expression (E.14). Instead,

$$
\begin{aligned}
2\chi_I^2 &= 2(\pi\rho T)^2 \simeq -(1 + G) + |1 + G|\left[1 + 2\frac{g^2}{(1 + G)^2}\right] \\
&\simeq -2(1 + G) + 2\frac{g^2}{|1 + G|} \simeq -2(1 + G).
\end{aligned}
\tag{E.15}
$$

From here we get $\langle p^2(2J - \epsilon)\rangle_{\text{GGE}} = T(2J - \epsilon) \propto \epsilon/\epsilon = \mathcal{O}(1)$. Consequently, $\langle s^2(2J - \epsilon)\rangle_{\text{GGE}} = \mathcal{O}(\epsilon)$.

- On the special line in II, the solution is derived in a different way. $T(\lambda) = J$ is finite for all $\lambda$ and thus $\langle s^2(2J)\rangle_{\text{GGE}}$ should diverge driven by the denominator $z - 2J$.

Finally, in phase IV, $z > 2J$, $a = 2J$ and $1 + G(2J) \neq 0$.

- Both $T(\lambda)$ and $\langle s^2(\lambda)\rangle_{\text{GGE}}$ diverge at the edge in the same way, driven by the divergence of $\langle I_N\rangle_{i.c.}$, and should therefore scale with $N$.

- On the special line in IV, we know $T(\lambda) = J[T_0/J_0(J + J_0) - \lambda]/[2J - \lambda]$ and it diverges at the edge.

# F  Saddle-point and broken symmetries

In this Appendix we explain how to implement symmetry breaking in the GGE formalism, in other words, how to let the $N$th mode coordinate and momentum acquire averages which scale as $N^{1/2}$.

Take the GGE action

$$
S(\vec{s}, \vec{p}, z) = \sum_{\mu=1}^{N}\left(\gamma_\mu + \frac{z}{2J}\right)s_\mu^2 - \frac{1}{mN}\sum_{\mu \neq \nu}\eta(\mu, \nu)(s_\mu^2 p_\nu^2 - s_\mu s_\nu p_\mu p_\nu).
\tag{F.16}
$$

Following the same approach as in Sec. 4, we decouple the quartic interactions using the auxiliary variables $A_\mu^{(p^2)}$, $A_\mu^{(sp)}$, and now also $m_s$ defined as

$$A_\mu^{(p^2)} = \frac{1}{N} \sum_{\nu(\neq\mu)} \eta(\mu,\nu) s_\nu^2, \qquad A_\mu^{(sp)} = \frac{1}{mN} \sum_{\nu(\neq\mu)} \eta(\mu,\nu) s_\nu p_\nu,$$

$$m_s = s_N. \tag{F.17}$$

The last variable will let the $N$th mode have non zero averages. The condition is introduced with an imaginary Lagrange multiplier:

$$1 = \int dm_s\, \delta(m_s - s_N) \propto \int dm_s\, dh_s\, \exp\left[h_s(m_s - s_n)\right]. \tag{F.18}$$

In this context the sum in the action does not need to be completed with the $\mu = \nu$ term. The fields $h_s$ and $m_s$ will guarantee (sub)extensive contributions to the last mode, making the previous continuation irrelevant. In order to make the notation more compact, we collect all other auxiliary variables in a single vector $\vec{\varphi}$:

$$\vec{\varphi} = \left(\{A_\mu^{(p^2)}\}_{\mu \in [\![1,N]\!]},\ \{A_\mu^{(sp)}\}_{\mu \in [\![1,N]\!]},\ \{l_\mu^{(p^2)}\}_{\mu \in [\![1,N]\!]},\ \{l_\mu^{(sp)}\}_{\mu \in [\![1,N]\!]},\ z\right). \tag{F.19}$$

In phases III we expect extensive contribution from the $N$th mode. The point of the following calculation is to perform single-valued saddle-points for $\vec{\varphi}$ and multi-valued saddle-points for $m_s$ and $h_s$ – the variables describing the last mode. With the introduction of $\vec{\varphi}$, $m_s$ and $h_s$ to decouple the quartic interactions, the action reduces to

$$\begin{aligned}
S(\vec{s}, \vec{p}, \vec{\varphi}, \tilde{m}, \tilde{h}) &= \sum_\mu \left[(\gamma_\mu + \frac{z}{2J})s_\mu^2 - \frac{p_\mu^2}{m}A_\mu^{(p^2)} + s_\mu p_\mu A_\mu^{(sp)}\right] - h_s(m_s - s_N) \\
&\quad - \sum_\mu l_\mu^{(p^2)}\left(A_\mu^{(p^2)} - \frac{1}{N}\sum_{\nu(\neq\mu)} \eta(\mu,\nu) s_\nu^2\right) \\
&\quad - \sum_\mu l_\mu^{(sp)}\left(A_\mu^{(sp)} - \frac{1}{mN}\sum_{\nu(\neq\mu)} \eta(\mu,\nu) s_\nu p_\nu\right) \\
&= \frac{1}{2}\sum_\mu V_\mu^\dagger M_\mu V_\mu - \sum_\mu\left(l_\mu^{(sp)} A_\mu^{(sp)} + l_\mu^{(p^2)} A_\mu^{(p^2)}\right) - h_s(m_s - s_N),
\end{aligned} \tag{F.20}$$

where

$$M_\mu \equiv \begin{bmatrix} M_\mu^{(ss)} & M_\mu^{(sp)} \\ M_\mu^{(ps)} & M_\mu^{(pp)} \end{bmatrix}, \tag{F.21}$$

with components

$$\begin{aligned}
M_\mu^{(ss)} &= 2\gamma_\mu + \frac{z}{J} + \frac{2}{N}\sum_{\nu(\neq\mu)} \eta(\mu,\nu) l_\nu^{(p^2)}, \\
M_\mu^{(sp)} &= A_\mu^{(sp)} + \frac{1}{mN}\sum_{\nu(\neq\mu)} \eta(\mu,\nu) l_\nu^{(sp)}, \\
M_\mu^{(ps)} &= A_\mu^{(sp)} + \frac{1}{mN}\sum_{\nu(\neq\mu)} \eta(\mu,\nu) l_\nu^{(sp)}, \\
M_\mu^{(pp)} &= -\frac{2}{m}A_\mu^{(p^2)}.
\end{aligned} \tag{F.22}$$

Finally,

$$V_\mu = \begin{bmatrix} s_\mu \\ p_\mu \end{bmatrix}, \quad \forall \mu \in [\![1, N]\!].$$ (F.23)

It is now important to remark that

$$\frac{\partial S}{\partial A_\mu^{(p^2)}} = -l_\mu^{(p^2)} - \frac{p_\mu^2}{m}, \qquad \frac{\partial S}{\partial l_\mu^{(p^2)}} = A_\mu^{(p^2)} - \frac{1}{N} \sum_{\nu(\neq\mu)} \eta(\mu, \nu) s_\nu^2,$$

$$\frac{\partial S}{\partial A_\mu^{(sp)}} = -l_\mu^{(sp)} + s_\mu p_\mu, \qquad \frac{\partial S}{\partial l_\mu^{(sp)}} = A_\mu^{(sp)} - \frac{1}{N} \sum_{\nu(\neq\mu)} \eta(\mu, \nu) s_\nu p_\nu,$$ (F.24)

and

$$\langle s_\mu^2 \rangle_{\vec{s},\vec{p}} - \langle s_\mu \rangle_{\vec{s},\vec{p}}^2 = M_\mu^{-1(ss)}, \qquad \langle p_\mu^2 \rangle_{\vec{s},\vec{p}} - \langle p_\mu \rangle_{\vec{s},\vec{p}}^2 = M_\mu^{-1(pp)},$$

$$\langle s_\mu p_\mu \rangle_{\vec{s},\vec{p}} - \langle s_\mu \rangle_{\vec{s},\vec{p}} \langle p_\mu \rangle_{\vec{s},\vec{p}} = M_\mu^{-1(sp)}, \langle s_N \rangle_{\vec{s},\vec{p}} = -h_s M_N^{-1(ss)},$$ (F.25)

where we used the definition

$$\langle \ldots \rangle_{\vec{s},\vec{p}} = \int d\vec{s} \, d\vec{p} \, e^{-S(\vec{s},\vec{p},\vec{\varphi},\tilde{m},\tilde{h})}.$$ (F.26)

The integration over $\vec{s}$ and $\vec{p}$ is quadratic and can be performed. The GGE partition function becomes (up to sub-extensive contributions)

$$Z_{\text{GGE}} \propto \int d\vec{\varphi} \, d\tilde{m} \, d\tilde{h} \, e^{-S(\vec{\varphi},\tilde{m},\tilde{h})},$$ (F.27)

with

$$S(\vec{\varphi}, \tilde{m}, \tilde{h}) = -N \int d\lambda \, \rho(\lambda) \left[ l^{(sp)}(\lambda) A^{(sp)}(\lambda) + l^{(p^2)}(\lambda) A^{(p^2)}(\lambda) \right]$$

$$+ \frac{N}{2} \int d\lambda \, \rho(\lambda) \ln \det M(\lambda) - \left[ l_N^{(sp)} A_N^{(sp)} + l_N^{(p^2)} A_N^{(p^2)} \right]$$

$$+ \frac{1}{2} \ln \det M_N - h_s \tilde{m}_s - \frac{1}{2} h_s^2 M_N^{-1(ss)},$$ (F.28)

where we took the continuum limit for the sums over modes in the bulk.

At this stage it is important to clarify what will be our procedure to obtain the different saddle-points. As mentioned earlier, $\vec{\varphi}$ will take one value while $m_s$ and $h_s$ can be multi-valued. Thus, the strategy will be to start with the saddle-point equations for $\vec{\varphi}$ and then focus on $m_s$ and $h_s$. Some of the equations for $\vec{\varphi}$ will explicitly depend on $m_s$ and $h_s$. Therefore, we will average the $\vec{\varphi}$ saddle-point equations over $m_s$ and $h_s$. As an example if we have something of the form

$$\frac{\partial S}{\partial l_\mu^{(sp)}} \bigg|_{\vec{\varphi}, m_s, h_s} = 0 \implies g(\vec{\varphi}, m_s, h_s) = 0,$$ (F.29)

we will in fact solve the averaged equation

$$\langle g(\vec{\varphi}, m_s, h_s) \rangle_{m_s, h_s} = 0,$$ (F.30)

where $\langle \cdots \rangle_{m_s, h_s}$ is the average over all the possible saddle-points of $m_s$ and $h_s$.

*Saddles on $\vec{l}^{(sp)}$ and $\vec{A}^{(sp)}$*

To begin with, in the saddle-point approximation

$$\frac{\partial S}{\partial l_N^{(sp)}}\bigg|_{\vec{\varphi},m_s,h_s} = -A_N^{(sp)} + \frac{1}{m}\int d\lambda\,\rho(\lambda)\frac{\gamma_N - \gamma(\lambda)}{\lambda_N - \lambda}\langle s(\lambda)p(\lambda)\rangle_{\vec{s},\vec{p}} = 0\,. \tag{F.31}$$

As we expect $\langle s(\lambda)p(\lambda)\rangle_{\vec{s},\vec{p}} = l^{(sp)}(\lambda) = 0$ for each mode in the bulk we will take as a saddle-point $A_N^{(sp)} = 0$. We will see in the following that this guess is consistent with the other saddle-point equations. We then focus on the saddle-point equations:

$$\frac{\partial S}{\partial h_s}\bigg|_{\vec{\varphi},m_s,h_s} = m_s - \langle s_N\rangle_{\vec{s},\vec{p}} = 0\,, \tag{F.32}$$

$$\frac{\partial S}{\partial A_N^{(sp)}}\bigg|_{\vec{\varphi},m_s,h_s} = -l_N^{(sp)} + \langle s_N\rangle_{\vec{s},\vec{p}}\langle p_N\rangle_{\vec{s},\vec{p}} = -l_N^{(sp)} = 0\,. \tag{F.33}$$

Focusing now on the bulk we have

$$0 = \frac{1}{N}\frac{\partial S}{\partial l^{(sp)}(\lambda)}\bigg|_{\vec{\varphi},m_s,h_s} = -\rho(\lambda)A^{(sp)}(\lambda) + \frac{1}{2}\int d\lambda\,\rho(\lambda)\frac{\partial\det M(\lambda)}{\partial l^{(sp)}(\lambda)}\frac{1}{\det M(\lambda)}$$
$$+ \frac{1}{2N}\frac{\partial\det M_N}{\partial l^{(sp)}(\lambda)}\frac{1}{\det M(\lambda)} - \frac{1}{2N}\frac{\partial h_s^{\,2}M_N^{-1(ss)}}{\partial l^{(sp)}}\,, \tag{F.34}$$

and

$$\frac{1}{N}\frac{\partial S}{\partial A^{(sp)}(\lambda)}\bigg|_{\vec{\varphi},m_s,h_s} = -\rho(\lambda)l^{(sp)}(\lambda) + \frac{1}{2}\int d\lambda\,\rho(\lambda)\frac{\partial\det M(\lambda)}{\partial A^{(sp)}(\lambda)}\frac{1}{\det M(\lambda)} = 0\,.$$

With the input $l_N^{(sp)} = 0$ it is straightforward to observe that $l^{sp}(\lambda) = A^{sp}(\lambda) = 0$ is a solution of the saddle points equations. It enables to verify a posteriori the assumption we made previously, $\langle s(\lambda)p(\lambda)\rangle_{\vec{s},\vec{p}} = 0$. Consequently, the system of saddle-point equations reduces to

$$2\rho(\lambda)\langle s^2(\lambda)\rangle_{\vec{s},\vec{p}} = \left[\gamma(\lambda) + \frac{z}{J} + \int d\lambda'\rho(\lambda')\frac{\gamma(\lambda) - \gamma(\lambda')}{\lambda - \lambda'}l^{(p^2)}(\lambda') + \frac{\gamma(\lambda) - \gamma_N}{\lambda - \lambda_N}l_N^{(p^2)}\right]^{-1}\,,$$

$$\rho(\lambda)\langle p^2(\lambda)\rangle_{\vec{s},\vec{p}} = \left(\frac{-2}{m}A^{(p^2)}(\lambda)\right)^{-1}, \quad \langle s_N\rangle_{\vec{s},\vec{p}} = -h_s\left(2\gamma_N + \frac{z}{J}\right)^{-1},$$

$$\left\langle\left(s_N - \langle s_N\rangle_{\vec{s},\vec{p}}\right)^2\right\rangle_{\vec{s},\vec{p}} = \left(2\gamma_N + \frac{z}{J}\right)^{-1}, \quad \left\langle\left(p_N - \langle p_N\rangle_{\vec{s},\vec{p}}\right)^2\right\rangle_{\vec{s},\vec{p}} = \left(\frac{-2}{m}A_N^{(p^2)}\right)^{-1}.$$

*Saddles on $\vec{l}^{(p^2)}$ and $\vec{A}^{(p^2)}$:*

Finally, using Eq. (F.24), the saddles over $\vec{l}^{(p^2)}$ and $\vec{A}^{(p^2)}$ yield

$$\frac{1}{N}\frac{\partial S}{\partial l^{(p^2)}(\lambda)}\bigg|_{\vec{\varphi},m_s,h_s} = -\rho(\lambda)A^{(p^2)}(\lambda) + \int d\lambda'\rho(\lambda')\frac{\gamma(\lambda) - \gamma(\lambda')}{\lambda - \lambda'}\langle s^2(\lambda')\rangle_{\vec{s},\vec{p}}$$
$$+ \frac{\gamma(\lambda) - \gamma_N}{N(\lambda - \lambda_N)}\langle s_N^2\rangle_{\vec{s},\vec{p},m_s,h_s} = 0\,,$$

$$\frac{1}{N}\frac{\partial S}{\partial A^{(p^2)}(\lambda)}\bigg|_{\vec{\varphi},m_s,h_s} = -\rho(\lambda)l^{(p^2)}(\lambda) - \rho(\lambda)\langle p^2(\lambda)\rangle_{\vec{s},\vec{p}} = 0\,, \tag{F.35}$$

$$\frac{\partial S}{\partial l_N^{(p^2)}}\bigg|_{\vec{\varphi},m_s,h_s} = -A_N^{(p^2)} + \frac{1}{m}\int d\lambda\,\rho(\lambda)\frac{\gamma_N - \gamma(\lambda)}{\lambda_N - \lambda}\langle s^2(\lambda)\rangle_{\vec{s},\vec{p}} = 0\,,$$

$$\frac{\partial S}{\partial A_N^{(p^2)}}\bigg|_{\vec{\varphi},m_s,h_s} = -l_N^{(p^2)} - \langle p_N^2\rangle_{\vec{s},\vec{p},m_s,h_s} = -l_N^{(p^2)} + \frac{1}{2A_N^{(p^2)}} = 0\,.$$

The equations in the bulk are equivalent to

$$2\rho(\lambda)\langle s^2(\lambda)\rangle_{\vec{s},\vec{p}} = \Big[\gamma(\lambda) + \frac{z}{2J} - \frac{1}{m}\int d\lambda'\,\rho(\lambda')\frac{\gamma(\lambda)-\gamma(\lambda')}{\lambda-\lambda'}\langle p^2(\lambda')\rangle_{\vec{s},\vec{p}}$$
$$-\frac{1}{m}\frac{\gamma(\lambda)-\gamma_N}{N(\lambda-\lambda_N)}\langle p_N^2\rangle_{\vec{s},\vec{p},m_s,h_s}\Big]^{-1},$$
$$\frac{2}{m}\rho(\lambda)\langle p^2(\lambda)\rangle_{\vec{s},\vec{p}} = -\Big[\int d\lambda'\,\rho(\lambda')\frac{\gamma(\lambda)-\gamma(\lambda')}{\lambda-\lambda'}\langle s^2(\lambda')\rangle_{\vec{s},\vec{p}} + \frac{\gamma(\lambda)-\gamma_N}{N(\lambda-\lambda_N)}\langle s_N^2\rangle_{\vec{s},\vec{p},m_s,h_s}\Big]^{-1}.$$

The same harmonic *Ansatz* can be proposed with the condition

$$\langle p_N^2\rangle_{\vec{s},\vec{p},m_s,h_s} = (z-\lambda_N)\langle s_N^2\rangle_{\vec{s},\vec{p},m_s,h_s}. \tag{F.36}$$

Using Eqs. (F.32) and (F.35) the terms in the action which depend on $m_s$ and $h_s$ explicitly read, in the thermodynamic limit,

$$S_{m_s,h_s}(\vec{\varphi},\tilde{m},\tilde{h}) = m_s^2\Big[\frac{z}{2J} + \gamma_N + 2\int d\lambda'\,\rho(\lambda')\frac{\gamma_N-\gamma(\lambda')}{\lambda_N-\lambda'}l^{(p^2)}(\lambda')\Big]$$
$$= \frac{m_s^2}{\langle s_N^2\rangle_{\vec{s},\vec{p},m_s,h_s} - \langle m_s^2\rangle_{m_s,h_s}}, \tag{F.37}$$

with again the condition for the saddle points:

$$\langle p_N^2\rangle_{\vec{s},\vec{p},\tilde{m},\tilde{h}} = (z-\lambda_N)\langle s_N^2\rangle_{\vec{s},\vec{p},\tilde{m},\tilde{h}}. \tag{F.38}$$

Finally with the harmonic *Ansatz* the action in the bulk becomes

$$S_{\text{bulk}}(\vec{s},\vec{p},T(\lambda),z) = N\int d\lambda\,\rho(\lambda)\frac{1}{T(\lambda)}\Big[\frac{p^2(\lambda)}{2m} + (z-\lambda)s^2(\lambda)\Big]. \tag{F.39}$$

We end here this detailed calculation of the GGE partition function. The analysis of the results and its physical implications can be found in Sec. 7.2.

# Acknowledgements

D. Barbier and L. F. Cugliandolo thank C. Aron and the Les Houches Oxy-jeunes meeting on Quantum Physics where very useful discussions with V. Kazakov were carried out. We are also grateful to A. Gambassi and G. Schehr for very useful suggestions.

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
