# Peer review of "Generalised Gibbs Ensemble for spherically constrained harmonic models"

_SciPost Physics, doi:SciPost Phys. 13, 048 (2022)_

## Round 2 · Referee Report · Anonymous (Referee 1) · 2022-6-1

Strengths

1- One of the few works on GGE and equilibration in classical integrable systems.
2- Rigorous analysis and careful numerical check.
3- Extensive study of a paradigmatic model.

Weaknesses

1- No major weaknesses except for the paper length.

Report

This paper presents a detailed investigation of out-of-equilibrium dynamics in the Soft Neumann Model. This is a paradigmatic model that has been very useful to understand equilibrium properties of classical systems. Indeed, the model is solvable but it is nontrivial as it is non mean field and not free. While the model has been studied extensively at equilibrium, its out-of-equilibrium dynamics has not been explored much. The authors investigate the effect of quantum quenches in the SNM. Their main result is an analytic derivation of the GGE that describes the local properties of the steady state. The analysis performed in the paper is very careful, also as pertains the numerical checks. The paper is well written although quite long. I think that this paper provides a solid reference for future research in this field. Therefore, I recommend publication in the present form.

Requested changes

At some places some sentences are in green, perhaps the authors want to correct that.

  • validity: top
  • significance: high
  • originality: high
  • clarity: good
  • formatting: excellent
  • grammar: excellent

Author:  Nicolas Nessi  on 2022-07-05  [id 2637]

(in reply to Report 1 on 2022-06-01)
Category:
answer to question
reply to objection

We thank the referee for the careful reading of our manuscript, we have removed the green color in some of the sentences of the paper.

---

## Round 2 · Referee Report · Anonymous (Referee 2) · 2022-6-8

Strengths

Impressive work on an exactly solvable model where the GGE can be worked out and studied in full detail.

This is clearly an on-going work with quite some history already but the authors make an effort to guide a reader to the main points and to recall necessary background.

Weaknesses

none

Report

This work is in a long series of studies on non-equilibrium dynamics of spin glasses. Here this framework is used to formulate and solve the construction of a GGE through the explicit handling and the conserved quantities. This works presents a veritable tour de force through non-equilibrium dynamics and does advance the subject.

To be accepted for publication.

Requested changes

Some minor remarks for consideration and eventual changes:

1. on p. 9 should one not have 2 Lagrange multipliers for the 2 constraints (6) ?
Or else do you assume from the outset such initial conditions that one of the constraints is automatic ? Please explain.

2. in eq. (11) it may be helpful to remind the reader that { , } are Poisson brackets (and not, e.g. anti-commutators)

3. in section 3.3 or Figure 3 (and elsewhere) you call the average <I(lambda)> a `spectrum'. I find this confusing, since I would naturally think of the eigenvalues of some operator, which I fail to see. Or is this long-standing jargon in this special type of study ?

4. in the phase diagram Figure 4a, all phases I, IIa, IV are white, in contrast what can be gleaned from the text, where it should only be phase I.
It is not directly clear if the phase IV goes up to T_0/J_0 =1 or not.
Please edit the colour coding of the phases.

5. in figure 4b, where is phase IV ? Or is this just meant to be the reproduction of an old figure from [10] ? Please explain.

6. similarly, at the end of p. 19 "... and a new one that we recognise here ..."
maybe the reader should be directly directed to figure 5 where phase IV appears.

  • validity: top
  • significance: top
  • originality: top
  • clarity: high
  • formatting: perfect
  • grammar: excellent

Author:  Nicolas Nessi  on 2022-07-05  [id 2636]

(in reply to Report 2 on 2022-06-08)
Category:
answer to question
reply to objection

We thank the referee for the careful reading of the manuscript, and for her/his comments that we address below.

1- We thank the referee for bringing up this issue. In page 9, in equation (9) specifically, we define the Hamiltonian that governs the dynamics of our system. It turns out that the Lagrange multiplier corresponding to the secondary constraint vanishes identically: imposing the primary constraint on the dynamics automatically enforces the secondary constraint, be it exactly or on average. This implies that if we chose suitable initial conditions satisfying the primary and secondary constraint, the dynamics generated by (9) will preserve both. The situation is different when we are dealing with measures over phase space, as in page 19, equation (45), where we have to introduce the Lagrange multiplier corresponding to the secondary constraint in order to pick up configurations that indeed satisfy both constraints. We added an explanatory statement around this issue after Eq. (9).

2- We have explicitly stated that the brackets denote the Poisson bracket.

3- We call the $\lambda_{\mu}$’s “spectrum” because they can be considered as the eigenvalues of a random symmetric interaction matrix $J_{i,j}$, in particular, when we consider the connection with the Sherrington-Kirkpatrick model (section 2.4 and Appendix C).

4- The color code of Fig. 4a refers only to the regions of the phase diagram in which $\langle I_N \rangle$ is larger (dashed blue) or smaller (white) than zero, we do not intend to show the limits between the different phases in this particular figure, this is done in Fig. 5.

5- We thank the referee for raising this point. Yes, indeed, Fig. 4a is a reproduction of the phase diagram that we were able to determine from our previous analysis in Ref. [10], where, using the magnitudes that we studied back then, we could not differentiate phase I from phase IV. We have added an explanatory sentence about this in the caption of Fig. 4b.

6- We added a reference to Section V and figure 5 for clarity.

---

## Round 3 · List of Changes

* We have removed the green color text from some of the sentences of the manuscript.
* We added an explanatory statement around the issue of the secondary constrain raised by referre 2 after Eq. (9).
* We have explicitly stated that the brackets denote the Poisson bracket.
* In the caption of Fig. 4b We have added an explanatory sentence about the fact that in Fig. 4 we reproduce the phase diagram determined in a previous publication, in which we were not able to differentiate phase I from phase IV.

---

## Editorial Decision

published